# DIFFUSION & ADVERSARIAL SCHRÖDINGER BRIDGES VIA ITERATIVE PROPORTIONAL MARKOVIAN FITTING

**Sergei Kholkin**[*†1]    **Grigoriy Ksenofontov**[*1,3]    **David Li**[*4]    **Nikita Kornilov**[*1,3,7]

**Nikita Gushchin**[*1,2]          **Alexandra Suvorikova**[5,6]          **Alexey Kroshnin**[5]

**Evgeny Burnaev**[1,2]                              **Alexander Korotin**[1,2]

[1]Applied AI Institute[‡], [2]AXXX[‡],
[3]Moscow Independent Research Institute of Artificial Intelligence[‡],
[4]Mohamed bin Zayed University of Artificial Intelligence,
[5]Weierstrass Institute for Applied Analysis and Stochastics,
[6]Mathematical Foundations of Machine Learning Institute[‡],
[7]Basic Research of Artificial Intelligence Laboratory[‡].

## ABSTRACT

The Iterative Markovian Fitting (IMF) procedure, which iteratively projects onto the space of Markov processes and the reciprocal class, successfully solves the Schrödinger Bridge (SB) problem. However, an efficient practical implementation requires a heuristic modification—alternating between fitting forward and backward time diffusion at each iteration. This modification is crucial for stabilizing training and achieving reliable results in applications such as unpaired domain translation. Our work reveals a close connection between the modified version of IMF and the Iterative Proportional Fitting (IPF) procedure—a foundational method for the SB problem, also known as Sinkhorn's algorithm. Specifically, we demonstrate that the heuristic modification of the IMF effectively integrates both IMF and IPF procedures. We refer to this combined approach as the Iterative Proportional Markovian Fitting (IPMF) procedure. Through theoretical and empirical analysis, we establish the convergence of the IPMF procedure under various settings, contributing to developing a unified framework for solving SB problems. Moreover, from a practical standpoint, the IPMF procedure enables a flexible trade-off between image similarity and generation quality, offering a new mechanism for tailoring models to specific tasks. The code for our method can be found at: `https://github.com/gregkseno/ipmf`.

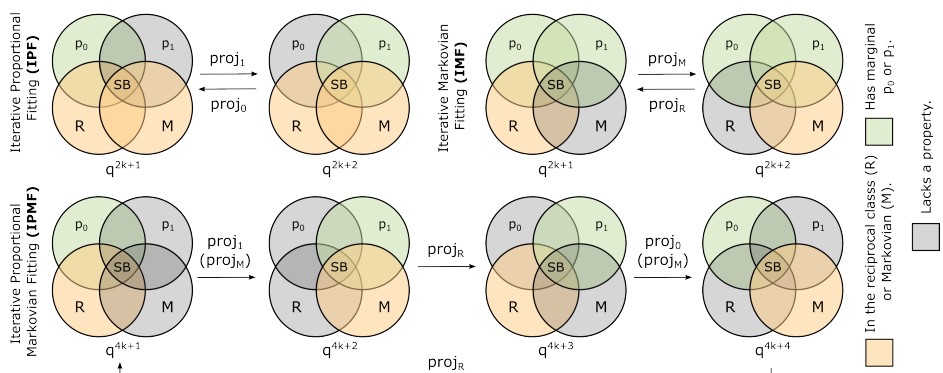

Figure 1: Diagrams of IPF, IMF, and unified IPMF procedure. All procedures aim to converge to the Schrödinger Bridge, i.e., a Markovian process in the reciprocal class, with marginals $p_0$ and $p_1$.

---

[*]Equal contribution.
[†]Corresponding author: `<kholkinsd@gmail.com>`.
[‡]Moscow, Russia.

# 1 INTRODUCTION

Diffusion Bridge models inspired by the Schrödinger Bridge (SB) theory, which connects stochastic processes with optimal transport, have recently become powerful approaches in biology (Tong et al., 2024; Bunne et al., 2023), chemistry (Somnath et al., 2023; Igashov et al.; Kim et al., 2024), image processing (Liu et al., 2023a; Shi et al., 2023; Korotin et al.; Gushchin et al., 2024a) and speech processing (Chen et al., 2023b). Most of these applications deal with either supervised domain translation, e.g., image super-resolution and inpainting (Liu et al., 2023a) or with unpaired translation, e.g., image style-transfer (Shi et al., 2023) or single-cell data analysis (Tong et al., 2024).

This work specifically focuses on *unpaired* domain translation (Zhu et al., 2017, Fig. 2). In this setting, given two domains represented solely by unpaired samples, the goal is to transform a sample from the input domain into a sample related to it in the target domain. In this context, researchers usually use SB-based algorithms because they enforce **two** key properties: *the optimality property*, ensuring similarity between the input and the translated object, and the *marginal matching property*, ensuring the translation of the input domain to the target domain. The motivation for relying on such specialized methods, rather than general text-to-data models, is further discussed in Appx. B.

Early works (De Bortoli et al., 2021; Vargas et al., 2021; Chen et al., 2021; Pavon et al., 2021) on using the SB for unpaired domain translation employed the well-celebrated **Iterative Proportional Fitting** (**IPF**) procedure (Kullback, 1968), also known as the Sinkhorn algorithm (Cuturi & Doucet, 2014). The IPF procedure is initialized with a simple prior process satisfying the optimality property. It then refines this process iteratively through optimality-preserving transformations until the marginal matching property is achieved. In each iteration, IPF decreases the *forward* KL-divergence $\text{KL}(q^*\|q)$ between the current approximation $q$ and the ground-truth Schrödinger Bridge $q^*$. However, in practice, approximation errors may cause IPF to suffer from the "prior forgetting", where the marginal matching property is achieved but optimality is lost (Vargas et al., 2024; 2021).

The **Iterative Markovian Fitting** (**IMF**) procedure (Shi et al., 2023; Peluchetti, 2023a; Gushchin et al., 2024b; Ksenofontov & Korotin, 2025) emerged as a promising competitor to IPF. Contrary to IPF, IMF starts from a stochastic process satisfying the marginal matching property and iteratively achieving optimality. Each iteration of IMF decreases the *reverse* KL-divergence $\text{KL}(q\|q^*)$ between the current approximation $q$ and the ground-truth SB $q^*$ (cf. with IPF). The approach generalizes rectified flows (Liu et al., 2022) to stochastic processes, which are employed (Liu et al., 2023b; Yan et al., 2024) in modern foundational models such as Stable Diffusion 3 (Esser et al., 2024). Like IPF, IMF may also accumulate errors. Specifically, it may fail to approximate data distributions due to an imperfect fit at each iteration, causing the marginal matching property to be lost.

In practice, to stabilize IMF training, prevent error accumulation and loss of marginal matching property, practitioners use a heuristic modification of IMF. This is a bidirectional procedure alternating between learning forward and backward processes, either by diffusion-based models in the **Diffusion** Schrödinger Bridge Matching (DSBM) algorithm (Shi et al., 2023) or GANs in **Adversarial** Schrödinger Bridge Matching (ASBM) algorithm (Gushchin et al., 2024b). In this work, we investigate the properties of the heuristic modification of the IMF. **Our contributions**:

1. **Theory.** We show that the heuristic bidirectional IMF procedure used in practice is closely related to IPF—in fact, it *secretly* uses IPF iterations. Therefore, we propose calling the bidirectional IMF procedure **Iterative Proportional Markovian Fitting** (IPMF, §3.1). We prove that the IPMF procedure *exponentially* converges for Gaussians under various settings. We also guarantee that IPMF converges to $q^*$, if $p_0$ and $p_1$ have bounded supports and *conjecture* that IPMF converges under very general settings, offering a promising way of developing a unified framework for solving the SB problem (§3.2).

2. **Practice I.** We empirically validate our conjecture through a series of experiments, including the Gaussian setup (§4), toy 2D setups (§4.2), the Schrödinger Bridge benchmark (§4.3), setup with real-world colored MNIST and CelebA image data (§4.4).

3. **Practice II.** Thanks to the proposed IPMF framework, we introduce a novel way to trade-off between generation quality and input-output similarity of Schrödinger Bridge solvers by designing the starting coupling. Empirically, we demonstrate on real-world image data that the proposed initializations outperform classical ones (§4.4).

These contributions demonstrate that the IPMF procedure has significant potential to **unify** a range of previously introduced SB methods—including IPF and IMF-based ones—in both discrete (Gushchin

et al., 2024b; De Bortoli et al., 2021) and continuous time (Shi et al., 2023; Peluchetti, 2023a; Vargas et al., 2021) settings, as well as their online versions (De Bortoli et al., 2024; Peluchetti, 2025; Karimi et al., 2024). Furthermore, the forward-backward IPMF framework could enable rectified flows to avoid error accumulation, making them even more powerful in generative modeling.

**Notations.** $\mathcal{P}_{2,ac}(\mathbb{R}^D)$ is a set of absolutely continuous distributions on $\mathbb{R}^D$ with finite second moment and finite entropy. We fix $N \geq 1$ intermediate time moments and set $0 = t_0 < t_1 < \cdots < t_N < t_{N+1} = 1$. Let $q \in \mathcal{P}_{2,ac}(\mathbb{R}^{D \times (N+2)})$ be an associated discrete stochastic process on this grid. For any such $q$, we denote the density at $(x_0, x_{t_1}, \ldots, x_{t_N}, x_1) \in \mathbb{R}^{D \times (N+2)}$ as $q(x_0, x_{\text{in}}, x_1)$, with $x_{\text{in}} = (x_{t_1}, \ldots, x_{t_N})$. $W^\epsilon$ is a Wiener process with volatility $\epsilon > 0$ and initial distribution $p_0$. Let $p^{W^\epsilon}$ be its discretization, i.e., $p^{W^\epsilon}(x_0, x_{\text{in}}, x_1) = p_0(x_0) \prod_{n=1}^{N+1} \mathcal{N}(x_{t_n} | x_{t_{n-1}}, \epsilon(t_n - t_{n-1})I_D)$, where $\mathcal{N}(\cdot|\cdot)$ is a conditional Gaussian distribution. $H(q)$ is the differential entropy of $q$.

## 2 BACKGROUND

This section details the study's key concepts; §2.1 introduces the Schrödinger Bridge (SB) problem, §2.2 presents the Iterative Proportional Fitting (IPF), §2.3 describes the Iterative Markovian Fitting (IMF), §2.4 discusses the heuristic modification of the IMF (Bidirectional IMF).

Recall that the SB problem (Schrödinger, 1931), IPF, and IMF admit both discrete– and continuous–time setups leading to the same problem solution. Moreover, the explicit formulas for IPF and IMF in the discrete setting are expressed in terms of probability densities, which helps to convey the main idea of our paper. Thus, for the sake of presentation flow, **the main text focuses exclusively on the discrete setup**, while Appendix C presents the continuous setup.

### 2.1 SCHRÖDINGER BRIDGE (SB) PROBLEM

The SB problem with a Wiener prior in the discrete-time setting (De Bortoli et al., 2021), given the initial distribution $p_0(x_0)$ and the final distribution $p_1(x_1)$, is stated as

$$\min_{q \in \Pi_N(p_0, p_1)} \text{KL}(q(x_0, x_{\text{in}}, x_1) \| p^{W^\epsilon}(x_0, x_{\text{in}}, x_1)), \tag{1}$$

where $\Pi_N(p_0, p_1) \subset \mathcal{P}_{2,ac}(\mathbb{R}^{D \times (N+2)})$ is the subset of discrete stochastic processes with marginals $q(x_0) = p_0(x_0)$, $q(x_1) = p_1(x_1)$. The objective function in (1) admits a decomposition

$$\text{KL}\big(q(x_0, x_{\text{in}}, x_1) \| p^{W^\epsilon}(x_0, x_{\text{in}}, x_1)\big) = \text{KL}\big(q(x_0, x_1) \| p^{W^\epsilon}(x_0, x_1)\big)$$
$$+ \int \text{KL}\big(q(x_{\text{in}} | x_0, x_1) \| p^{W^\epsilon}(x_{\text{in}} | x_0, x_1)\big) q(x_0, x_1) dx_0 dx_1.$$

All $q(x_{\text{in}} | x_0, x_1)$ can be chosen independently of $q(x_0, x_1)$. Thus, we can consider $q(x_{\text{in}} | x_0, x_1) = p^{W^\epsilon}(x_{\text{in}} | x_0, x_1)$ and get $\text{KL}(q(x_{\text{in}} | x_0, x_1) \| p^{W^\epsilon}(x_{\text{in}} | x_0, x_1)) = 0$.

This leads to the **Static SB problem**:

$$\min_{q \in \Pi(p_0, p_1)} \text{KL}\big(q(x_0, x_1) \| p^{W^\epsilon}(x_0, x_1)\big), \tag{2}$$

where $\Pi(p_0, p_1) \subset \mathcal{P}_{2,ac}(\mathbb{R}^{D \times D})$ is the subset of joint distributions $q(x_0, x_1)$ s.t. $q(x_0) = p_0(x_0)$, $q(x_1) = p_1(x_1)$. Finally, the static SB objective can be expanded (Gushchin et al., 2023a, Eq. 7)

$$\text{KL}(q(x_0, x_1) \| p^{W^\epsilon}(x_0, x_1)) = \int \frac{\|x_1 - x_0\|^2}{2\epsilon} dq(x_0, x_1) - H(q(x_0, x_1)) + C, \tag{3}$$

that is equivalent to the objective of the *entropic optimal transport* (EOT) problem with the *quadratic cost* up to an additive constant (Cuturi, 2013; Peyré et al., 2019; Léonard, 2013; Genevay, 2019).

### 2.2 ITERATIVE PROPORTIONAL FITTING (IPF)

Early works on SB (Vargas et al., 2021; De Bortoli et al., 2021; Tang et al., 2024) propose computational methods based on the IPF procedure (Kullback, 1968). The IPF-based algorithm is started by setting the process $q^0(x_0, x_{\text{in}}, x_1) = p_0(x_0) p^{W^\epsilon}(x_{\text{in}}, x_1 | x_0)$. Then, the algorithm alternates between two types of IPF projections, $\text{proj}_1$ and $\text{proj}_0$, given by (De Bortoli et al., 2021, Prop. 2):

$$q^{2k+1} = \text{proj}_1 \big( \underbrace{q^{2k}(x_1) \prod_{n=0}^{N} q^{2k}(x_{t_n} | x_{t_{n+1}})}_{q^{2k}(x_1) q^{2k}(x_0, x_{\text{in}} | x_1)} \big) \overset{\text{def}}{=} p_1(x_1) \underbrace{\prod_{n=0}^{N} q^{2k}(x_{t_n} | x_{t_{n+1}})}_{q^{2k}(x_0, x_{\text{in}} | x_1)}, \tag{4}$$

$$q^{2k+2} = \mathrm{proj}_0\big(\underbrace{q^{2k+1}(x_0)\prod_{n=1}^{N+1}q^{2k+1}(x_{t_n}|x_{t_{n-1}})}_{q^{2k+1}(x_0)q^{2k+1}(x_{\mathrm{in}},x_1|x_0)}\big) \overset{\text{def}}{=} p_0(x_0)\underbrace{\prod_{n=1}^{N+1}q^{2k+1}(x_{t_n}|x_{t_{n-1}})}_{q^{2k+1}(x_{\mathrm{in}},x_1|x_0)}. \quad (5)$$

Thus, $\mathrm{proj}_1$ and $\mathrm{proj}_0$ replace marginal distributions $q(x_1)$ and $q(x_0)$ in $q(x_0,x_{\mathrm{in}},x_1)$ by $p_1(x_1)$ and $p_0(x_0)$ respectively. The constructed sequence $\{q^k\}$ converges to the solution of the SB problem $q^*$ and causes the forward KL-divergence $\mathrm{KL}(q^*\|q^k)$ to decrease monotonically at each iteration. In practice, since the prior process $p^{W^\epsilon}$ is used only for initialization, the imperfect fit may lead to a deviation from the SB solution at some iteration. This problem is called "prior forgetting" and was discussed in (Vargas et al., 2024, Appx. E.3). The authors of Vargas et al. (2021) consider a continuous analog of the IPF procedure using inversions of diffusion processes (see Appx. C.2).

## 2.3 Iterative Markovian Fitting (IMF)

The Iterative Markovian Fitting (IMF) procedure (Peluchetti, 2023a; Shi et al., 2023; Gushchin et al., 2024b) emerged as a strong competitor to the IPF procedure. In contrast to IPF, IMF does not suffer from the "prior forgetting". The procedure is initialized with any $q^0 \in \Pi_N(p_0, p_1)$. Then it alternates between reciprocal projection $\mathrm{proj}_\mathcal{R}$ and Markovian projection $\mathrm{proj}_\mathcal{M}$:

$$q^{2k+1} = \mathrm{proj}_\mathcal{R}(q^{2k}) \overset{\text{def}}{=} q^{2k}(x_0,x_1)p^{W^\epsilon}(x_{\mathrm{in}}|x_0,x_1), \quad (6)$$

$$q^{2k+2} = \mathrm{proj}_\mathcal{M}(q^{2k+1}) \overset{\text{def}}{=} \underbrace{q^{2k+1}(x_0)\prod_{n=1}^{N+1}q^{2k+1}(x_{t_n}|x_{t_{n-1}})}_{\text{forward representation}} = \underbrace{q^{2k+1}(x_1)\prod_{n=0}^{N}q^{2k+1}(x_{t_n}|x_{t_{n+1}})}_{\text{backward representation}} \quad (7)$$

The reciprocal projection $\mathrm{proj}_\mathcal{R}$ creates a new (in general, non-Markovian) process combining the distribution $q(x_0,x_1)$ and $p^{W^\epsilon}(x_{\mathrm{in}}|x_0,x_1)$. The latter is called the discrete Brownian Bridge. The Markovian projection $\mathrm{proj}_\mathcal{M}$ uses the set of transitional densities $\{q(x_{t_n}|x_{t_{n-1}})\}$ or $\{q(x_{t_n}|x_{t_{n+1}})\}$ to create a new Markovian process starting from $q(x_0)$ or $q(x_1)$ respectively. Markovian projection keeps the marginal distributions at each timestep, but, in general, changes the joint distributions between them. The sequence $\{q^k\}$ converges to the SB $q^*$ and causes the reverse KL-divergence $\mathrm{KL}(q^k\|q^*)$ to decrease monotonically at each iteration (cf. with IPF). The authors of Shi et al. (2023); Peluchetti (2023a) consider a continuous-time version of the IMF.

## 2.4 Heuristic bidirectional modification of IMF

The result of the Markovian projection (7) admits both forward and backward representation. To learn the corresponding transitional densities, one uses neural networks $\{q_\theta(x_{t_n}|x_{t_{n-1}})\}$ (**forward parametrization**) or $\{q_\phi(x_{t_n}|x_{t_{n+1}})\}$ (**backward parametrization**). The starting distributions are as follows: $q_\theta(x_0) = p_0(x_0)$ for the forward parametrization and $q_\phi(x_1) = p_1(x_1)$ for the backward parametrization. In practice, the alternation between representations of Markovian processes is used in both implementations of continuous-time IMF by **DSBM** algorithm (Shi et al., 2023, Alg. 1) based on diffusion models and discrete-time IMF by **ASBM** algorithm (Gushchin et al., 2024b, Alg. 1) based on the GANs. This **bidirectional** procedure can be described as follows:

$$q^{4k+1} = \underbrace{q^{4k}(x_0,x_1)p^{W^\epsilon}(x_{\mathrm{in}}|x_0,x_1)}_{\mathrm{proj}_\mathcal{R}(q^{4k})}, \quad q^{4k+2} = p(x_1)\underbrace{\prod_{n=0}^{N}q_\phi^{4k+1}(x_{t_{n-1}}|x_{t_n})}_{\text{backward parametrization}}, \quad (8)$$

$$q^{4k+3} = \underbrace{q^{4k+2}(x_0,x_1)p^{W^\epsilon}(x_{\mathrm{in}}|x_0,x_1)}_{\mathrm{proj}_\mathcal{R}(q^{4k+2})}, \quad q^{4k+4} = p(x_0)\underbrace{\prod_{n=1}^{N+1}q_\theta^{4k+3}(x_{t_n}|x_{t_{n-1}})}_{\text{forward parametrization}}. \quad (9)$$

Thus, only one marginal is fitted perfectly, e.g., $q_\theta(x_0) = p_0(x_0)$ in the case of forward representation, while the other marginal is only learned, e.g., $q_\theta(x_1) = \int p_0(x_0)\prod_{n=1}^{N+1}q_\theta(x_{t_n}|x_{t_{n-1}})dx_0dx_1\cdots dx_N \approx p_1(x_1)$.

## 3 Iterative Proportional Markovian Fitting

This section demonstrates that the heuristic bidirectional IMF (§2.4) is, in fact, the alternating implementation of IPF and IMF projections. §3.1 establishes that this heuristic defines the unified Iterative

Proportional Markovian Fitting (IPMF) procedure. §3.2 provides the analysis of the convergence of the IPMF procedure under various settings, with the proofs provided in Appendix D.

## 3.1 BIDIRECTIONAL IMF IS IPMF

For a given Markovian process $q$, we recall that its IPF projections ($\text{proj}_0(q)$ (5) and $\text{proj}_1(q)$ (4)) replace the starting distribution $q(x_0)$ with $p_0(x_0)$ and $q(x_1)$ with $p_1(x_1)$, respectively. Further, the process $q^{4k+2}$ (8) is a result of a combination of the Markovian projection $\text{proj}_{\mathcal{M}}$ (7) in forward parametrization and of the IPF projection $\text{proj}_1$ (4):

$$q^{4k+2} = p(x_1) \prod_{n=0}^{N} q^{4k+1}(x_{t_n}|x_{t_{n+1}}) = \underbrace{\text{proj}_1\Big(q^{4k+1}(x_1) \prod_{n=0}^{N} q^{4k+1}(x_{t_n}|x_{t_{n+1}})\Big)}_{\text{proj}_1(\text{proj}_{\mathcal{M}}(q^{4k+1}))}.$$

Next, the process $q^{4k+4}$ (9) results from a combination of the Markovian projection $\text{proj}_{\mathcal{M}}$ (7) in backward parametrization and of the IPF projection $\text{proj}_0$ (5):

$$q^{4k+3} = p(x_0) \prod_{n=1}^{N+1} q^{4k+3}(x_{t_n}|x_{t_{n-1}}) = \underbrace{\text{proj}_0\big(q^{4k+3}(x_0) \prod_{n=1}^{N+1} q^{4k+3}(x_{t_n}|x_{t_{n-1}})\big)}_{\text{proj}_0(\text{proj}_{\mathcal{M}}(q^{4k+3}))}.$$

Thus, we can represent the heuristic bidirectional IMF given by (9) and (8) as follows:

**Iterative Proportional Markovian Fitting (Discrete time)**

$$q^{4k+1} = \underbrace{q^{4k}(x_0, x_1)p^{W^\epsilon}(x_{\text{in}}|x_0, x_1)}_{\text{proj}_{\mathcal{R}}(q^{4k})}, \quad q^{4k+2} = p(x_1) \underbrace{\prod_{n=0}^{N} q^{4k+1}(x_{t_{n-1}}|x_{t_n})}_{\text{proj}_1(\text{proj}_{\mathcal{M}}(q^{4k+1}))},$$

$$q^{4k+3} = \underbrace{q^{4k+2}(x_0, x_1)p^{W^\epsilon}(x_{\text{in}}|x_0, x_1)}_{\text{proj}_{\mathcal{R}}(q^{4k+2})}, \quad q^{4k+4} = p(x_0) \underbrace{\prod_{n=1}^{N+1} q^{4k+3}(x_{t_n}|x_{t_{n-1}})}_{\text{proj}_0(\text{proj}_{\mathcal{M}}(q^{4k+3}))}.$$

The heuristic bidirectional IMF alternates between two IMF projections ($\text{proj}_{\mathcal{M}}(\text{proj}_{\mathcal{R}}(\cdot))$) during which the process "became more optimal" (step towards optimality property) and two IPF projections ($\text{proj}_0$ and $\text{proj}_1$) during which the marginal fitting improves (step towards marginal matching property). We refer to this procedure as **Iterative Proportional Markovian Fitting (IPMF)**. *An IPMF step consists of two IMF projections and two IPF projections.* We hypothesize that IPMF converges from any initial process $q^0(x_0, x_{\text{in}}, x_1)$, unlike IPF and IMF, which require a specific form of the starting process. We emphasize that IPMF reduces to IMF when the initial coupling has the correct marginals $p_0$ and $p_1$ and has Brownian Bridge between the marginals. Similarly, if the initial coupling is Markovian, is in the reciprocal class, and has the correct initial marginal $p_0$ or $p_1$, then IPMF reduces to IPF. Fig. 1 visualizes these cases, clarifying the role of the initial coupling and the iterative steps. A similar analysis for continuous-time IPMF is provided in Appx. C.3 and the summarization of the conceptual differences between prior bidirectional SB heuristics and the proposed IPMF framework is presented in Table 4.

## 3.2 THEORETICAL CONVERGENCE ANALYSIS IN VARIOUS CASES

Our first result introduces a novel approach to quantify the optimality property for a Gaussian plan. We show that any $2D$ Gaussian distribution ($D \geq 1$) is an entropic OT plan between its marginals for a certain transport cost. Let $Q, S \in \mathbb{R}^{D \times D}$ be positive definite matrices ($Q, S \succ 0$) and $P \in \mathbb{R}^{D \times D}$ be s.t. $Q - P(S)^{-1}P^\top \succ 0$. Define

$$\Xi(P, Q, S) \stackrel{\text{def}}{=} (S)^{-1}P^\top (Q - P(S)^{-1}P^\top)^{-1}. \tag{10}$$

**Theorem 3.1.** *Let $q(x_0, x_1)$ be Gaussian with marginals $p = \mathcal{N}(\eta, Q)$ and $\tilde{p} = \mathcal{N}(\nu, S)$,*

$$q(x_0, x_1) = \mathcal{N}\left(\begin{pmatrix} \eta \\ \nu \end{pmatrix}, \begin{pmatrix} Q & P \\ P^\top & S \end{pmatrix}\right).$$

*Let $A = \Xi(P, Q, S)$. Then $q$ is the unique minimizer of*

$$\min_{q' \in \Pi(p, \tilde{p})} \left\{ \int (-x_1^\top A x_0) \cdot q'(x_0, x_1) dx_0 dx_1 - H(q') \right\}. \tag{11}$$

Problem (11) is the OT problem with the transport cost $c_A(x_0, x_1) := -x_1^\top A x_0$ and entropy regularization (with weight 1) (Cuturi, 2013; Genevay, 2019). In other words, for any $2D$ Gaussian distribution $q$, there exists a matrix $A(q) \in \mathbb{R}^{D \times D}$ that defines the cost function for which $q$ solves the EOT problem. We name $A(q)$ the **optimality matrix**. If $q$ is such that $A(q) = \epsilon^{-1} I_D$, then the corresponding transport cost is $c_A(x_0, x_1) = -\epsilon^{-1} \cdot \langle x_1, x_0 \rangle$ which is equivalent to $\epsilon^{-1} \cdot \|x_1 - x_0\|^2 / 2$. Consequently, $q$ is the static SB (2) between its $q_0(x_0)$ and $q_1(x_1)$ for the prior $W^\epsilon$, recall (3).

**Main result.** We prove the exponential convergence of IPMF (w.r.t. the parameters) to the solution $q^*$ of the static SB problem (2) between $p_0$ and $p_1$ under certain settings.

**Theorem 3.2** (Convergence of IPMF for Gaussians)**.** *Let $p_0 = \mathcal{N}(\mu_0, \Sigma_0)$ and $p_1 = \mathcal{N}(\mu_1, \Sigma_1)$ be $D$-dimensional Gaussians. Assume that we run IPMF with $\epsilon > 0$, starting from some $2D$ Gaussian*[1]

$$q^0(x_0, x_1) = \mathcal{N}\left( \begin{pmatrix} \mu_0 \\ \nu \end{pmatrix}, \begin{pmatrix} \Sigma_0 & P_0 \\ P_0 & S_0 \end{pmatrix} \right) \in \mathcal{P}_{2,ac}(\mathbb{R}^D \times \mathbb{R}^D).$$

*We denote the distribution obtained after $k$ IPMF steps by*
$$q^{4k}(x_0, x_1) \stackrel{def}{=} \mathcal{N}\left( \begin{pmatrix} \mu_0 \\ \nu_k \end{pmatrix}, \begin{pmatrix} \Sigma_0 & P_k \\ P_k & S_k \end{pmatrix} \right) \in \mathcal{P}_{2,ac}(\mathbb{R}^D \times \mathbb{R}^D)$$

*and $A_k \stackrel{def}{=} \Xi(P_k, \Sigma_0, S_k)$. Then in the following settings*

- *$D = 1$, discrete- or continuous-time IMF ($N = 1$), any $\epsilon > 0$;*

- *$D > 1$, discrete-time IMF, $\epsilon \gg 0$ (see Appendix D.4);*

*the following exponential convergence bounds hold:*
$$\|S_k^{-\frac{1}{2}} \Sigma_1 S_k^{-\frac{1}{2}} - I_D\|_2 \leq \alpha^{2k} \|S_0^{-\frac{1}{2}} \Sigma_1 S_0^{-\frac{1}{2}} - I_D\|_2,$$

$$\|\Sigma_1^{-\frac{1}{2}}(\nu_k - \mu_1)\|_2 \leq \alpha^k \|\Sigma_1^{-\frac{1}{2}}(\nu_0 - \mu_1)\|_2, \quad \|A_k - \epsilon^{-1} I_D\|_2 \leq \beta^{2k} \|A_0 - \epsilon^{-1} I_D\|_2, \quad (12)$$

*with $\alpha, \beta < 1$ and $\|\cdot\|_2$ being the spectral norm; $\alpha, \beta$ depend on IPMF type (discrete or continuous), initial parameters $S_0, \nu_0, P_0$, marginal distributions $p_0, p_1$, and $\epsilon$. Consequently,* $\text{KL}\left(q^{4k} \| q^*\right), \text{KL}\left(q^* \| q^{4k}\right) \stackrel{k \to \infty}{\to} 0.$

*Proof idea.* An IPF step does not change the "copula", i.e., the information about the joint distribution that is invariant w.r.t. changes in marginals $p_0$ and $p_1$. In the Gaussian case can be represented by the optimality matrix $A_k$ (see Lemma D.3). In contrast, an IMF iteration changes the copula but preserves the marginals. Next, we analyze closed formulas for the IMF step in the Gaussian case (Peluchetti, 2023a; Gushchin et al., 2024b) and show that the IMF step makes $A_k$ closer to $\epsilon^{-1} I_D$. Specifically, we verify the contractivity of each step w.r.t. $A_k$. $\square$

Our next result shows that the convergence of IPMF holds far beyond the Gaussian setting.

**Theorem 3.3** (Convergence of IPMF under boundness assumption)**.** *Assume $p_0$ and $p_1$ have bounded supports. Then for both discrete-time and continuous-time IPMF it holds $q^{4k}(x_0, x_1) \stackrel{w}{\to} q^*(x_0, x_1)$, where $\stackrel{w}{\to}$ denotes weak convergence.*

**General conjecture.** *Given our results, we believe that IPMF converges under very general settings (beyond the Gaussian and bounded cases). Moreover, in the Gaussian case, we expect exponential convergence for all $\epsilon > 0$, all $D$, and IMF types. We verify these claims experimentally (§4).*

**Related works and our novelty.** Our work provides the first theoretical analysis of bidirectional IMF, whereas prior studies analyzed only vanilla IMF. (Shi et al., 2023, Theorem 8) and (Gushchin et al., 2024b, Theorem 3.6) proved sublinear convergence of IMF in reverse KL divergence for continuous and discrete cases. For IPF, sublinear convergence in forward KL divergence under mild assumptions, as well as geometric convergence for Gaussians, are shown in (De Bortoli et al., 2021, Propositions 4 and 43). Previous results cannot be directly generalized to IPMF. First, IPF and IMF converge in different divergence measures, and a decrease in one does not imply a reduction in the other. Second, IPF updates marginals at each step, so IMF must optimize toward a moving target, whereas pure IMF has a fixed optimum. Unlike (Shi et al., 2023), which proves convergence only from the IPF starting coupling, our analysis applies to arbitrary starting couplings. We also view the starting coupling as a tunable hyperparameter and examine its effect in the next section.

---

[1]We assume that $q^0(x_0) = p_0(x_0)$, i.e., the initial process starts at $p_0$ at time $t = 0$. This is reasonable, as after the first IPMF round the process will satisfy this property thanks to the IPF projections involved.

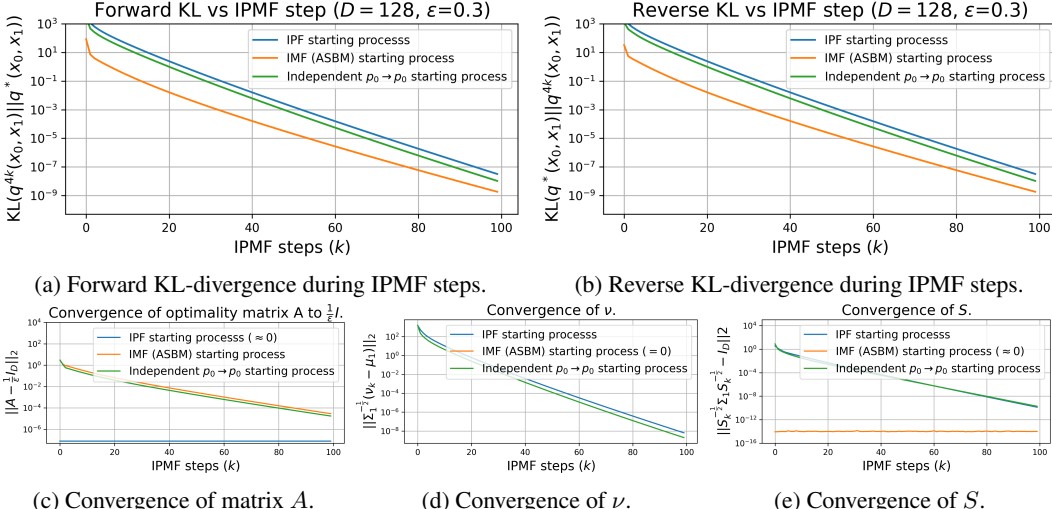

Figure 2: Convergence of IPMF procedure with different starting process $q^0$.

## 4 EXPERIMENTAL ILLUSTRATIONS

This section provides empirical evidence that IPMF converges under a more general setting—specifically, from any starting process—unlike IPF and IMF. The first goal is to achieve the same or similar results across all used starting couplings and for both discrete-time (ASBM) and continuous-time (DSBM) solvers on illustrative setups (§4.1, §4.2, §4.3, §4.4). The second one is to highlight that, while all initializations converge to qualitatively similar outcomes, in practice, some offer better generation quality and others better input-output similarity on real-world data, due to different starting points. This allows one to choose initializations based on specific task requirements (§4.4).

In §3.1 and Appx. C we show that the bidirectional IMF and the proposed IPMF differ only in the initial starting process. Since both practical implementations of continuous-time IMF (Shi et al., 2023, Alg. 1) and discrete-time IMF (Gushchin et al., 2024b, Alg. 1) use the considered bidirectional version, we use practical algorithms introduced in these works, i.e., Diffusion Schrödinger Bridge Matching (**DSBM**) and Adversarial Schrödinger Bridge Matching (**ASBM**) respectively.

**Experimental setups.** We consider multivariate Gaussian distributions for which we have closed-form IPMF update formulas, an illustrative 2D example, the Schrödinger Bridges Benchmark (Gushchin et al., 2023b) and real-life image data distributions, i.e., the colored MNIST dataset and the CelebA dataset (Liu et al., 2015b). All technical details can be found in the Appx. E.

**Starting processes.** We focus on running the IPMF procedure from various initializations, referred to as *starting processes*. The starting processes are constructed by selecting different couplings $q^0(x_0, x_1)$ and incorporating the Brownian Bridge process $W^\epsilon_{|x_0,x_1}$ (i.e., $W^\epsilon$ conditioned on $x_0, x_1$). In the discrete-time setup, for each selected coupling $q^0(x_0, x_1)$ we construct the starting process as $q^0(x_0, x_{\text{in}}, x_1) = q^0(x_0, x_1)p^{W^\epsilon}(x_{\text{in}}|x_0, x_1)$ and $T^0 = \int W^\epsilon_{|x_0,x_1} dq^0(x_0, x_1)$ for the continuous-time case (see Appx. C). We consider three "starting" scenarios: IMF-like starting process of the form $q^0(x_0, x_1) = p_0(x_0)p_1(x_1)$, IPF-like starting process of the form $q^0(x_0, x_1) = p_0(x_0)p^{W^\epsilon}(x_1|x_0)$, and various starting processes which cannot be used to initialize IMF or IPF. The latter demonstrates that IPMF converges under a more general setting.

The results of DSBM and ASBM with different starting processes are denoted as (D/A)SBM-*coupling*, e.g., DSBM-IMF for DSBM with IMF as the starting process.

**Remark.** Notably, in practice, both the IPF and IMF procedures can be recovered through different implementations. For example, IPF can be realized through (D/A)SBM with the IPF starting coupling, or alternatively via DSB (De Bortoli et al., 2021). IMF, in turn, can be implemented using (D/A)SBM with a one-directional parametrization. However, in practice, matching-based methods exhibit superior performance (Shi et al., 2023). Furthermore, the authors of (Shi et al., 2023; Peluchetti, 2023a; Gushchin et al., 2024b) observed that bidirectional IMF does not accumulate approximation errors, whereas relying solely on one direction parametrization leads to error accumulation and eventual divergence (De Bortoli et al., 2024, Appx. I). Therefore, we argue that a direct comparison between the IPMF procedure and previous practical implementations is unnecessary.

| | Algorithm Type | $\epsilon=0.1$ | | | | $\epsilon=1$ | | | | $\epsilon=10$ | | | |
|---|---|---|---|---|---|---|---|---|---|---|---|---|---|
| | | $D=2$ | $D=16$ | $D=64$ | $D=128$ | $D=2$ | $D=16$ | $D=64$ | $D=128$ | $D=2$ | $D=16$ | $D=64$ | $D=128$ |
| Best algorithm on benchmark† | Varies | 1.94 | 13.67 | 11.74 | 11.4 | 1.04 | 9.08 | 18.05 | 15.23 | 1.40 | 1.27 | 2.36 | **1.31** |
| DSBM-IMF | | 1.21 | 4.61 | 9.81 | 19.8 | 0.68 | **0.63** | 5.8 | 29.5 | 0.23 | 5.45 | 68.9 | 362 |
| DSBM-IPF | | 2.55 | 17.4 | 15.85 | 17.45 | 0.29 | 0.76 | **4.05** | 29.59 | 0.35 | 3.98 | 83.2 | 210 |
| DSBM-*Identity* | IPMF | 1.23 | 18.86 | 24.71 | 21.39 | 0.26 | 0.69 | 7.46 | 29.5 | 0.13 | 3.99 | 88.2 | 347 |
| ASBM-IMF† | | 0.89 | 8.2 | 13.5 | 53.7 | 0.19 | 1.6 | 5.8 | **10.5** | 0.13 | 0.4 | 1.9 | 4.7 |
| ASBM-IPF | | 3.06 | 14.37 | 44.35 | 32.5 | **0.18** | 1.68 | 9.25 | 20.47 | 0.13 | 0.36 | 2.28 | 4.97 |
| ASBM-*Identity* | | 0.58 | 24.9 | 29.1 | 85.2 | 0.19 | 2.44 | 8.28 | 11.61 | **0.12** | **0.35** | 1.66 | **2.86** |
| SF²M-Sink† | Bridge Matching | **0.54** | **3.7** | **9.5** | 10.9 | 0.2 | 1.1 | 9 | 23 | 0.31 | 4.9 | 319 | 819 |

Table 1: Comparisons of $cB\mathbb{W}_2^2$-UVP $\downarrow$ (%) between the static SB solution $q^*(x_0, x_1)$ and the learned solution on the SB benchmark. The best metric is **bolded**. Results marked with † are taken from (Gushchin et al., 2024b) and (Gushchin et al., 2023b). The results of DSBM and ASBM algorithms starting from different starting processes are denoted as (D/A)SBM-*name of starting process*

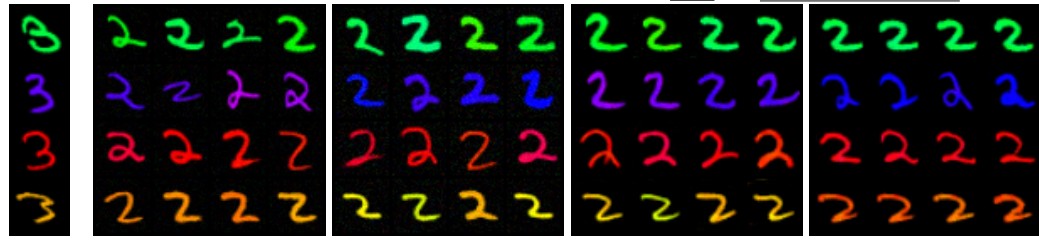

(a) $x \sim p_0$    (b) DSBM-IMF    (c) DSBM-*Inverted 7*    (d) ASBM-IMF    (e) ASBM-*Inverted 7*

Figure 3: Samples from DSBM and ASBM learned with IPMF using IMF and $q^{\mathrm{inv7}}$ starting processes on Colored MNIST *3→2* ($32 \times 32$) translation for $\epsilon = 10$.

### 4.1 HIGH DIMENSIONAL GAUSSIANS

This section experimentally validates the convergence of IPMF for the multivariate Gaussians (see our **General conjecture**, §3.2). We use explicit formulas for the discrete IPF and IMF (Gushchin et al., 2024b, Thm. 3.8) and follow the setup from (Gushchin et al., 2023a, Sec. 5.2). Specifically, we consider the Schrödinger Bridge (SB) problem with $D = 128$ and $\epsilon = 0.3$, where the marginal distributions are Gaussian: $p_0 = \mathcal{N}(\mathbf{0}, \Sigma_0)$ and $p_1 = \mathcal{N}(\mathbf{3}, \Sigma_1)$, with $\mathbf{0} \in \mathbb{R}^D$ denoting the vector of all zeros and $\mathbf{3} \in \mathbb{R}^D$ - the vector of all threes. The eigenvectors of $\Sigma_0$ and $\Sigma_1$ are sampled from the uniform distribution on the unit sphere. Their eigenvalues are sampled from the loguniform distribution on $[-\log 2, \log 2]$. We choose $N = 3$ intermediate time points uniformly between $t = 0$ and $t = 1$ and run 100 steps of the IPMF procedure, each consisting of two IPF projections and two Markovian–Reciprocal projections (see §3.1). Denote as $q^{4k} = q^{4k}(x_0, x_1)$ the IPMF output at the $k$-th step and let $q^* = q^*(x_0, x_1)$ be the solution of the static SB. Fig. 2 shows that both the forward KL$(q^*\|q^{4k})$ and reverse KL$(q^{4k}\|q^*)$ divergences converge. The quantities from (12) converge to zero exponentially, as expected. Note that particular starting processes can cause an immediate match of the parameters: an IMF starting process that already has the required marginals converges only in optimality, while IPF converges to the true marginals with the unchanged optimality matrix.

### 4.2 ILLUSTRATIVE 2D EXAMPLE

We consider the SB problem with $\epsilon = 0.1$, $p_0$ being a Gaussian distribution on $\mathbb{R}^2$ and $p_1$ being the Swiss roll. We train DSBM and ASBM algorithms using IMF and IPF starting processes. Additionally, we consider *Identity* starting processes induced by $q^0(x_0, x_1) = p_0(x_0)\delta_{x_0}(x_1)$, i.e. we set $x_1 = x_0$ after sampling $x_0 \sim p(x_0)$. The purpose of testing the Identity coupling is to verify that IPMF converges even when initialized with a naive coupling. Furthermore, we hypothesize that this coupling is the best in terms of the optimality property. We present the starting processes and the results in Fig. 7 in Appendix E. In all the cases, we observe similar results.

### 4.3 EVALUATION ON THE SB BENCHMARK

We use the SB mixtures benchmark (Gushchin et al., 2023b) with the ground truth solution to the SB problem to test ASBM and DSBM with IMF, IPF, and *Identity* (§4.2) as the starting processes. The benchmark provides continuous distribution pairs $p_0, p_1$ for dimensions $D \in \{2, 16, 64, 128\}$ that have known SB solutions for volatility $\epsilon \in \{0.1, 1, 10\}$. To evaluate the quality of the recovered SB solutions, we use the $cB\mathbb{W}_2^2$-UVP metric (Gushchin et al., 2023b). Tab. 1 provides the results. We also include the results of the standard baseline for Bridge Matching tasks called SF²M-Sink (Gushchin et al., 2023b). We provide training details and additional results in Appx. E. All starting processes yield similar results within each solver type (DSBM or ASBM).

### 4.4 UNPAIRED IMAGE-TO-IMAGE TRANSLATION

To test IPMF on real data, we consider two unpaired image-to-image translation setups: *colorized 3 → colorized 2* digits from the MNIST dataset with $32 \times 32$ resolution size and *male→female* faces from the CelebA dataset with $64 \times 64$ resolution size.

| | Initialisation (coupling) | | | | DSBM | | | | ASBM | | | |
|---|---|---|---|---|---|---|---|---|---|---|---|---|
| | IMF | DDPM SDEdit | SD SDEdit | Identity | IMF | DDPM SDEdit | SD SDEdit | Identity | IMF | DDPM SDEdit | SD SDEdit | Identity |
| FID↓ | 0.0 | 35.23 | 28.77 | 61.56 | **13.65** | 14.84 | 22.65 | 33.11 | **19.32** | 21.84 | 20.64 | 19.58 |
| MSE($x_0, \widehat{x}_1$)↓ | 0.16 | 0.02 | 0.02 | 0.0 | 0.16 | 0.09 | 0.04 | **0.03** | 0.17 | **0.07** | 0.08 | 0.07 |

Table 2: Qualitative results on CelebA ($64 \times 64$) for *male→female* translation with ASBM and DSBM across different starting processes. Generative quality (FID↓) and similarity (MSE($x_0, \widehat{x}_1$)↓) are reported on the test set. Best and second-best values for solvers are marked in **bold** and underline, respectively.

**Colored MNIST**. We construct train and test sets by RGB colorization of MNIST digits from corresponding train and test sets of classes "2" and "3". We train ASBM and DSBM algorithms starting from the IMF process. Additionally, we test a starting process induced by the independent coupling of the distribution of colored digits of class "3" ($p_0$) and the distribution of colored digits of class "7" with inverted RGB channels ($p^{\mathrm{inv}7}(x_1)$). We refer to this process as *Inverted 7*, i.e., $q^0(x_0, x_1) = p_0(x_0)p^{\mathrm{inv}7}(x_1)$ (see Fig. 10). Appx. E contains further technical details. We learn DSBM and ASBM on the *train* set of digits and visualize the translated *test* images (Fig. 3).

Both DSBM and ASBM algorithms starting from both IMF and *Inverted 7* starting process fit the target distribution of colored MNIST digits of class "2" and preserve the color of the input image during translation. This supports that the limiting behavior of IPMF resembles the solution of SB.

**CelebA.** We consider the IMF-OT variation of the IMF starting process. It is induced by a mini-batch optimal transport coupling $q^{\mathrm{OT}}(x_0, x_1)$ (Tong et al., 2024; Pooladian et al., 2023). We also test *Identity* (§4.2) starting process. Additionally, we test starting processes induced by *DDPM SDEdit* and *SD SDEdit* couplings, which is the SDEdit method (Meng et al., 2022) used for *male→female* translation with **(1)** DDPM (Ho et al., 2020) model trained on the female part of CelebA and **(2)** Stable Diffusion v1.5 (Rombach et al., 2022) with designed text prompt, see Appx. E.3. The aim of introducing such a coupling is to test the hypothesis that well-designed SDEdit couplings can improve the metrics of both properties. We use approximately the same number of parameters for the DSBM and the ASBM generators and 10% of images for evaluation (see Appx. E, other details).

We provide qualitative results in Fig. 4. Additionally, we report the final FID score (generation quality) and the Mean Squared Error (MSE) between the input $x_0$ and the translated image $\widehat{x}_1$ (input-output similarity) in Table 2. Figure 4 illustrates that the models (1) converge to the target distribution and (2) preserve semantic alignment between input and output (e.g., hair color, background).

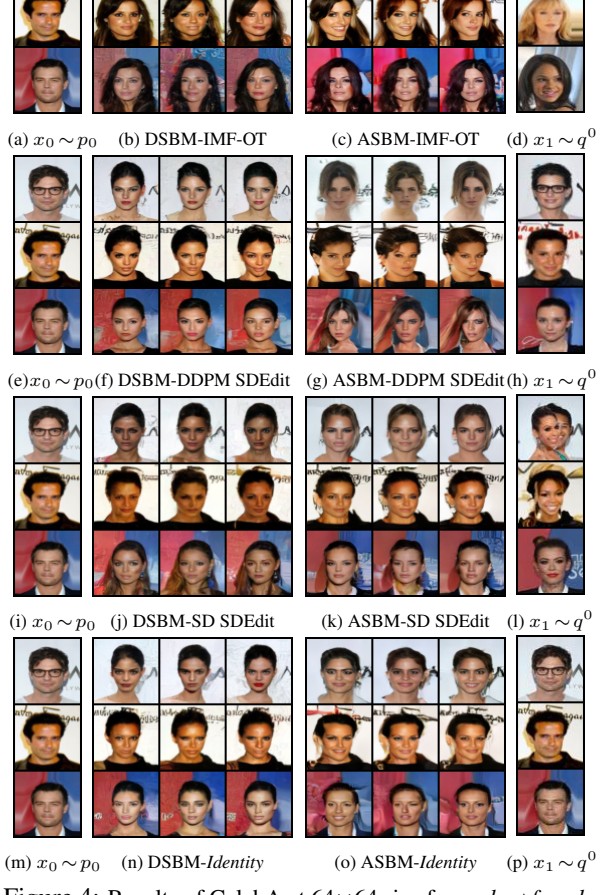

(a) $x_0 \sim p_0$ (b) DSBM-IMF-OT (c) ASBM-IMF-OT (d) $x_1 \sim q^0$

(e) $x_0 \sim p_0$ (f) DSBM-DDPM SDEdit (g) ASBM-DDPM SDEdit (h) $x_1 \sim q^0$

(i) $x_0 \sim p_0$ (j) DSBM-SD SDEdit (k) ASBM-SD SDEdit (l) $x_1 \sim q^0$

(m) $x_0 \sim p_0$ (n) DSBM-*Identity* (o) ASBM-*Identity* (p) $x_1 \sim q^0$

Figure 4: Results of CelebA at $64 \times 64$ size for *male→female* translation learned with ASBM and DSBM using various starting processes for $\epsilon = 1$. Samples $x_0 \sim p_0$ are samples from the source marginal. Samples $x_1 \sim q^0$ are samples from the initialization coupling $q^0(x_1|x_0)$ for a given $x_0$ from $p_0$.

Despite this, their outputs differ due to the influence of initialization on optimization trajectories. For DSBM, our couplings (SD SDEdit, DDPM SDEdit, *Identity*) maintain generation quality while greatly improving similarity. For ASBM, they boost similarity but slightly reduce quality. Results with Identity couplings support our hypothesis (§4.2), whereas experiments with SDEdit offer only partial validation and yield moderate FID.

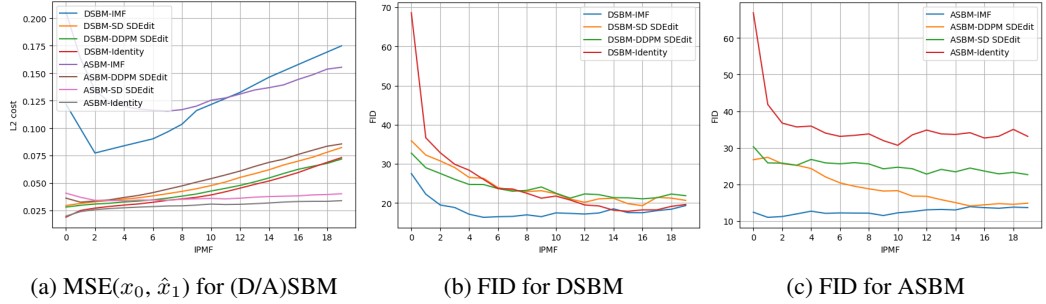

(a) MSE($x_0, \hat{x}_1$) for (D/A)SBM     (b) FID for DSBM     (c) FID for ASBM

Figure 5: Test metrics in CelebA *male→female* ($64 \times 64$) as a function of IPMF iteration for various starting couplings.

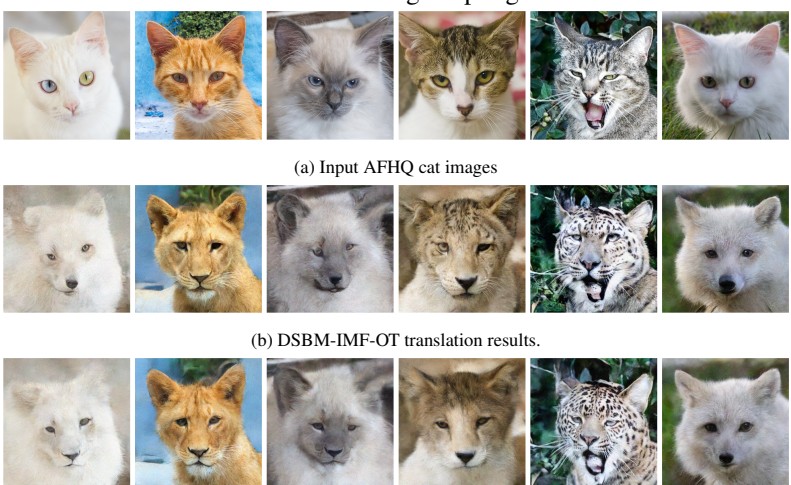

(a) Input AFHQ cat images

(b) DSBM-IMF-OT translation results.

(c) DSBM-*Identity* translation results.

Figure 6: Results of AFHQ at 512×512 size for *cat→wild* translation learned with DSBM using various starting processes for $\epsilon = 1$.

Furthermore, in Figure 5 we present a quantitative study of IPMF convergence, reporting FID and the Mean Squared Error (MSE) between the inputs and the translated outputs as functions of the IPMF iteration. Both metrics are computed on the CelebA *male→female* ($64 \times 64$) test set. We observe a consistent pattern: the higher the similarity or generation quality of the coupling, the better the model performs on the corresponding metric. For additional quantitative results on CelebA $64 \times 64$, we refer the reader to Appendix E.4.

**AFHQ.** For AFHQ (Choi et al., 2020), we consider classes cat and wild with $512 \times 512$ resolution images. Each class contains approximately 5000 samples. We run DSBM with IMF-OT and Identity couplings and present the results in Figure 6 and Table E.10. We observe similar quality-similarity tradeoff as Celeba setup. We provide technical details for this setup in Appendix E.10.

| Coupling | FID↓ | MSE↓ | CMMD↓ |
|---|---|---|---|
| DSBM-IMF-OT | **53.42** | 0.085 | **0.591** |
| DSBM-*Identity* | 65.19 | **0.054** | 0.731 |

Table 3: Results of AFHQ at 512×512 size for *cat→wild* translation learned with DSBM using various starting processes for $\epsilon = 1$.

For a broader discussion of the potential implications and limitations of this work, see Appendix A.

## 5   BROADER IMPACT

This paper presents work whose goal is to advance the field of Artificial Intelligence, Machine Learning and Generative Modeling. There are many potential societal consequences of our work, none which we feel must be specifically highlighted here.

**Acknowledgements.** The work was supported by the grant for research centers in the field of AI provided by the Ministry of Economic Development of the Russian Federation in accordance with the agreement 000000C313925P4F0002 and the agreement №139-10-2025-033. We thank Kirill Sokolov for his careful feedback and helpful discussions, which helped refine the proofs.

## 6 LLM USAGE

Large Language Models (LLMs) were used only to assist with rephrasing sentences and improving the clarity of the text. All scientific content, results, and interpretations in this paper were developed solely by the authors.

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

APPENDIX CONTENTS

## A DISCUSSION

**Potential impact.** The IPMF procedure demonstrates a potential to overcome the error accumulation problem observed in distillation methods—such as rectified flows (Liu et al., 2022; 2023b)—which are used to accelerate foundational image generation models like StableDiffusion 3 in (Esser et al., 2024). These distillation methods are the limit of one-directional IMF procedure with $\epsilon \to 0$. The one-directional version accumulates errors, which may lead to the divergence (De Bortoli et al., 2024, Appx. I). The use of the bidirectional version (with $\epsilon > 0$) should correct the marginals and make diffusion trajectories straighter to accelerate the inference of diffusion models. We believe

| Method | View of bidirectional procedure | Convergence guarantees | Starting coupling |
|---|---|---|---|
| DSBM (Shi et al., 2023) | A heuristic approach for mitigating error accumulation | Only for one-directional continuous-time procedure with IMF and IPF starting couplings | Only IMF and IPF |
| ASBM (Gushchin et al., 2024b) | A heuristic approach for mitigating error accumulation | Only for one-directional discrete-time procedure with IMF starting couplings | Only IMF |
| IPMF (our work) | A theoretically grounded approach for mitigating error accumulation and managing the trade-off between input–output similarity and generative quality | For Gaussian marginals in discrete and continuous time (Theorem 3.2), and convergence for bounded-support distributions in discrete and continuous time (Theorem 3.3). | Arbitrary |

Table 4: Positioning of our IPMF framework relative to prior bidirectional SB heuristics.

that considering such distillation techniques from the IPMF perspective may help to overcome the current limitations of these techniques.

Another potential impact of our contribution is the advancement of multi-marginal SB methods. This direction has been explored only rarely in the literature (Chen et al., 2019; 2023a; Shen et al., 2025; Howard et al., 2025; Lavenant et al., 2021; Theodoropoulos et al., 2025), mainly because the multi-marginal case is inherently difficult: it requires solving multiple two-marginal (classical) SB instances. A notable examples are (Howard et al., 2025; Theodoropoulos et al., 2025), which extends the IMF procedure to the multi-marginal setting. Within this context, our framework provides a way to select a suitable starting coupling for initialization, thereby offering a potential route to reducing the training burden. In this sense, our contribution may encourage more systematic and deeper analysis of multi-marginal SB.

**Limitations**. While we show the proof of exponential convergence of the IPMF procedure in the Gaussian case in various settings, and present a wide set of experiments supporting this procedure, the proof of its convergence in the general case still remains a promising avenue for future work.

# B    MOTIVATION FOR SB OVER FOUNDATIONAL MODELS

At first glance, one might consider foundational models as potential baselines, since translation via large text-to-image models trained on extensive image corpora may work adequately on the benchmark datasets we consider (CelebA, MNIST). However, they do not constitute a relevant baseline for the unpaired translation task, because their training data may lack the domain-specific examples required. In contrast, methods for solving the unpaired translation (including the SB) are designed to address domain-specific tasks across various scientific fields (Schneider et al., 2022; Singh et al., 2024; Shi et al., 2023). Moreover, these methods successfully address non-image-related downstream tasks such as single-cell data analysis (Tong et al., 2024, Section 6), where large text-to-image models are just irrelevant.

# C    CONTINUOUS-TIME SCHRÖDINGER BRIDGE SETUP

For considering the continuous version of Schrödinger Bridge we denote by $\mathcal{P}(C([0,1]), \mathbb{R}^D)$ the set of continuous stochastic processes with time $t \in [0,1]$, i.e., the set of distributions on continuous trajectories $f : [0,1] \to \mathbb{R}^D$. We use $dW_t$ to denote the differential of the standard Wiener process. We denote by $p^T \in \mathcal{P}(\mathbb{R}^{D \times (N+2)})$ the discrete process which is the finite-dimensional projection of $T$ to time moments $0 = t_0 < t_1 < \cdots < t_N < t_{N+1} = 1$.

## C.1    SCHRÖDINGER BRIDGE (SB) PROBLEM IN CONTINUOUS-TIME

This section covers the continuous-time formulation of SB as its IPF and IMF procedures. First, we introduce several new notations to better align the continuous version with the discrete-time version considered in the main text. Consider the Markovian process $T$ defined by the corresponding forward or backward (time-reversed) SDEs:

$$T : dx_t = v^+(x_t, t)dt + \sqrt{\epsilon}dW_t^+, \quad x_0 \sim p_0(x_0),$$

$$T : dx_t = v^-(x_t, t)dt + \sqrt{\epsilon}dW_t^-, \quad x_1 \sim p_1(x_1),$$

where we additionally denote by $W_t^+$ and $W_t^-$ the Wiener process in forward or backward time. We say $T_{|x_0}$ and $T_{|x_1}$ denotes the conditional process of $T$ fixing the marginals using delta functions, i.e., setting $p_0(x_0) = \delta_{x_0}(x)$ and $p_1(x_1) = \delta_{x_1}(x)$:

$$T_{|x_0} : dx_t = v^+(x_t, t)dt + \sqrt{\epsilon}dW_t^+, \quad x_0 \sim \delta_{x_0}(x),$$
$$T_{|x_1} : dx_t = v^-(x_t, t)dt + \sqrt{\epsilon}dW_t^-, \quad x_1 \sim \delta_{x_1}(x).$$

Moreover, we use $p(x_0)T_{|x_0}$ to denote the stochastic process which starts by sampling $x_0 \sim p(x_0)$ and then moving this $x_0$ according the SDE given by $T_{|x_0}$, i.e., $p(x_0)T_{|x_0}$ is short for the process $\int T_{|x_0}p(x_0)dx_0$. Finally, we use the shortened notation of the process $T_{|0,1}(x_0, x_1)$ conditioned on its values at times 0 and 1, saying $p^T(x_0, x_1)T_{|0,1}(x_0, x_1) = \int T_{|0,1}(x_0, x_1)p^T(x_0, x_1)dx_0dx_1$. This links the following equations with the discrete-time formulation.

**Schrödinger Bridge problem.** Considering the continuous case, the Schrödinger Bridge problem is stated using continuous stochastic processes instead of one with predefined timesteps. Thus, the Schrödinger Bridge problem finds the most likely in the sense of Kullback-Leibler divergence stochastic process $T$ with respect to prior Wiener process $W^\epsilon$, i.e.:

$$\min_{T \in \mathcal{F}(p_0, p_1)} \text{KL}(T||W^\epsilon), \tag{13}$$

where $\mathcal{F}(p_0, p_1) \subset \mathcal{P}(C([0, 1]), \mathbb{R}^D)$ is the set of all stochastic processes pinned by marginal distributions $p_0$ and $p_1$ at times 0 and 1, respectively. The minimization problem (13) has a unique solution $T^*$ which can be represented as forward or backward diffusion (Léonard, 2013):

$$T^* : dx_t = v^{*+}(x_t, t)dt + \sqrt{\epsilon}dW_t^+, \quad x_0 \sim p_0(x_0),$$
$$T^* : dx_t = v^{*-}(x_t, t)dt + \sqrt{\epsilon}dW_t^-, \quad x_1 \sim p_1(x_1),$$

where $v^{*+}$ and $v^{*-}$ are the corresponding drift functions.

**Static Schrödinger Bridge problem.** As in discrete-time, Kullback-Leibler divergence in (13) could be decomposed as follows:

$$\text{KL}(T||W^\epsilon) = \text{KL}(p^T(x_0, x_1)||p^{W^\epsilon}(x_0, x_1)) + \int \text{KL}(T_{|x_0, x_1}||W_{|x_0, x_1}^\epsilon)dp^T(x_0, x_1). \tag{14}$$

It has been proved (Léonard, 2013) that for the solution $T^*$ it's conditional process is given by $T_{|x_0, x_1}^* = W_{|x_0, x_1}^\epsilon$. Thus, we can set $T_{|x_0, x_1} = W_{|x_0, x_1}^\epsilon$ zeroing the second term in (14) and minimize over processes with $T_{|x_0, x_1} = W_{|x_0, x_1}^\epsilon$. This leads to the equivalent Static formulation of the Schrödinger Bridge problem:

$$\min_{q \in \Pi(p_0, p_1)} \text{KL}(q(x_0, x_1)||p^{W^\epsilon}(x_0, x_1)), \tag{15}$$

where $\Pi(p_0, p_1)$ is the set of all joint distributions with marginals $p_0$ and $p_1$. Whether time is discrete or continuous, the decomposition of SB leads to the same static formulation, which is closely related to Entropic OT as shown in (3).

## C.2 ITERATIVE PROPORTIONAL FITTING (IPF) FOR CONTINUOUS-TIME

Following the main text, we describe the IPF procedure for continuous-time setup using stochastic processes. Likewise, IPF starts with setting $T^0 = p_0(x_0)W_{|x_0}^\epsilon$ and then it alternates betwethe followinging projections:

$$T^{2k+1} = \text{proj}_1\left(p^{T^{2k}}(x_1)T_{|x_1}^{2k}\right) \stackrel{\text{def}}{=} p_1(x_1)T_{|x_1}^{2k}, \tag{16}$$

$$T^{2k+2} = \text{proj}_0\left(p^{T^{2k+1}}(x_0)T_{|x_0}^{2k+1}\right) \stackrel{\text{def}}{=} p_0(x_0)T_{|x_0}^{2k+1}. \tag{17}$$

As in the discrete-time case, these projections replace marginal distributions $p^T(x_1)$ and $p^T(x_0)$ in the processes $p^T(x_1)T_{|x_1}$ and $p^T(x_0)T_{|x_0}$ by $p_1(x_1)$ and $p_0(x_0)$ respectively. Similarly to discrete-time formulation, the sequence of $T^k$ converges to the solution of the Schrödinger Bridge problem

$T^*$ implicitly decreasing the reverse Kullback-Leibler divergence $\text{KL}(T^k||T^*)$ between the current process $T^k$ and the solution to the SB problem $T*$. Additionally, it should be mentioned that existing methods perform projections via numerical approximation of forward and time-reversed conditional processes, $T_{|x_0}$ and $T_{|x_1}$, by learning their drifts via one of the methods: score matching (De Bortoli et al., 2021) or maximum likelihood estimation (Vargas et al., 2021).

## C.3 ITERATIVE MARKOVIAN FITTING (IMF) FOR CONTINUOUS-TIME

IMF introduces new projections that alternate between reciprocal and Markovian processes starting from any process $T^0$ pinned by $p_0$ and $p_1$ at times 0 and 1, i.e., in $T^0 \in \mathcal{F}(p_0, p_1)$:

$$T^{2k+1} = \text{proj}_{\mathcal{R}}\left(T^{2k}\right) \overset{\text{def}}{=} p^{T^{2k}}(x_0, x_1)W^{\epsilon}_{|x_0,x_1}, \tag{18}$$

$$T^{2k+2} = \text{proj}_{\mathcal{M}}\left(T^{2k+1}\right) \overset{\text{def}}{=} \underbrace{p^{T^{2k+1}}(x_0)T^{2k+1}_{M|x_0}}_{\text{forward representation}} = \underbrace{p^{T^{2k+1}}(x_1)T^{2k+1}_{M|x_1}}_{\text{backward representation}}. \tag{19}$$

where we denote by $T_M$ the Markovian projections of the processes $T$, which can be represented as the forward or backward time diffusion as follows (Gushchin et al., 2024b, Section 2.1):

$$T_M : dx_t^+ = v_M^+(x_t^+, t)dt + \sqrt{\epsilon}dW_t^+, \quad x_0 \sim p^T(x_0), \quad v_M^+(x_t^+, t) = \int \frac{x_1 - x_t^+}{1-t}p^T(x_1|x_t)dx_1,$$

$$T_M : dx_t^- = v_M^-(x_t^-, t)dt + \sqrt{\epsilon}dW_t^-, \quad x_1 \sim p^T(x_1), \quad v_M^-(x_t^-, t) = \int \frac{x_0 - x_t^-}{1-t}p^T(x_0|x_t)dx_0.$$

This procedure converges to a unique solution, which is the Schrödinger bridge $T^*$ (Léonard, 2013). While reciprocal projection can be easily done by combining the joint distribution $p^T(x_0, x_1)$ of the process $T$ and Brownian bridge $W^{\epsilon}_{|x_0,x_1}$, the Markovian projection is much more challenging and must be fitted via Bridge matching (Shi et al., 2023; Liu et al.; Peluchetti, 2023b).

Since the result of the Markovian projection (19) can be represented both by forward and backward representation, in practice, neural networks $v_\theta^+$ (**forward parametrization**) or $v_\phi^-$ (**backward parametrization**) are used to learn the corresponding drifts of the Markovian projections. In turn, starting distributions are set to be $p_0(x_0)$ for forward parametrization and $p_1(x_1)$ for the backward parametrization. So, this **bidirectional** procedure can be described as follows:

$$T^{4k+1} = \underbrace{p^{T^{4k}}(x_0, x_1)W^{\epsilon}_{|x_0,x_1}}_{\text{proj}_{\mathcal{R}}(T^{4k})}, \quad T^{4k+2} = \underbrace{p_1(x_1)T^{4k+1}_{M|x_1}}_{\text{backward parametrization}}, \tag{20}$$

$$T^{4k+3} = \underbrace{p^{T^{4k+2}}(x_0, x_1)W^{\epsilon}_{|x_0,x_1}}_{\text{proj}_{\mathcal{R}}(T^{4k+2})}, \quad T^{4k+4} = \underbrace{p_0(x_0)T^{4k+3}_{M|x_0}}_{\text{forward parametrization}}. \tag{21}$$

## C.4 ITERATIVE PROPORTIONAL MARKOVIAN FITTING (IPMF) FOR CONTINUOUS-TIME

Here, we analyze the continuous version of the heuristical bidirectional IMF. First, we recall, that the IPF projections $\text{proj}_0(T)$ and $\text{proj}_1(T)$ given by (16) and (17) of the Markovian process $T$ is just change the starting distribution from $p^T(x_0)$ to $p_0(x_0)$ and $p^T(x_1)$ to $p_1(x_1)$.

Now we note that the process $T^{4k+2}$ in (20) is obtained by using a combination of Markovian projection $\text{proj}_{\mathcal{M}}$ given by (19) in backward parametrization and IPF projection $\text{proj}_1$ given by (17):

$$T^{4k+2} = p_1(x_1)T^{4k+1}_{M|x_1} = \underbrace{\text{proj}_1\left(p^{T^{4k+1}}(x_1)T^{4k+1}_{M|x_1}\right)}_{\text{proj}_1(\text{proj}_{\mathcal{M}}(T^{4k+1}))}.$$

In turn, the process $T^{4k+4}$ in (21) is obtained by using a combination of Markovian projection $\text{proj}_{\mathcal{M}}$ given by (19) in forward parametrization and IPF projection $\text{proj}_0$ given by (16):

$$T^{4k+3} = p_0(x_0)T^{4k+3}_{M|x_0} = \underbrace{\text{proj}_0\left(p^{T^{4k+3}}(x_0)T^{4k+3}_{M|x_0}\right)}_{\text{proj}_0(\text{proj}_{\mathcal{M}}(T^{4k+3}))}.$$

Combining these facts we can rewrite bidirectional IMF in the following manner:

**Iterative Proportional Markovian Fitting (Conitnious time setting)**

$$T^{4k+1} = \underbrace{p^{T^{4k}}(x_0, x_1)W^{\epsilon}_{|x_0,x_1}}_{\text{proj}_{\mathcal{R}}(T^{4k})}, \quad T^{4k+2} = \underbrace{p_1(x_1)T^{4k+1}_{M|x_1}}_{\text{proj}_1(\text{proj}_{\mathcal{M}}(T^{4k+1}))} \tag{22}$$

$$T^{4k+3} = \underbrace{p^{T^{4k+2}}(x_0, x_1)W^{\epsilon}_{|x_0,x_1}}_{\text{proj}_{\mathcal{R}}(T^{4k+2})}, \quad T^{4k+4} = \underbrace{p_0(x_0)T^{4k+3}_{M|x_0}}_{\text{proj}_0(\text{proj}_{\mathcal{M}}(T^{4k+3}))} \,. \tag{23}$$

Thus, we obtain the analog of the discrete-time IPMF procedure, which concludes our description of the continuous setups.

## D  THEORETICAL ANALYSIS FOR GAUSSIANS

Here, we study behavior of IPMF with volatility $\epsilon$ between $D$-dimensional Gaussians $p_0 = \mathcal{N}(\mu_0, \Sigma_0)$ and $p_1 = \mathcal{N}(\mu_1, \Sigma_1)$. For various settings, we prove that the parameters of $q^{4k}$ with each step geometrically converge to desired values $\mu_0, \mu_1, \Sigma_0, \Sigma_1, \epsilon$. The steps are as follows:

**1)** In Appendix D.1, we reveal the connection between $2D$-dimensional Gaussian distribution and solution of entropic OT problem with specific transport cost, i.e., we prove our Theorem 3.1.

**2)** In Appendix D.2, we study the effect of IPF steps on the current process. We show that during these steps, the marginals become close to $p_0$ and $p_1$, and the optimality matrix does not change. We also prove that the spectral norms of the marginal matrices are bounded during the whole IPMF procedure.

**3)** In Appendix D.3, we study the effect of IMF step on the current process when $D > 1$. We show that after a discrete IMF step, the distance between current optimality matrix and desired one can be bounded by the scaled previous distance.

**4)** In Appendix D.5, we study the effect of IMF step in a particular case $D = 1$. We show that after IMF step (continuous or discrete with $N = 1$), marginals remain the same, and the intermediate distribution becomes closer to the intermediate distribution of the static $\epsilon$-EOT solution between the marginals.

**5)** Finally, in Appendices D.4 and D.6, we prove our main Theorem 3.2 for the case $D > 1$ and $D = 1$, respectively.

### D.1  GAUSSIAN PLANS AS ENTROPIC OPTIMAL TRANSPORT PLANS

*Proof of Theorem 3.1.* The conditional distribution of $q(x_0|x_1)$ has a closed form:

$$\begin{aligned} q(x_0|x_1) &= \mathcal{N}\left(x_0|\eta + P(S)^{-1}(x_1 - \nu), Q - P(S)^{-1}P^{\top}\right) \\ &= Z_{x_0}Z_{x_1}\exp\left(x_0^{\top}(Q - P(S)^{-1}P^{\top})^{-1}P(S)^{-1}x_1\right) \\ &= Z_{x_0}Z_{x_1}\exp\left(x_1^{\top}Ax_0\right), \end{aligned} \tag{24}$$

where factors $Z_{x_0}$ and $Z_{x_1}$ depend only on $x_0$ and $x_1$, respectively, and the matrix $A$ is

$$A = (S)^{-1}P^{\top}(Q - P(S)^{-1}P^{\top})^{-1}. \tag{25}$$

Theorem 3.2 from Gushchin et al. (2023b) states that if the conditional distribution $q(x_1|x_0)$ can be expressed as:

$$q(x_0|x_1) \propto \exp\left(-c(x_0, x_1) + f_c(x_0)\right), \tag{26}$$

where $c(x_0, x_1)$ is a lower bounded cost function, and the function $f_c(x_0)$ depends only on $x_0$, then $q$ solves 1-entropic OT with the cost function $c(x_0, x_1)$. Equating the terms in (24) and (26) which depend on both $x_0$ and $x_1$, we derive the formula for the cost function is $c(x_0, x_1) = -x_0^{\top}Ax_1$. We denote it as $c_A(x_0, x_1) := -x_0^{\top}Ax_1$ to show the dependency on the optimality matrix $A$.

We only need to note that we can add any functions $f(x_0)$ and $g(x_1)$ depending only on $x_0$ or $x_1$, respectively, to the cost function $c_A(x_0, x_1) = -x_1^{\top}Ax_0$, and the OT solution will not change. This

is because the integrals of such functions over any transport plan will be constants, as they will depend only on the marginals (which are given) but not on the plan itself. Thus, for any $A \in \mathbb{R}^{D \times D}$, we can rearrange the cost term $c_A(x_0, x_1)$ so that it becomes lower-bounded:

$$\tilde{c}_A(x_0, x_1) = \|Ax_0\|^2/2 - x_1^\top A x_0 + \|x_1\|^2/2 = \|Ax_0 - x_1\|^2/2 \geq 0,$$

where $\tilde{c}_A(x_0, x_1)$ is a lower bounded function. □

## D.2 IPF STEP ANALYSIS

We run IPMF with the desired volatility parameter $\epsilon_*$ between the desired distributions $p_0 = \mathcal{N}(\mu_0, \Sigma_0)$ and $p_1 = \mathcal{N}(\mu_1, \Sigma_1)$, starting with the process $\mathcal{N}\left(\begin{pmatrix} \mu_0 \\ \nu \end{pmatrix}, \begin{pmatrix} \Sigma_0 & P \\ P & S \end{pmatrix}\right)$ which has the correlation matrix $P$.

One IPMF step can be decomposed into the following consecutive steps:

1. IMF step: projections $\text{proj}_\mathcal{M}(\text{proj}_\mathcal{R})$, refining the current optimality matrix,
2. IPF step: projection $\text{proj}_1$, changing final prior at time $t = 1$ to $p_1 = \mathcal{N}(\mu_1, \Sigma_1)$,
3. IMF step: projections $\text{proj}_\mathcal{M}(\text{proj}_\mathcal{R})$, refining the current optimality matrix,
4. IPF step: projection $\text{proj}_0$, changing starting prior at time $t = 0$ to $p_0 = \mathcal{N}(\mu_0, \Sigma_0)$.

We use the following notations for the covariance matrices changes during IPMF step:

$$\begin{pmatrix} \Sigma_0 & P \\ P^\top & S \end{pmatrix} \overset{IMF}{\Longrightarrow} \begin{pmatrix} \Sigma_0 & \tilde{P} \\ \tilde{P}^\top & S \end{pmatrix} \overset{IPF}{\Longrightarrow} \begin{pmatrix} Q & P' \\ (P')^\top & \Sigma_1 \end{pmatrix}$$

$$\overset{IMF}{\Longrightarrow} \begin{pmatrix} Q & \hat{P} \\ \hat{P}^\top & \Sigma_1 \end{pmatrix} \overset{IPF}{\Longrightarrow} \begin{pmatrix} \Sigma_0 & P'' \\ (P'')^\top & S' \end{pmatrix},$$

and for the means, the changes are:

$$\begin{pmatrix} \mu_0 \\ \nu \end{pmatrix} \overset{IMF}{\Longrightarrow} \begin{pmatrix} \mu_0 \\ \nu \end{pmatrix} \overset{IPF}{\Longrightarrow} \begin{pmatrix} \eta \\ \mu_1 \end{pmatrix} \overset{IMF}{\Longrightarrow} \begin{pmatrix} \eta \\ \mu_1 \end{pmatrix} \overset{IPF}{\Longrightarrow} \begin{pmatrix} \mu_0 \\ \nu' \end{pmatrix}.$$

**Lemma D.1** (Improvement after IPF steps). *Consider an initial 2D-dimensional Gaussian joint distribution* $\mathcal{N}\left(\begin{pmatrix} \mu_0 \\ \nu \end{pmatrix}, \begin{pmatrix} \Sigma_0 & P \\ P^\top & S \end{pmatrix}\right) \in \mathcal{P}_{2,ac}(\mathbb{R}^D \times \mathbb{R}^D)$. *We run IPMF step between distributions* $\mathcal{N}(\mu_0, \Sigma_0)$ *and* $\mathcal{N}(\mu_1, \Sigma_1)$ *and obtain new joint distribution* $\mathcal{N}\left(\begin{pmatrix} \mu_0 \\ \mu'' \end{pmatrix}, \begin{pmatrix} \Sigma_0 & P'' \\ (P'')^\top & S' \end{pmatrix}\right)$. *Then, the distance between ground truth* $\mu_1, \Sigma_1$ *and the new joint distribution parameters decreases as:*

$$\|(S')^{-\frac{1}{2}}\Sigma_1 (S')^{-\frac{1}{2}} - I_D\|_2 \leq \|\tilde{P}_n\|_2^2 \cdot \|P''_n\|_2^2 \cdot \|S^{-\frac{1}{2}}\Sigma_1 S^{-\frac{1}{2}} - I_D\|_2, \quad (27)$$

$$\|\Sigma_1^{-\frac{1}{2}}(\nu' - \mu_1)\|_2 \leq \|\hat{P}_n^\top\|_2 \cdot \|P'_n\|_2 \cdot \|\Sigma_1^{-\frac{1}{2}}(\nu - \mu_1)\|_2, \quad (28)$$

*where* $\tilde{P}_n := \Sigma_0^{-1/2}\tilde{P}S^{-1/2}, P'_n := (Q)^{-\frac{1}{2}}P'\Sigma_1^{-\frac{1}{2}}, \hat{P}_n := (Q)^{-1/2}\hat{P}\Sigma_1^{-1/2}$ *and* $P''_n := \Sigma_0^{-1/2}P''(S')^{-1/2}$ *are normalized matrices whose spectral norms are not greater than* 1.

*Proof.* During IPF steps, we keep the conditional distribution and change the marginal. For the first IPF, we keep the inner part $x_0|x_1$ for all $x_1 \in \mathbb{R}^D$:

$$\mathcal{N}\left(x_0|\mu_0 + \tilde{P}S^{-1}(x_1 - \nu), \Sigma_0 - \tilde{P}S^{-1}\tilde{P}^\top\right) = \mathcal{N}\left(x_0|\eta + P'\Sigma_1^{-1}(x_1 - \mu_1), Q - P'\Sigma_1^{-1}(P')^\top\right).$$

This is equivalent to the system of equations:

$$\Sigma_0 - \tilde{P}S^{-1}\tilde{P}^\top = Q - P'\Sigma_1^{-1}(P')^\top, \quad (29)$$

$$P'\Sigma_1^{-1} = \tilde{P}S^{-1}, \quad (30)$$

$$\mu_0 - \tilde{P}S^{-1}\nu = \eta - P'\Sigma_1^{-1}\mu_1. \quad (31)$$

Similarly, after the second IPF step, we have equations:

$$\Sigma_1 - \hat{P}^\top (Q)^{-1} \hat{P} = S' - (P'')^\top \Sigma_0^{-1} P'', \tag{32}$$

$$(P'')^\top \Sigma_0^{-1} = \hat{P}^\top (Q)^{-1}, \tag{33}$$

$$\mu_1 - \hat{P}^\top (Q)^{-1} \eta = \nu' - (P'')^\top \Sigma_0^{-1} \mu_0. \tag{34}$$

**Covariance matrices.** Combining equations (30), (29) and (33), (32) together, we obtain:

$$\Sigma_0 - Q = \tilde{P} S^{-1} (S - \Sigma_1) S^{-1} \tilde{P}^\top, \qquad //(29),(30) \tag{35}$$

$$I_D - \Sigma_0 (Q)^{-1} = \tilde{P} S^{-1} (\Sigma_1 - S) S^{-1} \tilde{P}^\top (Q)^{-1}, \qquad //(35) \cdot (Q)^{-1} \tag{36}$$

$$\Sigma_1 - S' = \hat{P}^\top (Q)^{-1} (I_D - \Sigma_0 (Q)^{-1}) \hat{P}, \qquad //(32),(33) \tag{37}$$

$$\Sigma_1 - S' = \hat{P}^\top (Q)^{-1} \tilde{P} S^{-1} (\Sigma_1 - S) S^{-1} \tilde{P}^\top (Q)^{-1} \hat{P}, \qquad //(36) \text{ insert to } (37)$$

$$\Sigma_1 - S' = (P'')^\top \Sigma_0^{-1} \tilde{P} S^{-1} (\Sigma_1 - S) S^{-1} \tilde{P}^\top \Sigma_0^{-1} P'', \qquad //\text{change using } (33)$$

$$(S')^{-\frac{1}{2}} \Sigma_1 (S')^{-\frac{1}{2}} - I_D = (S')^{-\frac{1}{2}} (P'')^\top \Sigma_0^{-\frac{1}{2}} \cdot \Sigma_0^{-\frac{1}{2}} \tilde{P} S^{-\frac{1}{2}}$$
$$\cdot (S^{-\frac{1}{2}} \Sigma_1 S^{-\frac{1}{2}} - I_D) \cdot S^{-\frac{1}{2}} \tilde{P}^\top \Sigma_0^{-\frac{1}{2}} \cdot \Sigma_0^{-\frac{1}{2}} P'' (S')^{-\frac{1}{2}}.$$

The matrices (29) and (32) must be SPD to be covariance matrices:

$$\Sigma_0 - \tilde{P} S^{-1} \tilde{P}^\top \succeq 0 \quad \Longrightarrow \quad I_D \succeq \Sigma_0^{-1/2} \tilde{P} S^{-1/2} \cdot S^{-1/2} \tilde{P}^\top \Sigma_0^{-1/2},$$

$$S' - (P'')^\top \Sigma_0^{-1} P'' \succeq 0 \quad \Longrightarrow \quad I_D \succeq \Sigma_0^{-1/2} P'' (S')^{-1/2} \cdot (S')^{-1/2} (P'')^\top \Sigma_0^{-1/2}.$$

In other words, denoting matrices $\tilde{P}_n := \Sigma_0^{-1/2} \tilde{P} S^{-1/2}$ and $P''_n := \Sigma_0^{-1/2} P'' (S')^{-1/2}$, we can bound their spectral norms as $\|\tilde{P}_n\|_2 \leq 1$ and $\|P''_n\|_2 \leq 1$. We write down the final transaction for covariance matrices:

$$(S')^{-\frac{1}{2}} \Sigma_1 (S')^{-\frac{1}{2}} - I_D = (P''_n)^\top \cdot \tilde{P}_n \cdot (S^{-\frac{1}{2}} \Sigma_1 S^{-\frac{1}{2}} - I_D) \cdot \tilde{P}_n^\top \cdot P''_n. \tag{38}$$

Hence, the spectral norm of the difference between ground truth $\Sigma_1$ and current $S'$ drops exponentially as:

$$\|(S')^{-\frac{1}{2}} \Sigma_1 (S')^{-\frac{1}{2}} - I_D\|_2 \leq \|\tilde{P}_n\|_2^2 \cdot \|P''_n\|_2^2 \cdot \|S^{-\frac{1}{2}} \Sigma_1 S^{-\frac{1}{2}} - I_D\|_2.$$

**Means.** Combining equations (31), (30) and (34), (33) together, we obtain:

$$\mu_0 - \eta = \tilde{P} S^{-1} \nu - P' \Sigma_1^{-1} \mu_1 = P' \Sigma_1^{-1} (\nu - \mu_1), \qquad //(31),(30) \tag{39}$$

$$\nu' - \mu_1 = (P'')^\top \Sigma_0^{-1} \mu_0 - \hat{P}^\top (Q)^{-1} \eta = \hat{P}^\top (Q)^{-1} (\mu_0 - \eta), \qquad //(34),(33) \tag{40}$$

$$\nu' - \mu_1 = \hat{P}^\top (Q)^{-1} P' \Sigma_1^{-1} (\nu - \mu_1), \qquad //\text{insert } (39) \text{ to } (40)$$

$$\Sigma_1^{-\frac{1}{2}} (\nu' - \mu_1) = \Sigma_1^{-\frac{1}{2}} \hat{P}^\top (Q)^{-\frac{1}{2}} \cdot (Q)^{-\frac{1}{2}} P' \Sigma_1^{-\frac{1}{2}} \cdot \Sigma_1^{-\frac{1}{2}} (\nu - \mu_1).$$

The matrices (29) and (32) must be SPD to be covariance matrices:

$$Q \succeq P' \Sigma_1^{-1} (P')^\top \quad \Longrightarrow \quad I_D \succeq (Q)^{-\frac{1}{2}} P' \Sigma_1^{-\frac{1}{2}} \cdot \Sigma_1^{-\frac{1}{2}} (P')^\top (Q)^{-\frac{1}{2}},$$

$$\Sigma_1 \succeq \hat{P}^\top (Q)^{-1} \hat{P} \quad \Longrightarrow \quad I_D \succeq \Sigma_1^{-1/2} \hat{P}^\top (Q)^{-1/2} \cdot (Q)^{-1/2} \hat{P} \Sigma_1^{-1/2}.$$

Denoting matrices $P'_n := (Q)^{-\frac{1}{2}} P' \Sigma_1^{-\frac{1}{2}}$ and $\hat{P}_n := (Q)^{-1/2} \hat{P} \Sigma_1^{-1/2}$, we can bound their spectral norms as $\|P'_n\|_2 \leq 1$ and $\|\hat{P}_n\|_2 \leq 1$. We use this to estimate the $\ell_2$-norm of the difference between the ground truth $\mu_1$ and the current mean:

$$\Sigma_1^{-\frac{1}{2}} (\nu' - \mu_1) = \hat{P}_n^\top \cdot P'_n \cdot \Sigma_1^{-\frac{1}{2}} (\nu - \mu_1), \tag{41}$$

$$\|\Sigma_1^{-\frac{1}{2}} (\nu' - \mu_1)\|_2 \leq \|\hat{P}_n^\top\|_2 \cdot \|P'_n\|_2 \cdot \|\Sigma_1^{-\frac{1}{2}} (\nu - \mu_1)\|_2.$$

$$\square$$

**Lemma D.2** (Marginals norm bound during IPMF procedure). *Consider an initial $2D$-dimensional Gaussian joint distribution $\mathcal{N}\left(\begin{pmatrix} \mu_0 \\ \nu_0 \end{pmatrix}, \begin{pmatrix} \Sigma_0 & P_0 \\ P_0^\top & S_0 \end{pmatrix}\right) \in \mathcal{P}_{2,ac}(\mathbb{R}^D \times \mathbb{R}^D)$. We run $k$ IPMF step between distributions $\mathcal{N}(\mu_0, \Sigma_0)$ and $\mathcal{N}(\mu_1, \Sigma_1)$ and obtain new joint distribution $\mathcal{N}\left(\begin{pmatrix} \mu_0 \\ \nu_k \end{pmatrix}, \begin{pmatrix} \Sigma_0 & P_k \\ P_k^\top & S_k \end{pmatrix}\right)$. Then the norm $\|S_k\|_2$ can be bounded independently of $k$ by:*

$$\|S_k\|_2 \leq \frac{\|\Sigma_1\|_2}{\min\{\lambda_{min}(S_0^{-\frac{1}{2}}\Sigma_1 S_0^{-\frac{1}{2}}), 1\}}, \quad \|S_k^{-1}\|_2 \leq \max\{\lambda_{max}(S_0^{-\frac{1}{2}}\Sigma_1 S_0^{-\frac{1}{2}}), 1\}\|\Sigma_1^{-1}\|_2. \quad (42)$$

*This statement also implies the invertibility of all matrices $S_k$.*

*For matrices $Q_k$, the results are analogous.*

*Proof.* Consider the last IPMF step. We denote symmetric matrices $\Delta_k := S_k^{-\frac{1}{2}}\Sigma_1 S_k^{-\frac{1}{2}} - I_D, \Delta_{k-1} := S_{k-1}^{-\frac{1}{2}}\Sigma_1 S_{k-1}^{-\frac{1}{2}} - I_D$ and $\hat{\lambda}_{min}(\Delta) := \min\{0, \lambda_{min}(\Delta)\}, \hat{\lambda}_{max}(\Delta) := \max\{0, \lambda_{max}(\Delta)\}$. Next, we estimate spectral norm of $S_{k-1}$ as follows:

$$\begin{aligned}
\Delta_k = S_k^{-\frac{1}{2}}\Sigma_1 S_k^{-\frac{1}{2}} - I_D &\succeq \lambda_{min}(\Delta_k)I_D \succeq \hat{\lambda}_{min}(\Delta_k)I_D, \\
S_k^{-\frac{1}{2}}\Sigma_1 S_k^{-\frac{1}{2}} &\succeq (\hat{\lambda}_{min}(\Delta_k) + 1)I_D, \\
\Sigma_1 &\succeq (\hat{\lambda}_{min}(\Delta_k) + 1)S_k
\end{aligned}$$

Note, that by design we have $\Delta_k \succeq -I_D \Rightarrow -1 \leq \hat{\lambda}_{min}(\Delta_k) \leq 0 \Rightarrow 0 \leq (\hat{\lambda}_{min}(\Delta_k) + 1) \leq 1$ and can obtain

$$\begin{aligned}
\Sigma_1 &\succeq (\hat{\lambda}_{min}(\Delta_k) + 1)S_k, \\
S_k &\preceq \frac{1}{\hat{\lambda}_{min}(\Delta_k) + 1}\Sigma_1, \\
\|S_k\|_2 &\leq \frac{\|\Sigma_1\|_2}{\hat{\lambda}_{min}(\Delta_k) + 1}.
\end{aligned} \quad (43)$$

Similarly, we prove that

$$S_k \succeq \frac{1}{\hat{\lambda}_{max}(\Delta_k) + 1}\Sigma_1 \Rightarrow \|S_k^{-1}\|_2 \leq (\hat{\lambda}_{max}(\Delta_k) + 1)\|\Sigma_1^{-1}\|_2.$$

Now, we prove that $\hat{\lambda}_{min}(\Delta_k) \geq \hat{\lambda}_{min}(\Delta_{k-1})$. We denote by $P_n''$ and $\tilde{P}_n$ normalized matrices after the second IPF step and the first IMF step on the last iteration, respectively (see Lemma D.1). For any $x \in \mathbb{R}^D, \|x\|_2 \leq 1$, we calculate the bilinear form:

$$\begin{aligned}
x^\top \Delta_k x &\overset{(38)}{=} x^\top (P_n'')^\top \cdot \tilde{P}_n \cdot (S_{k-1}^{-\frac{1}{2}}\Sigma_1 S_{k-1}^{-\frac{1}{2}} - I_D) \cdot \tilde{P}_n^\top \cdot P_n'' x \\
&= (\tilde{P}_n^\top \cdot P_n'' x)^\top \Delta_{k-1}(\tilde{P}_n^\top \cdot P_n'' x), \\
\hat{\lambda}_{min}(\Delta_k) &= \min\left\{0, \min_{\|x\|_2=1} x^\top \Delta_k x\right\} \\
&\geq \min\left\{0, \min_{\|x\|_2=1} (\tilde{P}_n^\top \cdot P_n'' x)^\top \Delta_{k-1}(\tilde{P}_n^\top \cdot P_n'' x)\right\} \\
&\geq \|\tilde{P}_n^\top \cdot P_n'' x\|_2^2 \cdot \min\{0, \lambda_{min}(\Delta_{k-1})\} \\
&\geq \|\tilde{P}_n\|^2 \|P_n''\|^2 \|x\|_2^2 \cdot \hat{\lambda}_{min}(\Delta_{k-1}) \geq \hat{\lambda}_{min}(\Delta_{k-1}).
\end{aligned}$$

Hence, after each IPMF step $\hat{\lambda}_{min}(\Delta_k)$ increases and can be lower bounded by the initial value $\hat{\lambda}_{min}(\Delta_k) \geq \hat{\lambda}_{min}(\Delta_0)$ using math induction. It implies the invertibility of all matrices $S_k$ and boundness of norms

$$\|S_k\|_2 \overset{(43)}{\leq} \frac{\|\Sigma_1\|_2}{\hat{\lambda}_{min}(\Delta_k) + 1} \leq \frac{\|\Sigma_1\|_2}{\hat{\lambda}_{min}(\Delta_0) + 1} = \frac{\|\Sigma_1\|_2}{\min\{\lambda_{min}(S_0^{-\frac{1}{2}}\Sigma_1 S_0^{-\frac{1}{2}}), 1\}}.$$

Similarly, we prove that

$$
\begin{aligned}
\hat{\lambda}_{max}(\Delta_k) \le \hat{\lambda}_{max}(\Delta_{k-1}) \Rightarrow \|S_k^{-1}\|_2 &\le (\hat{\lambda}_{max}(\Delta_0) + 1)\|\Sigma_1^{-1}\|_2 \\
&\le \max\{\lambda_{max}(S_0^{-\frac{1}{2}}\Sigma_1 S_0^{-\frac{1}{2}}), 1\}\|\Sigma_1^{-1}\|_2.
\end{aligned}
$$

$\square$

**Lemma D.3** (IPF step does not change optimality matrix $A$). *Consider an initial $2D$-dimensional Gaussian joint distribution* $\mathcal{N}\left(\begin{pmatrix} \mu_0 \\ \nu \end{pmatrix}, \begin{pmatrix} \Sigma_0 & \tilde{P} \\ \tilde{P}^\top & S \end{pmatrix}\right) \in \mathcal{P}_{2,ac}(\mathbb{R}^D \times \mathbb{R}^D)$. *We run IPF step between distributions* $\mathcal{N}(\mu_0, \Sigma_0)$ *and* $\mathcal{N}(\mu_1, \Sigma_1)$ *and obtain new joint distribution* $\mathcal{N}\left(\begin{pmatrix} \eta \\ \mu_1 \end{pmatrix}, \begin{pmatrix} Q & P' \\ (P')^\top & \Sigma_1 \end{pmatrix}\right)$. *Then, IPF step does not change optimality matrix A, i.e.,*

$$
A = \Xi(\tilde{P}, \Sigma_0, S) = \Xi(P', Q, \Sigma_1).
$$

*For the second IPF step, the results are analogous.*

*Proof.* The explicit formulas for $\Xi(\tilde{P}, \Sigma_0, S)$ and $\Xi(P', Q, \Sigma_1)$ are

$$
\begin{aligned}
\Xi(\tilde{P}, \Sigma_0, S) &= S^{-1}\tilde{P}^\top \cdot (\Sigma_0 - \tilde{P}S^{-1}\tilde{P}^\top), \\
\Xi(P', Q, \Sigma_1) &= \Sigma_1^{-1}(P')^\top \cdot (Q - P'\Sigma_1^{-1}(P')^\top).
\end{aligned}
$$

The first terms are equal due to equation (30), and the second terms are equal due to (29).

We can prove this lemma in more general way. We derive the formula (25) for $A$ only from the shape of the conditional distribution $q(x_0|x_1)$ (24). During IPF step, this distribution remains the same by design, while parameters $S, \tilde{P}$ change. Hence, IPF step has no effect on the optimality matrix.

For the second IPF step, the proof is similar. $\square$

### D.3 DISCRETE IMF STEP ANALYSIS: MULTIDIMENSIONAL CASE FOR LARGE $\epsilon$

Consider a $2D$-dimensional Gaussian distribution $\mathcal{N}\left(\begin{pmatrix} \eta \\ \nu \end{pmatrix}, \begin{pmatrix} Q & P \\ P^\top & S \end{pmatrix}\right)$. We run a discrete IMF step consisting of reciprocal and Markovian projections with $N$ intermediate timesteps $0 = t_0 < t_1 < \cdots < t_N < t_{N+1} = 1$ and volatility parameter $\epsilon$.

Following (Gushchin et al., 2024b), we have an explicit formula for the reciprocal step. For any $0 \le i, j \le N + 1$, we have marginal covariance $\Sigma_{t_i, t_i}$ at time moment $t_i$ and joint covariance $\Sigma_{t_i, t_j}$ between time moments $t_i$ and $t_j$:

$$
\begin{aligned}
\Sigma_{t_i, t_j} &= (1 - t_i)(1 - t_j)Q + (1 - t_i)t_j P + (1 - t_j)t_i P^\top + t_i t_j S + t_i(1 - t_j)\epsilon \\
&= (1 - t_i)(1 - t_j)Q + (1 - t_i)t_j Q^{1/2}P_n S^{1/2} + (1 - t_j)t_i S^{1/2}P_n^\top Q^{1/2} \\
&\quad + t_i t_j S + t_i(1 - t_j)\epsilon, \\
\Sigma_{t_i, t_i} &= (1 - t_i)^2 Q + t_i(1 - t_i)(P + P^\top) + t_i^2 S + t_i(1 - t_i)\epsilon \\
&= (1 - t_i)^2 Q + t_i(1 - t_i)(Q^{1/2}P_n S^{1/2} + S^{1/2}P_n^\top Q^{1/2}) + t_i^2 S + t_i(1 - t_i)\epsilon, \\
\Sigma_{0, t_1} &= (1 - t_1)Q + t_1 P = (1 - t_1)Q + t_1 Q^{1/2}P_n S^{1/2}, \\
\Sigma_{t_N, 1} &= t_N S + (1 - t_N)P = t_N S + (1 - t_N)Q^{1/2}P_n S^{1/2},
\end{aligned}
$$

where $P_n := Q^{-\frac{1}{2}}PS^{-\frac{1}{2}}$. Marginals $\Sigma_{0,0} = Q$ and $\Sigma_{t_{N+1}, t_{N+1}} = S$ at time moments 0 and 1 and covariance $\Sigma_{t_0, t_{N+1}} = P$ do not change.

For the Markovian step, we write down an analytical formula for the new correlation $\tilde{P}$ in the resulting process $\mathcal{N}\left(\begin{pmatrix} \eta \\ \nu \end{pmatrix}, \begin{pmatrix} Q & \tilde{P} \\ \tilde{P}^\top & S \end{pmatrix}\right)$, namely:

$$
\tilde{P} := \Sigma_{0,0} \cdot \prod_{i=0}^{N} \left(\Sigma_{t_i, t_i}^{-1}\Sigma_{t_i, t_{i+1}}\right) = \Sigma_{0, t_1}\Sigma_{t_1, t_1}^{-1}\Sigma_{t_1, t_2} \ldots \Sigma_{t_N, t_N}^{-1}\Sigma_{t_N, 1}.
$$

For the normalized correlation $\tilde{P}_n = Q^{-\frac{1}{2}}\tilde{P}S^{-\frac{1}{2}} = \Sigma_{0,0}^{-\frac{1}{2}}\tilde{P}\Sigma_{1,1}^{-\frac{1}{2}}$, we can simplify the formula:

$$
\begin{aligned}
\tilde{P}_n = f(P_n) \quad =: \quad & \Sigma_{0,0}^{-\frac{1}{2}}\Sigma_{0,t_1}\Sigma_{t_1,t_1}^{-1/2} \cdot \Sigma_{t_1,t_1}^{-1/2}\Sigma_{t_1,t_2}\Sigma_{t_2,t_2}^{-1/2} \cdot \Sigma_{t_1,t_1}^{-1/2} \dots \Sigma_{t_N,t_N}^{-1/2} \cdot \Sigma_{t_N,t_N}^{-1/2}\Sigma_{t_N,1}\Sigma_{1,1}^{-\frac{1}{2}} \\
= \quad & \prod_{i=0}^{N}\left(\Sigma_{t_i,t_i}^{-1/2}\Sigma_{t_i,t_{i+1}}\Sigma_{t_{i+1},t_{i+1}}^{-1/2}\right) = \prod_{i=0}^{N}\left(\Sigma_{n;t_i,t_{i+1}}\right), \quad (44)
\end{aligned}
$$

where $\Sigma_{n;t_i,t_{i+1}} := \Sigma_{t_i,t_i}^{-1/2}\Sigma_{t_i,t_{i+1}}\Sigma_{t_{i+1},t_{i+1}}^{-1/2}$ denotes normalized correlation between marginals at time moments $t_i$ and $t_{i+1}$ and satisfies $\|\Sigma_{n;t_i,t_{i+1}}\|_2 \leq 1$.

**Lemma D.4** (IMF step correlation transition properties). *Let matrices $Q, S \succ 0$ be the marginals of 2D-dimensional Gaussian distribution $\mathcal{N}\left(\begin{pmatrix}\eta\\\nu\end{pmatrix}, \begin{pmatrix}Q & P\\P^\top & S\end{pmatrix}\right)$. The function $f$ from (44) defined on the ball $\{P_n : \|P_n\|_2 \leq 1\}$ transforms the normalized correlation $P_n := Q^{-\frac{1}{2}}PS^{-\frac{1}{2}}$ to a new one after a discrete IMF step. Then $f(P_n)$ is Lipschitz on the unit ball with constant*

$$
\gamma(Q,S,\epsilon) := \frac{\|Q^{\frac{1}{2}}\|_2\|S^{\frac{1}{2}}\|_2}{\sqrt{\epsilon}}\left(\sqrt{\frac{t_1\|Q^{-\frac{1}{2}}\|_2^2}{(1-t_1)}} + \sqrt{\frac{t_N\|S^{-\frac{1}{2}}\|_2^2}{(1-t_N)}} + \sum_{i=1}^{N-1}\frac{(1-t_i)t_{i+1} + (1-t_{i+1})t_i}{\sqrt{\epsilon t_i t_{i+1}(1-t_i)(1-t_{i+1})}}\right). \quad (45)
$$

*Moreover,*

$$
\|f(P_n)\|_2 \leq 1 - \frac{t_1 t_N(1-t_1)(1-t_N)\epsilon}{(\|Q^{1/2}\|_2 + \|S^{1/2}\|_2 + \sqrt{\epsilon})^2}. \quad (46)
$$

*Proof.* We differentiate $f(P_n)$ w.r.t. $P_n$ and obtain

$$
\begin{aligned}
df \quad = \quad & d\left(\Sigma_{0,0}^{1/2} \cdot \prod_{i=0}^{N}\left(\Sigma_{t_i,t_i}^{-1}\Sigma_{t_i,t_{i+1}}\right) \cdot \Sigma_{1,1}^{-1/2}\right) \\
= \quad & \sum_{i=0}^{N}\left(\Sigma_{0,0}^{1/2} \cdot \prod_{l<i}(\Sigma_{t_l,t_l}^{-1}\Sigma_{t_l,t_{l+1}}) \cdot (\Sigma_{t_i,t_i}^{-1}d\Sigma_{t_i,t_{i+1}} - \Sigma_{t_i,t_i}^{-1}d\Sigma_{t_i,t_i}\Sigma_{t_i,t_i}^{-1}\Sigma_{t_i,t_{i+1}}) \cdot \prod_{j>i}(\Sigma_{t_j,t_j}^{-1}\Sigma_{t_j,t_{j+1}}) \cdot \Sigma_{1,1}^{-1/2}\right) \\
= \quad & \sum_{i=0}^{N}\left(\prod_{l<i}(\Sigma_{n;t_l,t_{l+1}}) \cdot (\Sigma_{t_i,t_i}^{-1/2}d\Sigma_{t_i,t_{i+1}}\Sigma_{t_{i+1},t_{i+1}}^{-1/2}) \cdot \prod_{j>i}(\Sigma_{n;t_j,t_{j+1}})\right) \\
- \quad & \sum_{i=0}^{N}\left(\prod_{l<i}(\Sigma_{n;t_l,t_{l+1}}) \cdot (\Sigma_{t_i,t_i}^{-1/2}d\Sigma_{t_i,t_i}\Sigma_{t_i,t_i}^{-1/2}) \cdot \prod_{j\geq i}(\Sigma_{n;t_j,t_{j+1}})\right).
\end{aligned}
$$

Since all normalized correlations are bounded by $\|\Sigma_{n;t_i,t_{i+1}}\|_2 \leq 1$, we can also bound $df$ by

$$
\begin{aligned}
\|df\|_2 \quad \leq \quad & \sum_{i=0}^{N}(\|\Sigma_{t_i,t_i}^{-1/2}d\Sigma_{t_i,t_{i+1}}\Sigma_{t_{i+1},t_{i+1}}^{-1/2}\|_2 + \|\Sigma_{t_i,t_i}^{-1/2}d\Sigma_{t_i,t_i}\Sigma_{t_i,t_i}^{-1/2}\|_2) \\
\leq \quad & \sum_{i=0}^{N}(\|\Sigma_{t_i,t_i}^{-1/2}\|_2\|d\Sigma_{t_i,t_{i+1}}\|_2\|\Sigma_{t_{i+1},t_{i+1}}^{-1/2}\|_2 + \|\Sigma_{t_i,t_i}^{-1/2}\|_2\|d\Sigma_{t_i,t_i}\|_2\|\Sigma_{t_i,t_i}^{-1/2}\|_2).
\end{aligned}
$$

Since $\Sigma_{t_i,t_i} \succeq t_i(1-t_i)\epsilon I, \forall i \in \overline{1,N}$, we get estimate $\|\Sigma_{t_i,t_i}^{-1/2}\|_2 \leq 1/\sqrt{t_i(1-t_i)\epsilon}$. For differential $d\Sigma_{t_i,t_{i+1}}$, we have explicit formula and bound

$$
\begin{aligned}
d\Sigma_{t_i,t_{i+1}} \quad = \quad & (1-t_i)t_{i+1}Q^{1/2}dPS^{1/2} + (1-t_{i+1})t_iS^{1/2}dP^\top Q^{1/2}, \\
\|d\Sigma_{t_i,t_{i+1}}\|_2 \quad \leq \quad & ((1-t_i)t_{i+1} + (1-t_{i+1})t_i)\|Q^{1/2}\|_2\|S^{1/2}\|_2\|dP\|_2.
\end{aligned}
$$

In total, we can bound $\|df\|_2 \leq \gamma(Q,S,\epsilon)\|dP_n\|_2$ where $\gamma(Q,S,\epsilon) =$

$$
\frac{\|Q^{1/2}\|_2\|S^{1/2}\|_2}{\sqrt{\epsilon}}\left(\sqrt{\frac{t_1}{(1-t_1)}}\|Q^{-1/2}\|_2 + \sqrt{\frac{t_N}{(1-t_N)}}\|S^{-1/2}\|_2 + \sum_{i=1}^{N-1}\frac{(1-t_i)t_{i+1} + (1-t_{i+1})t_i}{\sqrt{\epsilon t_i t_{i+1}(1-t_i)(1-t_{i+1})}}\right).
$$

Now we prove (46). We bound the norm of $f$ using formula (44)

$$
\begin{aligned}
\|f(P_n)\|_2 &= \|\prod_{i=0}^{N}(\Sigma_{n;t_i,t_{i+1}})\|_2 \le \prod_{i=0}^{N}(\|\Sigma_{n;t_i,t_{i+1}}\|_2) \le \|\Sigma_{n;t_0,t_1}\|_2 \cdot \|\Sigma_{n;t_N,t_{N+1}}\|_2 \\
&\le \|Q^{-1/2}\Sigma_{0,t_1}\Sigma_{t_1,t_1}^{-1/2}\|_2 \cdot \|\Sigma_{t_N,t_N}^{-1/2}\Sigma_{t_N,1}S^{-1/2}\|_2 \\
&= \|((1-t_1)Q^{1/2} + t_1 P_n S^{1/2})\Sigma_{t_1,t_1}^{-1/2}\|_2 \cdot \|\Sigma_{t_N,t_N}^{-1/2}(t_N S^{1/2} + (1-t_N)Q^{1/2}P)\|_2.
\end{aligned}
$$

We note that $I - P_n^\top P_n \succcurlyeq 0, I - P_n P_n^\top \succcurlyeq 0$ and

$$
\begin{aligned}
(Q^{-1/2}\Sigma_{0,t_1})^\top(Q^{-1/2}\Sigma_{0,t_1}) &= ((1-t_1)Q^{1/2} + t_1 S^{1/2}P_n^\top)((1-t_1)Q^{1/2} + t_1 P_n S^{1/2}) \\
&= (1-t_1)^2 Q + t_1^2 S^{1/2}P_n^\top P_n S^{1/2} \\
&\quad + t_1(1-t_1)[S^{1/2}P_n^\top Q^{1/2} + Q^{1/2}P_n S^{1/2}] \\
&= \Sigma_{t_1,t_1} - t_1^2 S^{1/2}(I - P_n^\top P_n)S^{1/2} - t_1(1-t_1)\epsilon I \\
&\preccurlyeq \Sigma_{t_1,t_1} - t_1(1-t_1)\epsilon I.
\end{aligned}
$$

Similarly, we have

$$
\begin{aligned}
(\Sigma_{t_N,1}S^{-1/2})(\Sigma_{t_N,1}S^{-1/2})^\top &= \Sigma_{t_N,t_N} - (1-t_N)^2 Q^{1/2}(I - P_n P_n^\top)Q^{1/2} - t_N(1-t_N)\epsilon I \\
&\preccurlyeq \Sigma_{t_N,t_N} - t_N(1-t_N)\epsilon I.
\end{aligned}
$$

Next, we consider

$$
\begin{aligned}
\|((1-t_1)Q^{1/2} + t_1 P_n S^{1/2})\Sigma_{t_1,t_1}^{-1/2}\|_2^2 &= \|\Sigma_{t_1,t_1}^{-1/2}((1-t_1)Q^{1/2} + t_1 P_n S^{1/2})^\top((1-t_1)Q^{1/2} + t_1 P_n S^{1/2})\Sigma_{t_1,t_1}^{-1/2}\|_2 \\
&= \|\Sigma_{t_1,t_1}^{-1/2}(\Sigma_{t_1,t_1} - t_1^2 S^{1/2}(I - P_n^\top P_n)S^{1/2} - t_1(1-t_1)\epsilon)\Sigma_{t_1,t_1}^{-1/2}\|_2 \\
&\le \|\Sigma_{t_1,t_1}^{-1/2}(\Sigma_{t_1,t_1} - t_1(1-t_1)\epsilon)\Sigma_{t_1,t_1}^{-1/2}\|_2 \\
&= \|I - t_1(1-t_1)\epsilon\Sigma_{t_1,t_1}^{-1}\|_2 \le 1 - t_1(1-t_1)\epsilon\lambda_{min}(\Sigma_{t_1,t_1}^{-1}) \\
&\le 1 - t_1(1-t_1)\epsilon/\lambda_{max}(\Sigma_{t_1,t_1}) = 1 - t_1(1-t_1)\epsilon/\|\Sigma_{t_1,t_1}\|_2. \quad (47)
\end{aligned}
$$

We also can see that

$$
\begin{aligned}
\|\Sigma_{t_i,t_i}\|_2 &\le (1-t_i)^2\|Q\|_2 + 2t_i(1-t_i)\|Q^{1/2}\|_2\|S^{1/2}\|_2 + t_i^2\|S\|_2 + t_i(1-t_i)\epsilon \\
&\le ((1-t_i)\|Q^{1/2}\|_2 + t_i\|S^{1/2}\|_2)^2 + t_i(1-t_i)\epsilon.
\end{aligned}
$$

Thus, we conclude that

$$
\begin{aligned}
\|((1-t_1)Q^{1/2} + t_1 P_n S^{1/2})\Sigma_{t_1,t_1}^{-1/2}\|_2^2 &\le 1 - \frac{t_1(1-t_1)\epsilon}{((1-t_1)\|Q^{1/2}\|_2 + t_1\|S^{1/2}\|_2)^2 + t_1(1-t_1)\epsilon} \\
&\le 1 - \frac{t_1(1-t_1)\epsilon}{(\|Q^{1/2}\|_2 + \|S^{1/2}\|_2 + \sqrt{\epsilon})^2}
\end{aligned}
$$

Similarly, we have

$$
\|\Sigma_{t_N,t_N}^{-1/2}(t_N S^{1/2} + (1-t_N)Q^{1/2}P)\|_2^2 \le 1 - \frac{t_N(1-t_N)\epsilon}{(\|Q^{1/2}\|_2 + \|S^{1/2}\|_2 + \sqrt{\epsilon})^2}.
$$

The final result follows:

$$
\begin{aligned}
\|f(P_n)\|_2 &\le \|((1-t_1)Q^{1/2} + t_1 P_n S^{1/2})\Sigma_{t_1,t_1}^{-1/2}\|_2 \cdot \|\Sigma_{t_N,t_N}^{-1/2}(t_N S^{1/2} + (1-t_N)Q^{1/2}P)\|_2 \\
&\le 1 - \frac{t_1 t_N(1-t_1)(1-t_N)\epsilon}{(\|Q^{1/2}\|_2 + \|S^{1/2}\|_2 + \sqrt{\epsilon})^2}.
\end{aligned}
$$

$\square$

Next, we switch from tracking the changes of normalized correlation matrices to tracking the changes of the optimality matrices. Recall that, for a 2D-dimensional Gaussian process $\mathcal{N}\left(\begin{pmatrix} \eta \\ \nu \end{pmatrix}, \begin{pmatrix} Q & P \\ P^\top & S \end{pmatrix}\right)$, the optimality matrix $A$ from the definition (10) is calculated as

$$
A(P) = \Xi(P, Q, S) = (S)^{-1}P^\top(Q - P(S)^{-1}P^\top)^{-1}.
$$

Functions $\Xi(P_n, Q, S)$ and $A(P_n)$ can take the normalized correlation $P_n := Q^{-\frac{1}{2}} P S^{-\frac{1}{2}}$ as the first argument. In this case, the formulas and notations are

$$\Xi_n(P_n, Q, S) := S^{-1/2} P_n^\top \left(I - P_n P_n^\top\right)^{-1} Q^{-1/2}, \quad A(P_n) = \Xi_n(P_n, Q, S). \quad (48)$$

**Lemma D.5** (Optimality matrix map properties). *Let matrices $Q, S \succ 0$ be the marginals of a 2D-dimensional Gaussian distribution $\mathcal{N}\left(\begin{pmatrix} \eta \\ \nu \end{pmatrix}, \begin{pmatrix} Q & P \\ P^\top & S \end{pmatrix}\right)$ with the normalized correlation $P_n := Q^{-\frac{1}{2}} P S^{-\frac{1}{2}}$. Then the map from normalized correlations to optimality matrices $A(P_n) = S^{-1/2} P_n^\top \left(I - P_n P_n^\top\right)^{-1} Q^{-1/2}$ is bi-Lipschitz on the set $\{P_n \in \mathbb{R}^{D \times D} : \|P_n\|_2 \le \sqrt{1-\omega}\}$ for any $0 < \omega < 1$. Specifically, for any $P_n$ and $\tilde{P}_n$ from this set, the following inequalities hold*

$$L\|P_n - \tilde{P}_n\|_2 \le \|A(P_n) - A(\tilde{P}_n)\|_2 \le M_\omega \|P_n - \tilde{P}_n\|_2,$$

*where*

$$L = \frac{1}{\sqrt{2D}\|S\|_2^{1/2} \cdot \|Q\|_2^{1/2}}, \quad M_\omega = \|S^{-1}\|_2^{1/2} \cdot \|Q^{-1}\|_2^{1/2} \left(\frac{1}{\omega} + \frac{2}{\omega^2}\right).$$

Before proving the lemma, we introduce some notations. Let $h$ be a scalar function. For any diagonal matrix $\Lambda = \mathrm{diag}(\lambda_1, \ldots, \lambda_D)$, we define

$$h(\Lambda) = \mathrm{diag}\big(h(\lambda_1), \ldots, h(\lambda_D)\big).$$

Next, given a symmetric matrix $B \in \mathbb{R}^{D \times D}$ with spectral decomposition $B = Z\Lambda Z^\top$, we set

$$h(B) = Zh(\Lambda)Z^\top.$$

*Proof.* To estimate $M_\omega$, we differentiate $A(P_n)$ w.r.t. $P_n$ that

$$\begin{aligned} dA &= S^{-1/2} P_n^\top \left(I - P_n P_n^\top\right)^{-1} (dP_n P_n^\top + P_n dP_n^\top) \left(I - P_n P_n^\top\right)^{-1} Q^{-1/2} \\ &+ S^{-1/2} dP_n^\top \left(I - P_n P_n^\top\right)^{-1} Q^{-1/2}. \end{aligned} \quad (49)$$

By the conditions of the lemma, $0 \preccurlyeq P_n P_n^\top \preccurlyeq (1-\omega)I$, hence $\left\|\left(I - P_n P_n^\top\right)^{-1}\right\|_2 \le \frac{1}{\omega}$ and $\|P_n\|_2 \le 1$. Thus,

$$\|dA\|_2 \le \|S^{-1/2}\|_2 \|Q^{-1/2}\|_2 \left(\frac{1}{\omega} + \frac{2}{\omega^2}\right) \|dP_n\|_2.$$

Since the ball $\{P_n : \|P_n\|_2 \le \sqrt{1-\omega}\}$ is convex, this yields the bound $M_\omega$ on the Lipschitz constant.

To estimate $L$, we define $B = S^{1/2} A Q^{1/2} = P_n^\top \left(I - P_n P_n^\top\right)^{-1}$ and note that

$$B^\top B = \left(I - P_n P_n^\top\right)^{-1} P_n P_n^\top \left(I - P_n P_n^\top\right)^{-1} = \left(I - P_n P_n^\top\right)^{-2} - \left(I - P_n P_n^\top\right)^{-1}.$$

Next, we define $h(x) = \frac{2}{1+\sqrt{1+4x}}$, $x \ge 0$, so that $h^{-1}(y) = y^{-2} - y^{-1}$, $0 < y \le 1$. Therefore, we have

$$I - P_n P_n^\top = h(B^\top B), \quad (50)$$
$$P_n^\top = B \left(I - P_n P_n^\top\right) = Bh(B^\top B).$$

For now, consider $B$ such that its singular values are positive and distinct (note that the set of such matrices is dense in $\mathbb{R}^{D \times D}$). Then the SVD map $B \mapsto (U, \Lambda, V)$ such that $B = U\Lambda V^*$ is differentiable at $B$ (see Magnus & Neudecker, 2019, Section 3.8.8), thus so is the polar decomposition map $B \mapsto (Q, S)$ such that $B = KC$, where $K$ is orthogonal and $C$ is PSD matrices. As

$$P_n^\top = Bh(B^\top B) = KCh(C^2) = U\Lambda h(\Lambda^2)V^*$$

and $xh(x^2)$ is differentiable, we obtain that

$$dP_n^\top = dKCh(C^2) + Kd(Ch(C^2)).$$

Furthermore, $0 < h(x) \le 1$ and $(xh(x^2))' = \frac{2}{(1+\sqrt{1+4x^2})\sqrt{1+4x^2}} \in (0,1]$, hence $0 \prec h(S^2) \preccurlyeq I$ and $Ch(C^2)$ is 1-Lipschitz w.r.t. the Frobenius norm (Wihler, 2009, Thm. 1.1). Note that $\|KdC\|_F^2 = \text{Tr}[(KdC)^\top dKC] = \text{Tr}[(CdC)(K^\top dK)] = 0$ since $K^\top dK$ is skew-symmetric. It can be shown from the orthogonality of $K$:

$$I = K^\top K \quad \Rightarrow \quad 0 = dI = dK^\top \cdot K + K^\top dK \quad \Rightarrow \quad dK^\top \cdot K = -K^\top dK.$$

Thus, we have

$$\|dB\|_F^2 = \|dKC + KdC\|_F^2 = \|dKC\|_F^2 + \|KdC\|_F^2 = \|dKC\|_F^2 + \|dC\|_F^2.$$

Therefore,

$$
\begin{aligned}
\|dP_n\|_F &= \|dKCh(C^2) + Kd(Ch(C^2))\|_F \le \|dKC\|_F \|h(C^2)\|_2 + \|d(Ch(C^2))\|_F \\
&\le \|dKC\|_F + \|dC\|_F \le \sqrt{2}\|dB\|_F.
\end{aligned}
\tag{51}
$$

In particular,

$$\|dP_n\|_2 \le \|dP_n\|_F \le \sqrt{2}\|dB\|_F \le \sqrt{2D}\|dB\|_2 \le \sqrt{2D}\|S^{1/2}\|_2 \|Q^{1/2}\|_2 \|dA\|_2.$$

By continuity of the SVD and thus of the map $Bh(B^\top B)$, this yields that

$$L^{-1} = \sqrt{2D}\|S^{1/2}\|_2 \|Q^{1/2}\|_2.$$

$\square$

Now we can show that the function, changing the optimality matrix during IMF step, is Lipschitz. This function is constructed as follows: first, it transforms optimality matrix into the normalized correlation via $\Xi_n^{-1}$, then it makes an IMF step to obtain new normalized correlation via $f$ from (44), finally it transforms new correlation back to new optimality matrix via $\Xi_n$.

**Corollary D.6** (IMF step optimality matrix transition properties). *Let matrices $Q, S \succ 0$ be the marginals of a 2D-dimensional Gaussian distribution $\mathcal{N}\left(\begin{pmatrix} \eta \\ \nu \end{pmatrix}, \begin{pmatrix} Q & P \\ P^\top & S \end{pmatrix}\right)$ with the optimality matrix $A$ defined in (48). Set the function $g(A) := \Xi_n(f(\Xi_n^{-1}(A, Q, S)), Q, S)$, where $\Xi_n$ and $f$ defined in (48) and (44), and $\Xi_n^{-1}(\cdot; Q, S)$ denotes the inverse map of $\Xi_n$ w.r.t. the first argument.*

*Then $g$ is Lipschitz continuous with constant $\frac{M_\omega}{L}\gamma$ on the set $\{A \mid \|\Xi_n^{-1}(A)\|_2 \le \sqrt{1-\omega}\}$ for any $0 < \omega < 1$.*

*Proof.* Lipschitz constant of the functions composition is the product of Lipschitz constants of the combined functions. From Lemma D.5, we know that the constant for $\Xi_n$ is $M_\omega$, for $\Xi_n^{-1}$ is $1/L$ as inverse of $\Xi_n$. For transition function $f$, the constant $\gamma$ comes from Lemma D.4. $\square$

We also prove the upper bound for the normalized correlation for further proofs.

**Corollary D.7** (Bound for $\|P_n\|_2^2$). *Let matrices $Q, S \succ 0$ be the marginals of a 2D-dimensional Gaussian distribution $\mathcal{N}\left(\begin{pmatrix} \eta \\ \nu \end{pmatrix}, \begin{pmatrix} Q & P \\ P^\top & S \end{pmatrix}\right)$ with optimality matrix $A = \Xi(P, Q, S)$ and normalized correlation $P_n = Q^{-\frac{1}{2}}PS^{-\frac{1}{2}}$. Then the following bound holds true:*

$$\|P_n\|_2^2 \le 1 - \frac{2}{1 + \sqrt{1 + 4\|Q\|_2\|S\|_2\|A\|_2^2}}.
\tag{52}$$

*Proof.* We recall the explicit formula (50) connecting $P_n$ and $A$:

$$I - P_n P_n^\top = h(B^\top B),
\tag{53}$$

where matrix $B := S^{1/2}AQ^{1/2}$ and scalar function $h(x) := \frac{2}{1+\sqrt{1+4x}}, x \ge 0$. Given a $D \times D$ symmetric positive definite matrix $C$ with spectral decomposition $C = U\Lambda U^*$, we set $h(C) = Uh(\Lambda)U^*$. We start with estimate

$$\lambda_{min}(h(B^\top B)) = \lambda_{min}(I - P_n P_n^\top) = 1 - \lambda_{max}(P_n P_n^\top) = 1 - \|P_n\|_2^2.
\tag{54}$$

Since function $h$ is monotonously decreasing on $[0, +\infty)$ and matrix $B^\top B$ has non-negative eigenvalues, we have $\lambda_{min}(h(B^\top B)) = h(\lambda_{max}(B^\top B))$ and continue with:

$$
\begin{aligned}
\lambda_{min}(h(B^\top B)) &= h(\lambda_{max}(B^\top B)) = h(\|B^\top B\|_2) \\
&\geq h(\|A\|_2^2 \|\Sigma\|_2 \|\tilde{\Sigma}\|_2) \\
&= \frac{2}{1 + \sqrt{1 + 4\|Q\|_2\|S\|_2\|A\|_2^2}}.
\end{aligned}
$$

Combining bounds together, we conclude:

$$
\|P_n\|_2^2 \leq 1 - \frac{2}{1 + \sqrt{1 + 4\|Q\|_2\|S\|_2\|A\|_2^2}}.
$$

$\square$

Finally, we are ready to demonstrate convergence of the optimality matrix to the desired solution $A^* = \epsilon^{-1} I_d$ after an IMF step.

**Lemma D.8.** *IMF step convergence Let matrices $Q, S \succ 0$ be the marginals of a 2D-dimensional Gaussian distribution $\mathcal{N}\left(\begin{pmatrix} \eta \\ \nu \end{pmatrix}, \begin{pmatrix} Q & P \\ P^\top & S \end{pmatrix}\right)$ with the optimality matrix $A$. Then after IMF step, we obtain a new optimality matrix $\tilde{A} = g(A)$ (see Corollary D.6), satisfying the inequality*

$$
\|\tilde{A} - \epsilon^{-1} I_d\|_2 \leq \frac{M_\omega}{L} \gamma(Q, S, \epsilon) \|A - \epsilon^{-1} I_d\|_2,
$$

*where*

$$
\begin{aligned}
\gamma &= \frac{\|Q^{\frac{1}{2}}\|_2 \|S^{\frac{1}{2}}\|_2}{\sqrt{\epsilon}} \left( \sqrt{\frac{t_1 \|Q^{-\frac{1}{2}}\|_2^2}{(1 - t_1)}} + \sqrt{\frac{t_N \|S^{-\frac{1}{2}}\|_2^2}{(1 - t_N)}} + \sum_{i=1}^{N-1} \frac{(1 - t_i)t_{i+1} + (1 - t_{i+1})t_i}{\sqrt{\epsilon t_i t_{i+1}(1 - t_i)(1 - t_{i+1})}} \right), \\
\omega &= \min\left\{ 1 - \|P_n\|_2^2, 1 - \frac{t_1 t_N (1 - t_1)(1 - t_N)\epsilon}{(\|Q^{1/2}\|_2 + \|S^{1/2}\|_2 + \sqrt{\epsilon})^2} \right\}, \\
L^{-1} &= \sqrt{2D} \|S^{1/2}\|_2 \|Q^{1/2}\|_2, \\
M_\omega &= \|S^{-1}\|_2^{1/2} \cdot \|Q^{-1}\|_2^{1/2} \left( \frac{1}{\omega} + \frac{2}{\omega^2} \right).
\end{aligned}
$$

*Proof.* The IMF method with volatility parameter $\epsilon$ can be viewed as an iterative application of the transition function $g$. Since IMF converges to $A^* = \epsilon^{-1} I_d$, it follows that $A^*$ is a stationary point of $g$, i.e., $g(A^*) = A^*$. Hence, we apply Corollary D.6 to get

$$
\|\tilde{A} - A^*\|_2 = \|g(A) - g(A^*)\|_2 \leq \frac{M_\omega}{L} \gamma \|A - A^*\|_2,
$$

where explicit values for $\gamma$ and $\omega, M_\omega, L$ are taken from Lemmas D.4 and D.5, respectively. We only need to satisfy condition on $\omega$ from Corollary D.6 for matrices $A, \tilde{A}, A^*$:

$$
\|\Xi_n^{-1}(A)\|_2 \leq \sqrt{1 - \omega}.
$$

In terms of normalized correlations $P_n, \tilde{P}_n = f(P_n)$ from (44) and $P_n^* = \Xi_n^{-1}(A^*)$ from (48), the conditions are

$$
1 - \|P_n\|_2^2 \geq \omega, \quad 1 - \|\tilde{P}_n\|_2^2 \geq \omega. \tag{55}
$$

For the second inequality in (55), we use bound (46) from Lemma (D.4) for the result of applying $f$:

$$
1 - \|\tilde{P}_n\|_2^2 = 1 - \|f(P_n)\|_2^2 \geq \frac{t_1 t_N (1 - t_1)(1 - t_N)\epsilon}{(\|Q^{1/2}\|_2 + \|S^{1/2}\|_2 + \sqrt{\epsilon})^2}.
$$

Finally, we combine all the bounds under the single minimum. $\square$

### D.4 PROOF OF D-IPMF CONVERGENCE THEOREM 3.2, $D > 1$

*Proof.* We denote by $Q_0$ marginal matrix at $t = 0$ after the first IPF step. First, we note that all marginal matrices $Q$ at $t = 0$ and $S$ at $t = 1$ emerging during IPMF procedure are bounded by the initial ones (Lemma D.2):

$$\|S\|_2 \ \leq \ \frac{\|\Sigma_1\|_2}{\min\{\lambda_{min}(S_0^{-\frac{1}{2}}\Sigma_1 S_0^{-\frac{1}{2}}), 1\}} =: u_S, \|S^{-1}\|_2 \leq \max\{\lambda_{max}(S_0^{-\frac{1}{2}}\Sigma_1 S_0^{-\frac{1}{2}}), 1\}\|\Sigma_1^{-1}\|_2 =: r_S \tag{56}$$

$$\|Q\|_2 \ \leq \ \frac{\|\Sigma_0\|_2}{\min\{\lambda_{min}(Q_0^{-\frac{1}{2}}\Sigma_0 Q_0^{-\frac{1}{2}}), 1\}} =: u_Q, \|Q^{-1}\|_2 \leq \max\{\lambda_{max}(Q_0^{-\frac{1}{2}}\Sigma_0 Q_0^{-\frac{1}{2}}), 1\}\|\Sigma_0^{-1}\|_2 =: r_Q \tag{57}$$

**Optimality convergence and condition on $\epsilon$.** Consider any IMF step during IPMF procedure which we denote by

$$\begin{pmatrix} Q & P \\ P^\top & S \end{pmatrix} \overset{IMF}{\Longrightarrow} \begin{pmatrix} Q & \tilde{P} \\ \tilde{P}^\top & S \end{pmatrix}, \quad P_n := Q^{-\frac{1}{2}}PS^{-\frac{1}{2}}.$$

We want to find such $\epsilon$ that new optimality matrix $\tilde{A} = \Xi(P, Q, S)$ becomes close to solution $A^* = \epsilon^{-1}I_D$ than starting $A = \Xi(P, Q, S)$. This transition from $A$ to $\tilde{A}$ satisfies (Lemma D.8):

$$\|\tilde{A} - A^*\|_2 \ \leq \ \left(\sqrt{2D}\left(\frac{1}{\omega} + \frac{2}{\omega^2}\right) \cdot \kappa(Q^{\frac{1}{2}})\kappa(S^{\frac{1}{2}})\right)\gamma(Q, S, \epsilon)\|A - A^*\|_2, \tag{58}$$

$$\gamma \qquad \text{is defined in (45)},$$

$$\omega \ = \ \min\left\{1 - \|P_n\|_2^2, 1 - \frac{t_1 t_N(1 - t_1)(1 - t_N)\epsilon}{(\|Q^{1/2}\|_2 + \|S^{1/2}\|_2 + \sqrt{\epsilon})^2}\right\}, \tag{59}$$

where $\kappa(\cdot)$ is condition number of a matrix.

**Estimate $\omega$.** The second term of $\omega$ in (59) can be lower bounded by

$$1 - \frac{t_1 t_N(1 - t_1)(1 - t_N)\epsilon}{(\|Q^{1/2}\|_2 + \|S^{1/2}\|_2 + \sqrt{\epsilon})^2} \geq 1 - t_1 t_N(1 - t_1)(1 - t_N). \tag{60}$$

To estimate $1 - \|P_n\|_2^2$ in the second term, we use lower bound (Corollary D.7):

$$1 - \|P_n\|_2^2 \ \geq \ \frac{2}{1 + \sqrt{1 + 4\|Q\|_2\|S\|_2\|A\|_2^2}} \geq \frac{1}{\sqrt{1 + 4\|Q\|_2\|S\|_2\|A\|_2^2}}.$$

Hence, we have lower bound for $\omega$:

$$\omega \geq \min\left\{\frac{1}{\sqrt{1 + 4\|Q\|_2\|S\|_2\|A\|_2^2}}, 1 - t_1 t_N(1 - t_1)(1 - t_N)\right\} \geq \frac{(1 - t_1 t_N(1 - t_1)(1 - t_N))}{\sqrt{1 + 4\|Q\|_2\|S\|_2\|A\|_2^2}}.$$

The change of difference norm after one IMF step is

$$\|A' - A_*\|_2 \leq \underbrace{6\sqrt{D} \cdot \kappa(Q^{\frac{1}{2}})\kappa(S^{\frac{1}{2}}) \cdot \frac{(1 + 4\|Q\|_2\|S\|_2\|A\|_2^2)}{(1 - t_1 t_N(1 - t_1)(1 - t_N))^2} \cdot \gamma(Q, S, \epsilon)}_{:=l(Q,S,\|A\|_2,\epsilon)} \cdot \|A - A_*\|_2. \tag{61}$$

Now we need to make this map contractive, i.e., bound the coefficient $l(Q, S, \|A\|_2, \epsilon) < 1$ for all matrices $Q, S, A$ appearing during IPMF procedure.

Universal bounds (57) and (56) state that matrices $Q$ and $S$ lie on matrix compacts $B_Q := \{Q \succ 0 | \|Q\|_2 \leq u_Q, \|Q^{-1}\|_2 \leq r_Q\}$ and $B_S := \{S \succ 0 | \|S\|_2 \leq u_S, \|S^{-1}\|_2 \leq r_S\}$, respectively. Moreover, the function $l(Q, S, \|A\|_2, \epsilon)$ is continuous w.r.t. all its parameters on these compacts. Hence, we can get rid of $Q, S$ dependency, since the following maximum is attained

$$l(\|A\|_2, \epsilon) = \max_{Q \in B_Q, S \in B_S} l(Q, S, \|A\|_2, \epsilon).$$

$\|A\|_2$ **dependency**. IPF steps do not change optimality matrices (Lemma D.3), hence, we consider only IMF steps here. We prove by induction that if at the first IMF step with initial optimality matrix

$A_0$ coefficient $l(\|A_0\|_2 + 2\epsilon^{-1}, \epsilon) < 1$ is less than 1, then all optimality matrices $\{A_i\}$ during IPMF procedure will be bounded by

$$\|A_i\|_2 \leq \|A_0\|_2 + 2\epsilon^{-1}, \quad \|A_i - A^*\|_2 \leq \|A_0 - A^*\|_2.$$

First, we note that the coefficient $l(\|A\|_2, \epsilon)$ is increasing w.r.t. $\|A\|_2$. As the base, we show that after the first IMF step new matrix $A_1$ is bounded:

$$
\begin{aligned}
\|A_1\|_2 &\leq \|A_1 - A^*\|_2 + \|A_*\|_2 \leq l(\|A_0\|_2, \epsilon)\|A_0 - A^*\|_2 + \|A^*\|_2 \\
&\leq l(\|A_0\|_2 + 2\epsilon^{-1}, \epsilon)\|A_0 - A^*\|_2 + \|A^*\|_2 \leq \|A_0 - A^*\|_2 + \|A^*\|_2 \\
&\leq \|A_0\|_2 + 2\|A^*\|_2 \leq \|A_0\|_2 + 2\epsilon^{-1}.
\end{aligned}
$$

Moreover, we have

$$\|A_1 - A^*\|_2 \leq \|A_0 - A^*\|_2.$$

Assume that the bounds $\|A_i\|_2 \leq \|A_0\|_2 + 2\epsilon^{-1}$ and $\|A_i - A^*\|_2 \leq \|A_0 - A^*\|_2$ hold for the $i$-th matrix, then, for the next matrix $A_{i+1}$, we prove:

$$
\begin{aligned}
\|A_{i+1} - A^*\|_2 &\leq l(\|A_i\|_2, \epsilon)\|A_i - A^*\|_2 \leq l(\|A_0\|_2 + 2\epsilon^{-1}, \epsilon)\|A_i - A^*\|_2 \\
&\leq \|A_i - A^*\|_2 \leq \|A_0 - A^*\|_2, \\
\|A_{i+1}\|_2 &\leq \|A_{i+1} - A^*\|_2 + \|A^*\|_2 \leq \|A_0 - A^*\|_2 + \|A^*\|_2 \leq \|A_0\|_2 + 2\epsilon^{-1}.
\end{aligned}
$$

Thus, we take maximal possible norm among all matrices $\|A_i\|_2 \leq \|A_0\|_2 + 2\epsilon^{-1}$ to upper bound the coefficient $l(\|A_i\|_2, \epsilon) \leq l(\|A_0\|_2 + 2\epsilon^{-1}, \epsilon) < 1$.

**The final condition (62) on $\epsilon$ is**

$$\beta(Q_0, S_0, P_0, \epsilon) := \max_{Q \in B_Q, S \in B_S} \left[ 6\sqrt{D} \cdot \kappa(Q^{\frac{1}{2}})\kappa(S^{\frac{1}{2}}) \cdot \frac{(1 + 4\|Q\|_2\|S\|_2(\|A_0\|_2 + 2\epsilon^{-1})^2)}{(1 - t_1 t_N(1 - t_1)(1 - t_N))^2} \cdot \gamma(Q, S, \epsilon) \right] < 1.$$

We can see from the definition

$$\gamma(Q, S, \epsilon) := \frac{\|Q^{\frac{1}{2}}\|_2\|S^{\frac{1}{2}}\|_2}{\sqrt{\epsilon}} \left( \sqrt{\frac{t_1\|Q^{-\frac{1}{2}}\|_2^2}{(1 - t_1)}} + \sqrt{\frac{t_N\|S^{-\frac{1}{2}}\|_2^2}{(1 - t_N)}} + \sum_{i=1}^{N-1} \frac{(1 - t_i)t_{i+1} + (1 - t_{i+1})t_i}{\sqrt{\epsilon t_i t_{i+1}(1 - t_i)(1 - t_{i+1})}} \right)$$

that the largest value of $\gamma$ is achieved when $\|Q^{1/2}\|_2, \|S^{1/2}\|_2, \|Q^{-1/2}\|_2, \|S^{-1/2}\|_2$ are the largest. Since these values are bounded by $\|Q^{1/2}\|_2 \leq \sqrt{u_Q}, \|S^{1/2}\|_2 \leq \sqrt{u_S}$ and $\|Q^{-1/2}\|_2 \leq \sqrt{r_Q}, \|S^{-1/2}\|_2 \leq \sqrt{r_Q}$, we can estimate the maximum and get lower bound for $\epsilon$:

$$
\begin{aligned}
\beta(Q_0, S_0, P_0, \epsilon) &\leq 6\sqrt{D \cdot u_Q r_Q u_S r_S} \cdot \frac{(1 + 4u_Q u_S(\|A_0\|_2 + 2\epsilon^{-1})^2)}{(1 - t_1 t_N(1 - t_1)(1 - t_N))^2} \\
&\cdot \frac{\sqrt{u_Q u_S}}{\sqrt{\epsilon}} \left( \sqrt{\frac{t_1 r_Q}{(1 - t_1)}} + \sqrt{\frac{t_N r_S}{(1 - t_N)}} + \sum_{i=1}^{N-1} \frac{(1 - t_i)t_{i+1} + (1 - t_{i+1})t_i}{\sqrt{\epsilon t_i t_{i+1}(1 - t_i)(1 - t_{i+1})}} \right) \leq 1, \\
\epsilon &= O\left( D \cdot r_Q^2 r_S^2 \cdot u_Q^4 u_S^4 \cdot \|A_0\|_2^4 \right). \quad (62)
\end{aligned}
$$

If the above $\epsilon$-condition (62) holds true, then $A_k$ exponentially converges to $A^*$ (square appears since IPMF step includes two IMF steps):

$$\|A_k - A^*\|_2 \leq \beta(Q_0, S_0, P_0, \epsilon)^{2k}\|A_0 - A^*\|_2.$$

**Marginals convergence.** Furthermore, we prove that marginals converge to ground truth $\Sigma_1$ as well. We note that, during any condition $\begin{pmatrix} Q & P \\ P^\top & S \end{pmatrix}$ of IPMF procedure, the norm of the normalized matrix $P_n = Q^{-\frac{1}{2}}PS^{-\frac{1}{2}}$ is bounded:

$$\|P_n\|_2^2 \leq 1 - \frac{2}{1 + \sqrt{1 + 4\|Q\|_2\|S\|_2\|A\|_2^2}}.$$

Since $\|Q\|_2 \leq u_Q$ holds from (57), $\|S\|_2 \leq u_S$ holds from (56) and $\|A\|_2 \leq \|A_0\|_2 + 2\epsilon^{-1}$ (due to contractivity of $A$), we can upper bound the normalized correlation

$$\|P_n\|_2^2 \leq 1 - \frac{2}{1 + \sqrt{1 + 4u_Q u_S(\|A_0\|_2 + 2\epsilon^{-1})^2}} =: \alpha(Q_0, S_0, P_0, \epsilon)^2 < 1.$$

Finally, we apply bounds from IPF steps Lemma D.1 at $k$-th step and put maximal norm value $\alpha(Q_0, S_0, P_0, \epsilon)^2$:

$$
\begin{aligned}
\|S_k^{-\frac{1}{2}} \Sigma_1 S_k^{-\frac{1}{2}} - I_D\|_2 &\leq \alpha(Q_0, S_0, P_0, \epsilon)^2 \cdot \|S_0^{-\frac{1}{2}} \Sigma_1 S_0^{-\frac{1}{2}} - I_D\|_2, \\
\|\Sigma_1^{-\frac{1}{2}} (\nu_k - \mu_1)\|_2 &\leq \alpha(Q_0, S_0, P_0, \epsilon)^2 \cdot \|\Sigma_1^{-\frac{1}{2}} (\nu_0 - \mu_1)\|_2.
\end{aligned}
$$

$\square$

### D.5 IMF STEP ANALYSIS IN $1D$

**Preliminaries.** In case $D = 1$, we change notation from matrices to scalars:

$$
\begin{pmatrix} q & \rho \\ \rho & s \end{pmatrix} \overset{IMF}{\Longrightarrow} \begin{pmatrix} q & \tilde{\rho} \\ \tilde{\rho} & s \end{pmatrix}, \quad \rho_n := \rho/\sqrt{sq}, \tilde{\rho}_n := \tilde{\rho}/\sqrt{sq}.
$$

Using these notations, formula (48) for optimality coefficient $\chi \in \mathbb{R}$ (instead of matrix $A$) can be expressed as

$$
\Xi_n(\rho_n, q, s) = \frac{\rho_n}{\sqrt{sq}(1 - \rho_n^2)} = \chi \in (-\infty, +\infty). \tag{63}
$$

The function $\Xi_n$ is monotonously increasing w.r.t. $\rho_n \in (-1, 1)$ and, thus, invertible, i.e., there exists a function $\Xi_n^{-1} : (-\infty, +\infty) \times \mathbb{R}_+ \times \mathbb{R}_+ \to (-1, 1)$ such that

$$
\Xi_n^{-1}(\chi, s, q) = \frac{\sqrt{\chi^2 sq + 1/4} - 1/2}{\chi \sqrt{sq}}. \tag{64}
$$

The inverse function is calculated via solving quadratic equation w.r.t. $\rho_n$.

In our paper, we consider both discrete and continuous IMF. By construction, IMF step does change marginals of the process it works with. Moreover, for both continuous and discrete IMF, the new correlation converges to the correlation of the $\epsilon$-EOT between marginals.

**Lemma D.9** (Correlation improvement after (D)IMF step). *Consider a 2-dimensional Gaussian distribution with marginals $\mathcal{N}(\eta, q)$ and $\mathcal{N}(\nu, s)$ and normalized correlation $\rho_n \in (-1, 1)$ between its components. After continuous IMF or DIMF with single time point $t$, we obtain normalized correlation $\tilde{\rho}_n$. The distance between $\tilde{\rho}_n$ and EOT correlation $\rho_n^* = \Xi_n^{-1}(1/\epsilon, q, s)$ decreases as:*

$$
|\tilde{\rho}_n - \rho_n^*| \leq \gamma \cdot |\rho_n - \rho_n^*|,
$$

*where factor $\gamma$ for continuous and discrete IMF (with $N = 1$) is, respectively,*

$$
\gamma_c(q, s) = \left| \frac{2\epsilon^2 qs \cdot f(0)}{(\epsilon^2 - 4q^2 s^2)^{\frac{3}{2}}} \left( \tanh^{-1}\left( \frac{\epsilon - 2q^2}{\sqrt{\epsilon - 4q^2 s^2}} \right) + \tanh^{-1}\left( \frac{\epsilon - 2s^2}{\sqrt{\epsilon - 4q^2 s^2}} \right) \right) - \frac{\sqrt{\epsilon^2 + 4q^2 s^2}}{\epsilon} \right| \tag{65}
$$

$$
\gamma_d(q, s, t) = \frac{1}{1 + \frac{t^2(1-t)^2 qs + t(1-t)(t^2 s + (1-t)^2 q)\epsilon + t^2(1-t)^2 \epsilon^2}{(1-t)^2((1-t)q + t\sqrt{qs})^2 + t^2(ts + (1-t)\sqrt{qs})^2 + t(1-t)((1-t)\sqrt{q} + t\sqrt{s})^2 \epsilon}}. \tag{66}
$$

*Proof.* **Continuous case.** Following (Peluchetti, 2023a, Eq. 42), we have the formula for $\tilde{\rho}_n$:

$$
\begin{aligned}
\tilde{\rho}_n = f(\rho_n) &= \exp\left\{ -\epsilon \frac{\tanh^{-1}\left( \frac{c_1}{c_3} \right) + \tanh^{-1}\left( \frac{c_2}{c_3} \right)}{c_3} \right\} > 0, \tag{67} \\
c_1 &= \epsilon + 2s(\rho_n q - s), c_3 = \sqrt{(\epsilon + 2(\rho_n + 1)qs)(\epsilon + 2(\rho_n - 1)qs)}, \\
c_2 &= \epsilon + 2q(\rho_n s - q).
\end{aligned}
$$

Note that the function $f(\rho_n)$ is positive and concave on $(-1, 1)$, i.e., its derivative is decreasing on $(-1, 1)$. Hence, in negative segment $(-1, 0]$, the distance until the fixed point $\rho_n^* > 0$ is decreasing faster, than in positive segment, and we need to deal only with the positive segment $[0, 1)$. We will show that the function $f$ has a derivative norm bounded by 1 on $[0, 1)$, and, hence, it is contractive. Due to concavity, its derivative is decreasing on $[0, 1)$, and we can check the bound only for derivative at the point $\rho_n = 0$. Direct calculation gives us:

$$
f'(0) = \frac{f(0) \cdot 2\epsilon qs}{(\epsilon^2 - 4q^2 s^2)^2} \left( \epsilon\sqrt{\epsilon - 4q^2 s^2} \left[ \tanh^{-1}\left( \frac{\epsilon - 2q^2}{\sqrt{\epsilon - 4q^2 s^2}} \right) + \tanh^{-1}\left( \frac{\epsilon - 2s^2}{\sqrt{\epsilon - 4q^2 s^2}} \right) \right] - \epsilon^2 + 4q^2 s^2 \right),
$$

$$\gamma_c(q,s) \quad =: \quad |f'(0)| < 1.$$

Thus, we can bound $|f'(\rho_n)| \leq \gamma_c(q,s), \forall \rho_n \in [0,1)$ and get on the whole interval $(-1,1)$

$$|\tilde{\rho}_n - \rho_n^*| = |f(\rho_n) - f(\rho_n^*)| \leq \gamma_c(q,s)|\rho_n - \rho_n^*|.$$

**Discrete case** ($N = 1$). We use explicit formula (44) for a new correlation $\tilde{\rho}_n = f(\rho_n)$ after D-IMF step from (Gushchin et al., 2024b) provided in the beginning of Section D.3.

In the case of single point $t = t_1$ ($N = 1$), we prove that the function $f(\rho_n)$ is a contraction map. The sufficient condition for the map to be contraction is to have derivative's norm bounded by $\gamma_d < 1$. First, we can write down the simplified formula $f(\rho_n)$:

$$f(\rho_n) = \frac{((1-t)\sqrt{q} + t\rho_n\sqrt{s})(t\sqrt{s} + (1-t)\rho_n\sqrt{q})}{(1-t)q + 2t(1-t)\rho_n\sqrt{sq} + t^2 s + t(1-t)\epsilon}. \tag{68}$$

Next, we simplify derivative $f'(\rho_n)$:

$$\sigma_{0,t} = (1-t) \cdot q + t \cdot \rho,$$
$$\sigma_{t,1} = t \cdot s + (1-t) \cdot \rho,$$
$$\sigma_{t,t} = (1-t)^2 \cdot q + 2(1-t)t \cdot \rho + t^2 \cdot s + t(1-t)\epsilon = (1-t) \cdot \sigma_{0,t} + t \cdot \sigma_{t,1} + t(1-t)\epsilon,$$
$$f'(\rho_n) = \frac{(1-t)\sigma_{0,t}}{\sigma_{t,t}} + \frac{t\sigma_{t,1}}{\sigma_{t,t}} - 2 \cdot \frac{t\sigma_{t,1} \cdot (1-t)\sigma_{0,t}}{\sigma_{t,t} \cdot \sigma_{t,t}}.$$

We define new variables $\hat{\sigma}_{0,t} \stackrel{def}{=} (1-t)\sigma_{0,t}, \hat{\Sigma}_{t,1} \stackrel{def}{=} t\sigma_{t,1}, \hat{\epsilon} = t(1-t)\epsilon$ and restate $f'$ as:

$$f' = \frac{\hat{\sigma}_{0,t}}{\hat{\sigma}_{0,t} + \hat{\sigma}_{1,t} + \hat{\epsilon}} + \frac{\hat{\sigma}_{1,t}}{\hat{\sigma}_{0,t} + \hat{\sigma}_{1,t} + \hat{\epsilon}} - \frac{2\hat{\sigma}_{0,t}\hat{\sigma}_{1,t}}{(\hat{\sigma}_{0,t} + \hat{\sigma}_{1,t} + \hat{\epsilon})^2} \tag{69}$$

$$= \frac{(\hat{\sigma}_{0,t} + \hat{\sigma}_{1,t})(\hat{\sigma}_{0,t} + \hat{\sigma}_{1,t} + \hat{\epsilon}) - 2\hat{\sigma}_{0,t}\hat{\sigma}_{1,t}}{(\hat{\sigma}_{0,t} + \hat{\sigma}_{1,t} + \hat{\epsilon})^2}$$

$$= \frac{\hat{\sigma}_{0,t}^2 + \hat{\sigma}_{1,t}^2 + (\hat{\sigma}_{0,t} + \hat{\sigma}_{1,t})\hat{\epsilon}}{(\hat{\sigma}_{0,t} + \hat{\sigma}_{1,t} + \hat{\epsilon})^2} \tag{70}$$

$$= \frac{\hat{\sigma}_{0,t}^2 + \hat{\sigma}_{1,t}^2 + (\hat{\sigma}_{0,t} + \hat{\sigma}_{1,t})\hat{\epsilon}}{\hat{\sigma}_{0,t}^2 + 2\hat{\sigma}_{0,t}\hat{\sigma}_{1,t} + \hat{\sigma}_{1,t}^2 + 2(\hat{\sigma}_{0,t} + \hat{\sigma}_{1,t})\hat{\epsilon} + \hat{\epsilon}^2} \tag{71}$$

$$= \frac{1}{1 + \frac{2\hat{\sigma}_{0,t}\hat{\sigma}_{1,t} + (\hat{\sigma}_{0,t} + \hat{\sigma}_{1,t})\hat{\epsilon} + \hat{\epsilon}^2}{\hat{\sigma}_{0,t}^2 + \hat{\sigma}_{1,t}^2 + (\hat{\sigma}_{0,t} + \hat{\sigma}_{1,t})\hat{\epsilon}}}. \tag{72}$$

We note that all terms in (70) are greater than $0$ and, thus, $f$ is monotone:

$$0 < f'(\rho_n), \quad \rho_n \in (-1,1). \tag{73}$$

In the negative segment $\rho_n \in (-1,0]$, the derivative norm $|f'|$ is greater than in the positive segment $[0,1)$, and value of the function is always larger than its argument. Thus, in negative segment, the distance until the fixed point $\rho_n^* > 0$ is decreasing faster, than in positive segment.

For $\rho_n \in [0,1)$, we can bound the fraction in denominator of (72) by taking its numerator's minimum at $\rho_n = 0$ and its denominator's maximum at $\rho_n = 1$, i.e,

$$0 < f' \leq \gamma_d(q,s,t) < 1,$$

$$\gamma_d(q,s,t) = \frac{1}{1 + \frac{t^2(1-t)^2 qs + t(1-t)(t^2 s + (1-t)^2 q)\epsilon + t^2(1-t)^2\epsilon^2}{(1-t)^2((1-t)q + t\sqrt{qs})^2 + t^2(ts + (1-t)\sqrt{qs})^2 + t(1-t)((1-t)\sqrt{q} + t\sqrt{s})^2\epsilon}}.$$

We note that $\gamma_d(q,s,t)$ is increasing function w.r.t. $q, s$.

If we put into the function $f$ argument $\rho_n^*$ corresponding to the $\epsilon$-EOT correlation, DIMF does not change it. Hence, $\rho_n^*$ is the fixed point of $f(\rho_n)$, and we have

$$|\tilde{\rho}_n - \rho_n^*| = |f(\rho_n) - f(\rho_n^*)| \leq \gamma_d(q,s,t)|\rho_n - \rho_n^*|.$$

$\square$

**Lemma D.10** ($\chi$ improvement after (D)IMF step). *Consider a 2-dimensional Gaussian distribution with marginals $\mathcal{N}(\eta, q)$ and $\mathcal{N}(\nu, s)$ and normalized correlation $\rho_n \in (-1, 1)$ between its components. After continuous IMF or DIMF with a single time point $t$, we obtain new correlation $\tilde{\rho}_n$, such that $|\tilde{\rho}_n - \rho_n^*| \leq \gamma |\rho_n - \rho_n^*|$ where $\rho_n^* = \Xi_n^{-1}(1/\epsilon, q, s)$ and $\gamma < 1$ is from (65) for IMF and from (66) for DIMF. We have bound in terms of $\chi = \Xi_n(\rho_n, q, s)$ and $\tilde{\chi} = \Xi_n(\tilde{\rho}_n, q, s)$:*

$$|\tilde{\chi} - 1/\epsilon| \quad \leq \quad l(\rho_n, \rho_n^*, \gamma) \cdot |\chi - 1/\epsilon|, \tag{74}$$

$$l(\rho_n, \rho_n^*, \gamma) \quad = \quad \left[ 1 - (1 - \gamma) \frac{(1 - \max\{\rho_n^*, |\rho_n|\}^2)^2}{1 + \max\{\rho_n^*, |\rho_n|\}^2} \right] < 1.$$

*Proof.* **Monotone.** The function $f(\rho_n)$ from (67) for continuous IMF and from (68) for DIMF is monotonously increasing on $(-1, 1)$. The monotone means that the value $\tilde{\rho}_n$ always remains from the same side from $\rho_n^*$:

$$\begin{cases} \rho_n > \rho_n^* \implies f(\rho_n) > \rho_n^*, \\ \rho_n \leq \rho_n^* \implies f(\rho_n) \leq \rho_n^*, \end{cases} \tag{75}$$

The same inequalities hold true for $\chi = \Xi_n(\rho_n, q, s), \tilde{\chi} = \Xi(\tilde{\rho}_n, q, s)$ and $\chi_* = 1/\epsilon$ as well: if $\chi < \chi_*$, then $\tilde{\chi} < \chi_*$ and vice versa, since $\Xi_n(\rho_n, q, s)$ is monotonously increasing w.r.t. $\rho_n$.

$\Xi_n$ **Properties.** In this proof, we omit arguments $q, s$ of $\Xi_n^{-1}(\chi, q, s)$ and $\Xi_n(\rho_n, q, s)$, because they do not change during IMF step. The second derivative of the function $\Xi_n(\rho_n)$ is

$$\frac{d^2 \Xi_n}{d\rho_n^2}(\rho_n) = \frac{2\rho_n(3 + \rho_n^2)}{\sqrt{sq}(1 - \rho_n^2)^3}.$$

Hence, we have $\frac{d^2\Xi}{d\rho_n^2}(\rho_n) \leq 0$ for $\rho_n \in (-1, 0]$ and $\frac{d^2\Xi}{d\rho_n^2}(\rho_n) \geq 0$ for $\rho_n \in [0, 1)$. It means that the function $\Xi_n(\rho_n)$ is concave on $(-1, 0]$ and convex on $[0, 1)$.

The function $\Xi_n(\rho_n)$ is monotonously increasing w.r.t. $\rho_n$, thus, decreasing of the radius $h \overset{\text{def}}{=} |\rho_n - \rho_n^*|$ around $\rho_n^*$ causes the decreasing of $|\chi - \chi_*|$ around $\chi_*$. We consider two cases: $\chi > \chi_*$ and $\chi < \chi_*$.

**Case** $\chi > \chi_*$. We have $\rho_n = \rho_n^* + h, \chi = \Xi_n(\rho_n^* + h) = \Xi_n(\rho_n)$ and $\Xi_n(\rho_n^* + \gamma h) \geq \tilde{\chi}$. We compare the difference using convexity on $[0, 1)$:

$$\chi - \tilde{\chi} \quad \geq \quad \Xi_n(\rho_n^* + h) - \Xi_n(\rho_n^* + \gamma h) \geq (\rho_n^* + h - (\rho_n^* + h\gamma)) \cdot \frac{d\Xi_n}{d\rho_n}(\rho_n^* + \gamma h)$$

$$= \quad (1 - \gamma)h \cdot \frac{d\Xi_n}{d\rho_n}(\rho_n^* + \gamma h).$$

Since the derivative of $\Xi_n$ is always positive, we continue the bound:

$$\Xi_n(\rho_n^* + h) - \Xi_n(\rho_n^* + \gamma h) \geq \min_{\rho_n' \in [\rho_n^*, \rho_n^* + h]} \left| \frac{d\Xi_n}{d\rho_n}(\rho_n') \right| (1 - \gamma)|\rho_n - \rho_n^*|.$$

Next, we use Lipschitz property of $\Xi_n$, i.e.,

$$|\chi - \chi_*| = |\Xi_n(\rho_n) - \Xi_n(\rho_n^*)| \leq \max_{\rho_n' \in [\rho_n^*, \rho_n^* + h]} \left| \frac{d\Xi_n}{d\rho_n}(\rho_n) \right| |\rho_n - \rho_n^*|,$$

and combine it with the previous bound

$$\chi - \tilde{\chi} \geq \Xi_n(\rho_n^* + h) - \Xi_n(\rho_n^* + \gamma h) \quad \geq \quad \frac{\min\limits_{\rho_n' \in [\rho_n^*, \rho_n]} \left| \frac{d\Xi_n}{d\rho_n}(\rho_n') \right|}{\max\limits_{\rho_n' \in [\rho_n^*, \rho_n]} \left| \frac{d\Xi_n}{d\rho_n}(\rho_n') \right|} (1 - \gamma)|\chi - \chi_*|.$$

**Case** $\chi < \chi_*$. We have $\rho_n = \rho_n^* - h, \chi = \Xi_n(\rho_n^* - h) = \Xi_n(\rho_n)$ and $\Xi_n(\rho_n^* - \gamma h) \leq \tilde{\chi}$. There are three subcases for $\chi, \tilde{\chi}$ positions around 0:

1. For positions $\chi_* > \tilde{\chi} \geq \Xi_n(\rho_n^* - \gamma h) > \chi \geq 0$, we use *convexity* of $\Xi_n$ on $[0, 1)$ and obtain

$$\tilde{\chi} - \chi \quad \geq \quad \Xi_n(\rho_n^* - \gamma h) - \Xi_n(\rho_n^* - h) \geq (1 - \gamma)h \cdot \frac{d\Xi_n}{d\rho_n}(\rho_n^* - h)$$

$$\geq \quad \min_{\rho'_n \in [\rho^*_n - h, \rho^*_n]} \left| \frac{d\Xi_n}{d\rho_n}(\rho'_n) \right| (1-\gamma)|\rho_n - \rho^*_n|.$$

2. For positions $\chi_* > 0 \geq \tilde{\chi} \geq \Xi_n(\rho^*_n - \gamma h) > \chi$ and $\chi_* \geq \tilde{\chi} \geq 0 \geq \Xi_n(\rho^*_n - \gamma h) > \chi$, we use *concavity* of $\Xi_n$ on $(-1, 0]$ and obtain

$$\tilde{\chi} - \chi \quad \geq \quad \Xi_n(\rho^*_n - \gamma h) - \Xi_n(\rho^*_n - h) \geq (1-\gamma)h \cdot \frac{d\Xi_n}{d\rho_n}(\rho^*_n - \gamma h)$$

$$\geq \quad \min_{\rho'_n \in [\rho^*_n - h, \rho^*_n]} \left| \frac{d\Xi_n}{d\rho_n}(\rho'_n) \right| (1-\gamma)|\rho_n - \rho^*_n|.$$

3. For positions $\chi_* > \tilde{\chi} \geq \Xi_n(\rho^*_n - \gamma h) > 0 > \chi$, we use *concavity* of $\Xi_n$ on $(-1, 0]$ and *convexity* of $\Xi_n$ on $[0, 1)$ and obtain

$$\tilde{\chi} - \chi \quad \geq \quad \Xi_n(\rho^*_n - \gamma h) - \Xi_n(\rho^*_n - h) = [\Xi_n(\rho^*_n - \gamma h) - \Xi_n(0)] + [\Xi_n(0) - \Xi_n(\rho^*_n - h)]$$

$$\geq \quad (\rho^*_n - \gamma h) \cdot \frac{d\Xi_n}{d\rho_n}(0) + (h - \rho^*_n) \cdot \frac{d\Xi_n}{d\rho_n}(0) = (1-\gamma)h \cdot \frac{d\Xi_n}{d\rho_n}(0)$$

$$\geq \quad \min_{\rho'_n \in [\rho^*_n - h, \rho^*_n]} \left| \frac{d\Xi_n}{d\rho_n}(\rho'_n) \right| (1-\gamma)|\rho_n - \rho^*_n|.$$

Overall, we make the bound

$$\tilde{\chi} - \chi \quad \geq \quad \min_{\rho'_n \in [\rho^*_n - h, \rho^*_n]} \left| \frac{d\Xi_n}{d\rho_n}(\rho'_n) \right| (1-\gamma)|\rho_n - \rho^*_n|$$

$$\geq \quad \frac{\min_{\rho'_n \in [\rho_n, \rho^*_n]} |\frac{d\Xi_n}{d\rho_n}(\rho'_n)|}{\max_{\rho'_n \in [\rho_n, \rho^*_n]} |\frac{d\Xi_n}{d\rho_n}(\rho'_n)|} (1-\gamma)|\chi - \chi_*|.$$

For the function $\Xi_n(\rho_n) = \frac{\rho_n}{\sqrt{sq(1-\rho_n^2)}}$, the centrally symmetrical derivative is

$$\frac{d\Xi_n}{d\rho_n}(\rho_n) \quad = \quad \frac{1 + \rho_n^2}{\sqrt{sq(1-\rho_n^2)^2}}.$$

The derivative $\frac{d\Xi_n}{d\rho_n}$ has its global minimum at $\rho_n = 0$. It grows as $\rho_n \to \pm 1$, hence, the maximum value is achieved at points which are farthest from 0:

$$\max_{\rho'_n \in [\rho^*_n, \rho_n]} \left| \frac{d\Xi_n}{d\rho_n}(\rho'_n) \right| \quad \leq \quad \frac{d\Xi_n}{d\rho_n}(\rho_n),$$

$$\max_{\rho'_n \in [\rho_n, \rho^*_n]} \left| \frac{d\Xi_n}{d\rho_n}(\rho'_n) \right| \quad \leq \quad \max\left\{ \frac{d\Xi_n}{d\rho_n}(\rho^*_n), \frac{d\Xi_n}{d\rho_n}(|\rho_n|) \right\},$$

$$\min_{\rho'_n \in [-1, +1]} \left| \frac{d\Xi_n}{d\rho_n}(\rho'_n) \right| \quad \geq \quad \frac{1}{\sqrt{sq}}.$$

Thus, we prove the bound

$$|\chi - \chi_*| - |\tilde{\chi} - \chi_*| = |\tilde{\chi} - \chi| \geq \frac{(1 - \max\{\rho^*_n, |\rho_n|\}^2)^2}{1 + \max\{\rho^*_n, |\rho_n|\}^2} (1-\gamma)|\chi - \chi_*|.$$

$$|\tilde{\chi} - \chi_*| \leq \left[ 1 - (1-\gamma) \frac{(1 - \max\{\rho^*_n, |\rho_n|\}^2)^2}{1 + \max\{\rho^*_n, |\rho_n|\}^2} \right] |\chi - \chi_*|.$$

$\square$

## D.6  PROOF OF IPMF CONVERGENCE THEOREM 3.2, $D = 1$

*Proof.* **Notations.** We introduce the notations for a $k$-th IPMF step in terms of scalars

$$\begin{pmatrix} \sigma_0 & \rho_k \\ \rho_k & s_k \end{pmatrix} \overset{IMF}{\Longrightarrow} \begin{pmatrix} \sigma_0 & \tilde{\rho}_k \\ \tilde{\rho}_k & s_k \end{pmatrix} \overset{IPF}{\Longrightarrow} \begin{pmatrix} q_k & \rho'_k \\ \rho'_k & \sigma_1 \end{pmatrix}$$

$$\overset{IMF}{\Longrightarrow} \begin{pmatrix} q_k & \hat{\rho}_k \\ \hat{\rho}_k & \sigma_1 \end{pmatrix} \overset{IPF}{\Longrightarrow} \begin{pmatrix} \sigma_0 & \rho_{k+1} \\ \rho_{k+1} & s_{k+1} \end{pmatrix},$$

and means

$$\begin{pmatrix} \mu_0 \\ \nu_k \end{pmatrix} \overset{IMF}{\Longrightarrow} \begin{pmatrix} \mu_0 \\ \nu_k \end{pmatrix} \overset{IPF}{\Longrightarrow} \begin{pmatrix} \eta_k \\ \mu_1 \end{pmatrix} \overset{IMF}{\Longrightarrow} \begin{pmatrix} \eta_k \\ \mu_1 \end{pmatrix} \overset{IPF}{\Longrightarrow} \begin{pmatrix} \mu_0 \\ \nu_{k+1} \end{pmatrix}.$$

We denote the variance of the 0-th marginal after the $k$-th IPMF step as $q_k$. For the first one, we have formula (35) $q_0 = \sigma_0 - \sigma_0 \tilde{\rho}_{n,0}^2 \left( 1 - \frac{\sigma_1}{s_0} \right)$, where $\tilde{\rho}_{n,0} = \rho_0 / \sqrt{\sigma_0 s_0}$ is the normalized correlation after the first IMF step. More explicitly, $\tilde{\rho}_{n,0} \overset{\text{def}}{=} f(\rho_{n,0})$, where $\tilde{\rho}_{n,0}$ is taken from (67) for continuous IMF and from (68) for DIMF. We denote optimality coefficients $\chi_k \overset{\text{def}}{=} \Xi_n\left(\frac{\rho_k}{\sqrt{\sigma_0 s_k}}, \sigma_0, s_k\right)$ and $\chi_* = 1/\epsilon$.

**Ranges.** We note that IMF step keeps $q_k, s_k, \eta_k, \nu_k$, while IPF keeps $\chi_k$. Due to contractive update equations for $\chi_k$ (75) and for $s_k$ (38), the parameters $s_k, \chi_k$ remain on the same side from $\sigma_1, \frac{1}{\epsilon}$, respectively. Namely, we have ranges for the variances $s_k \in [\min\{\sigma_1, s_0\}, \max\{\sigma_1, s_0\}] \overset{\text{def}}{=} [\sigma_1^{min}, \sigma_1^{max}], q_k \in [\min\{\sigma_0, q_0\}, \max\{\sigma_0, q_0\}] \overset{\text{def}}{=} [\sigma_0^{min}, \sigma_0^{max}]$ and parameters $\chi_k \in [\min\{\chi_*, |\chi_0|\}, \max\{\chi_*, |\chi_0|\}] \overset{\text{def}}{=} [\chi^{min}, \chi^{max}]$.

**Update bounds.** We use update bounds for $\chi$ (74) twice, for $s$ (38) and for $\nu$ (41), however, we need to limit above the coefficients $|\Xi_n^{-1}(\chi, q, s)|$ and $l(\Xi_n^{-1}(\chi, q, s), \Xi_n^{-1}(\chi_*, q, s), \gamma(q, s))$ over the considered ranges of the parameters $q \in [\sigma_0^{min}, \sigma_0^{max}], s \in [\sigma_1^{min}, \sigma_1^{max}]$ and $\chi \in [\chi^{min}, \chi^{max}]$. The functions $\Xi_n^{-1}, l, \gamma$ are defined in (64), (74), (65) (or (66) with fixed $t$), respectively.

Since the function $|\Xi_n^{-1}(\chi, q, s)|$ is increasing w.r.t. $q, s'$ and $\chi$ (growing symmetrically around 0 for $\chi$), we take maximal values $\sigma_0^{max}, \sigma_1^{max}$ and $\chi^{max}$. Similarly, the function $l(\Xi_n^{-1}(\chi, q, s), \Xi_n^{-1}(\chi_*, q, s), \gamma(q, s))$ is increasing w.r.t. all arguments symmetrically around 0. Hence, we maximize the function $|\Xi_n^{-1}|$ and the function $\gamma$, which is also increasing w.r.t. $q$ and $s$.

**Final bounds.** The final bound after $k$ step of IPMF are:

$$\begin{aligned} |s_k^2 - \sigma_1^2| &\leq \alpha^{2k} |s_0^2 - \sigma_1^2|, \\ |\nu_k - \mu_1| &\leq \alpha^k |\nu_0 - \mu_1|, \\ |\chi_k - 1/\epsilon| &\leq \beta^{2k} |\chi_0 - 1/\epsilon|, \end{aligned}$$

where $\beta \overset{\text{def}}{=} l(\Xi_n^{-1}(\chi^{max}, \sigma_0^{max}, \sigma_1^{max}), \Xi_n^{-1}(\chi_*, \sigma_0^{max}, \sigma_1^{max}), \gamma(\sigma_0^{max}, \sigma_1^{max}))$ and $\alpha \overset{\text{def}}{=} \Xi_n^{-1}(\chi^{max}, \sigma_0^{max}, \sigma_1^{max})$ taking $l$ from (74), $\gamma$ from (65) for continuous IMF and from (66) with fixed $t$ for discrete IMF. □

## D.7  PROOF OF IPMF GENERAL CONVERGENCE THEOREM 3.3

*Proof.* We split the proof into two parts. First, consider the discrete case.

**Discrete case.** Let $k \geq 1$. Note that the transition probabilities $q^{4k+1}(x_{t_1} | x_0)$ can be bounded from below with $\alpha \mu(x_{t_1})$, where $\alpha \in (0, 1)$ and $\mu$ depend only on $t_1$, $\epsilon$ and supports of $p_0$ and $p_1$. Thus, we can bound $q^{4k+1}(x_1 | x_0) \geq \alpha \mu'(x_1)$, with $\mu'(x_1)$ depending on $q_{0,1}^{4k+1}$,

$$proj_{\mathcal{M}}[q^{4k+1}](x_1 | x_0) = \int proj_{\mathcal{M}}[q^{4k+1}](x_1 | x_{t_1}) q(x_{t_1} | x_0) dx_{t_1}$$

$$\geq \alpha \int proj_{\mathcal{M}}[q](x_1 | x_{t_1}) d\mu(x_{t_1}) =: \alpha \mu'(x_1). \tag{76}$$

Similar statement holds for $q_{0,1}^{4k+3}$. Thus, the IPMF step is contracting. Specifically,

$$\|q_0^{4k+2} - p_0\|_{TV} \le (1-\alpha)\|q_1^{4k} - p_1\|_{TV},$$
$$\|q_1^{4k+4} - p_1\|_{TV} \le (1-\alpha)\|q_0^{4k+2} - p_0\|_{TV\cdot},$$

where TV denotes Total Variation distance. Thus,

$$q_0^{4k} \overset{TV}{\to} p_0, \quad q_1^{4k+2} \overset{TV}{\to} p_1. \tag{77}$$

Since $p_0$ and $p_1$ have compact supports, Prokhorov's theorem ensures the existence of a weakly converging subsequence $q_{0,1}^{4k_j} \overset{w}{\to} \tilde{q}_{0,1}$. Moreover, (77) ensures that $\tilde{q}_{0,1} \in \Pi(p_0, p_1)$.

Let IMF[$q$] be the result of the IMF-step applied to $q$, and let IPMF[$q$] be the result of IPMF-step applied to $q$. Note that the IMF step is continuous w.r.t. weak convergence, since all intermediate steps have smooth transition (i.e., conditional) densities. Combining the above results, we get that

$$\text{IPMF}[q_{0,1}^{4k_j}] \overset{w}{\to}_j \text{IPMF}[\tilde{q}_{0,1}] = \text{IMF}[\text{IMF}[\tilde{q}_{0,1}]] \tag{78}$$

The equality holds due to $\tilde{q}_{0,1} \in \Pi(p_0, p_1)$. Note that we also use the fact that convergence in TV is stronger than weak convergence.

Recall that $q_{0,1}^{4k_j+4} = \text{IPMF}[q_{0,1}^{4k_j}]$. (78) ensures that for any fixed $n > 0$ it holds $q_{0,1}^{4k_j+4n} \overset{w}{\to}_j$ IMF$^{2n}[\tilde{q}_{0,1}]$. Moreover, by Theorem 3.6 in [ASBM], it holds that $IMF^{2n}[\tilde{q}_{0,1}] \overset{w}{\to}_n q_{0,1}^*$.

Thus, there exists a weakly converging subsequence

$$q_{0,1}^{4l_i} \overset{w}{\to} q_{0,1}^*. \tag{79}$$

Finally, we argue by contradiction: if $q_{0,1}^{4k} \overset{w}{\nrightarrow} q_{0,1}^*$, we can select a weakly converging subsequence $q_{0,1}^{4l_i} \overset{w}{\to} q_{0,1}' \ne q_{0,1}^*$. But by (79) $q_{0,1}' = q_{0,1}^*$. This finishes the proof.

**Continuous case.** We now explain how to extend the above argument to the continuous-time setting. The key point is to verify a Doeblin minorization condition for the Markovian process obtained after the projection step (see, e.g., Section 2 in Stroock (2005)).

Fix some $\delta \in (0, 1/2)$. For each $k \in \mathbb{N}$, let $(X_t^{4k+2})_{t \in [0,1]}$ denote the Markov diffusion corresponding to the law $q^{4k+2}$, and let

$$P_k(x, A) := \mathbb{P}(X_1^{4k+2} \in A \mid X_0^{4k+2} = x), \qquad x \in \mathbb{R}^D, \ A \in \mathcal{B}(\mathbb{R}^D),$$

be its transition kernel from time 0 to time 1. This admits a decomposition of the evolution on $[0,1]$ into three subintervals $[0, \delta]$, $[\delta, 1-\delta]$ and $[1-\delta, 1]$.

**Disspaptivity.** Next, we show that, by construction of the Markovian projection, the drift on $[\delta, 1-\delta]$ is Lipschitz and dissipative. Set

$$b^+(t, x) := \mathbb{E}\left[\frac{X_1 - x}{1-t} \mid X_t = x\right] = \frac{m_t(x) - x}{1-t}, \qquad m_t(x) := \mathbb{E}[X_1 \mid X_t = x],$$

and recall that since the supports of $p_0(x)$ and $p_1(x)$ are bounded, $\|X_1\| \le M_1$ and $\|m_t(x)\| \le M_1$ for any $x$. Then

$$\langle b^+(t, x), x \rangle = \frac{\langle m_t(x), x \rangle - \|x\|^2}{1-t} \le \frac{M_1\|x\| - \|x\|^2}{1-t} \le -\frac{1}{2(1-t)}\|x\|^2 + \frac{M_1^2}{2(1-t)}.$$

This yields uniform coercivity (Lyapunov-type dissipativity) for any $t \in [\delta, 1-\delta]$.

**The drift is Lipshitz.** Consider

$$\nabla_x b^+(t, x) = \frac{\nabla_x m_t(x) - I}{1-t},$$

and recall that the transition density is

$$\pi_x(x_1) := q_{t,1}(x_1 \mid x) \propto g(x_1) \exp\left(-\frac{\|x_1 - x\|^2}{2\varepsilon^2(1-t)}\right).$$

Consequently,

$$\nabla_x m_t(x) = \nabla_x \mathbb{E}[X_1 | X_t = x] = \nabla_x \int x_1 \pi_x(x_1) d\,x_1 = \int x_1 \nabla_x \pi_x(x_1)\, dx_1$$

$$= \int (x_1 \times \nabla_x \log \pi_x(x_1)) \pi_x(x_1)\, dx_1 = \frac{1}{2\varepsilon^2(1-t)} \mathrm{Cov}_{\pi_x}[X_1].$$

Thus,

$$\|\nabla_x m_t(x)\| \leq C \frac{\mathrm{diam}^2(\mathrm{supp}\, p_1)}{\varepsilon^2(1-t)}.$$

This ensures

$$\|\nabla_x b^+(t,x)\| \leq \frac{1}{1-t} + C \frac{\mathrm{diam}(\mathrm{supp}\, p_1)^2}{\varepsilon^2(1-t)^2}.$$

A similar result holds for the backward drift

$$b^-(t,x) := \mathbb{E}\left[\frac{x - X_0}{t} \,\Big|\, X_t = x\right] = \frac{x - m'_t(x)}{t}, \qquad m'_t(x) := \mathbb{E}[X_0 \mid X_t = x].$$

**Small-set condition on $[\delta, 1-\delta]$.** Fix $R > 0$ and consider the transition from time $\delta$ to $1-\delta$. Denote by $p_k(x,y)$ the transition density of $X_{1-\delta}^{4k+2}$ given $X_\delta^{4k+2} = x$. Since the diffusion coefficient is constant non-degenerate ($\varepsilon^2 I_D$) and the drift is locally Lipschitz (with a uniform bound on $\|\nabla b^+(\cdot)\|$ on $[\delta, 1-\delta]$), heat-kernel lower bounds for uniformly elliptic diffusions yield that $p_k$ is continuous and strictly positive, and admits a lower bound on compact sets with constants independent of $k$; see, e.g., Theorem 7 in Aronson (1968).

In particular, for any fixed $r > 0$ there exists $m_{R,r} > 0$ such that

$$\inf_{k \in \mathbb{N}} \inf_{x \in B_R,\, y \in B_r} p_k(x,y) \geq m_{R,r} > 0.$$

Let $\nu_r$ be the uniform probability measure on $B_r$ and set $\beta_R := m_{R,r} \mathrm{Leb}(B_r)$. Then for all $k \in \mathbb{N}$, all $x \in B_R$, and all measurable $A$,

$$\mathbb{P}\big(X_{1-\delta}^{4k+2} \in A \,\big|\, X_\delta^{4k+2} = x\big) \geq \beta_R\, \nu_r(A) \tag{80}$$

.

By the Markov property, for any measurable $A$,

$$\mathbb{P}\big(X_1^{4k+2} \in A \,\big|\, X_\delta^{4k+2} = x\big) = \int \mathbb{P}\big(X_1^{4k+2} \in A \,\big|\, X_{1-\delta}^{4k+2} = z\big) \mathbb{P}\big(X_{1-\delta}^{4k+2} \in dz \,\big|\, X_\delta^{4k+2} = x\big).$$

Combining this with (80), we obtain

$$\mathbb{P}\big(X_1^{4k+2} \in A \,\big|\, X_\delta^{4k+2} = x\big) \geq \beta_R \mu_{r,k}(A), \qquad x \in B_R,$$

where $\mu_{r,k}$ is the probability measure defined by

$$\mu_{r,k}(A) := \int \mathbb{P}\big(X_1^{4k+2} \in A \,\big|\, X_{1-\delta}^{4k+2} = z\big) \nu_r(dz). \tag{81}$$

Next, we control the tails of distribution of $X_\delta^{4k+2}$ uniformly in $k$. By the definition of the reciprocal projection, the segment $[0,1]$ between $X_0$ and $X_1$ is (conditionally on $(X_0, X_1)$) distributed as a Brownian bridge with variance parameter $\sigma^2 = \varepsilon^2 \delta(1-\delta)$. Hence, the marginal at time $\delta$ is a mixture of Gaussian laws with covariance matrix $\sigma^2 I_D$ and mean

$$m_\delta(x_0, x_1) = (1-\delta)x_0 + \delta x_1,$$

where $(x_0, x_1)$ ranges over the support of the endpoint coupling. Since the supports of $p_0$ and $p_1$ are bounded, there exists $R_0 > 0$ such that $\|m_\delta(x_0, x_1)\| \leq R_0$ for all $(x_0, x_1)$ in this support. Standard Gaussian tail bounds then imply that, for any $\eta \in (0, 1)$, we can choose $R > 0$ large enough so that

$$\sup_{k \in \mathbb{N}} \sup_{x \in \mathrm{supp}(p_0)} \mathbb{P}\big(X_\delta^{4k+2} \notin B_R \,\big|\, X_0^{4k+2} = x\big) \;\leq\; \eta.$$

Equivalently,

$$\mathbb{P}\big(X_\delta^{4k+2} \in B_R \,\big|\, X_0^{4k+2} = x\big) \;\geq\; 1 - \eta, \qquad x \in \mathrm{supp}(p_0), \; k \in \mathbb{N}. \tag{82}$$

Combining (81) and (82), we obtain, for $x \in \mathrm{supp}(p_0)$ and any measurable $A \subset \mathbb{R}^D$,

$$\begin{aligned}
P_k(x, A) &= \mathbb{E}\Big[\mathbb{P}\big(X_1^{4k+2} \in A \,\big|\, X_\delta^{4k+2}\big) \,\Big|\, X_0^{4k+2} = x\Big] \\
&\geq \mathbb{E}\Big[\mathbb{P}\big(X_1^{4k+2} \in A \,\big|\, X_\delta^{4k+2}\big)\mathbf{1}_{\{X_\delta^{4k+2} \in B_R\}} \,\Big|\, X_0^{4k+2} = x\Big] \\
&\geq \beta_R \nu_R(A)\, \mathbb{P}\big(X_\delta^{4k+2} \in B_R \,\big|\, X_0^{4k+2} = x\big) \\
&\geq \beta_R(1 - \eta)\, \mu_{r,k}(A).
\end{aligned}$$

Thus, for all $x \in \mathrm{supp}(p_0)$ and all $k \in \mathbb{N}$,

$$P_k(x, \cdot) \;\geq\; \alpha\, \mu_{r,k}(\cdot) \quad \text{with} \quad \alpha := \beta_R(1 - \eta) \in (0, 1), \;\; \mu := \nu_R.$$

That is, the family of kernels $(P_k)_k$ satisfies a uniform Doeblin minorization on $\mathrm{supp}(p_0)$.

It is well known that such a minorization implies total-variation contraction: for any probability measures $\lambda, \lambda'$ on $\mathrm{supp}(p_0)$,

$$\|\lambda P_k - \lambda' P_k\|_{\mathrm{TV}} \;\leq\; (1 - \alpha)\, \|\lambda - \lambda'\|_{\mathrm{TV}}, \qquad k \in \mathbb{N}.$$

Applying this with $\lambda = q_0^{4k+1}$ and $\lambda' = p_0$ yields

$$\|q_1^{4k+1} - p_1\|_{\mathrm{TV}} \;=\; \|q_0^{4k+1} P_k - p_0 P_k\|_{\mathrm{TV}} \;\leq\; (1 - \alpha)\, \|q_0^{4k+1} - p_0\|_{\mathrm{TV}}.$$

The convergence $q_0^{4k+1} \to_{\mathrm{TV}} p_0$ is shown by the same argument applied backward in time (interchanging the roles of $p_0$ and $p_1$), and we conclude that

$$q_1^{4k+1} \xrightarrow[k \to \infty]{\mathrm{TV}} p_1.$$

In particular, the continuous-time analogue of (77) holds.

Having established the convergence of the marginals via the Doeblin minorization, it remains to ensure the well-posedness and stability of the iterative procedure. Next, we recall that the drift $b^\pm(t)$ is locally Lipschitz and dissipative.

Next, by the symmetry of the IMF procedure, it suffices to consider the transitions to the midpoint, $(0, 1/2)$ and $(1, 1/2)$. By standard diffusion theory, the regularity of the drift ensures that the corresponding Markovian transition kernels are smooth. Consequently, because the updated joint distribution $q^{i+1}(x_0, x_{1/2}, x_1)$ is constructed by integrating these transition kernels against the previous distribution $q^i$, the mapping $q^i \mapsto q^{i+1}$ preserves weak convergence. Therefore, the IMF iteration is continuous with respect to the weak topology. The rest of the proof is similar to the discrete case. $\qquad\square$

## E    EXPERIMENTAL SUPPLEMENTARY

### E.1    ILLUSTRATIVE $2D$ EXAMPLE VISUALIZATION.

We provide the visualization of the starting processes and corresponding learned processes for *Gaussian→Swiss roll* translation in Fig. 7. One can visually observe that all the particle trajectories or relatively straight and therefore close to the Schrödinger Bridge problem solution.

Table 5: Datasets and code used in our work along with their licenses.

| Name | URL | Citation | License |
|------|-----|----------|---------|
| Colored MNIST | GitHub Link | Gushchin et al. (2023b) | MIT |
| CelebA | Dataset Link | Liu et al. (2015a) | Non-commercial research only |
| SB Benchmark | GitHub Link | Gushchin et al. (2023b) | MIT |
| ASBM Code | GitHub Link | Gushchin et al. (2024b) | MIT |
| DSBM Code | GitHub Link | Shi et al. (2023) | MIT |

## E.2 SB BENCHMARK $B\mathbb{W}_2^2$-UVP

We additionally study how well implementations of IPMF procedure starting from different starting processes map initial distribution $p_0$ into $p_1$ by measuring the metric $B\mathbb{W}_2^2$-UVP also proposed by the authors of the benchmark (Gushchin et al., 2023b). We present the results in Table 6. One can observe that DSBM initialized from different starting processes has quite close results and so is the case for ASBM experiments with $\epsilon \in \{1, 10\}$, but with $\epsilon = 0.1$ one can notice that ASBM starting from IPF and *Identity* experience a decline in $B\mathbb{W}_2^2$-UVP metric.

| | | $\epsilon = 0.1$ | | | | $\epsilon = 1$ | | | | $\epsilon = 10$ | | | |
|---|---|---|---|---|---|---|---|---|---|---|---|---|---|
| | Algorithm Type | $D=2$ | $D=16$ | $D=64$ | $D=128$ | $D=2$ | $D=16$ | $D=64$ | $D=128$ | $D=2$ | $D=16$ | $D=64$ | $D=128$ |
| Best algorithm on benchmark[†] | Varies | 0.016 | 0.05 | 0.25 | 0.22 | 0.005 | 0.09 | 0.56 | 0.12 | 0.01 | 0.02 | 0.15 | 0.23 |
| DSBM-IMF | | 0.1 | 0.14 | 0.44 | 3.2 | 0.13 | **0.1** | **0.91** | 6.67 | 0.1 | 5.17 | 66.7 | 356 |
| DSBM-IPF | | 0.35 | 0.6 | 0.6 | 1.62 | **0.01** | 0.18 | **0.91** | 6.64 | 0.2 | 3.78 | 81 | 206 |
| DSBM-*Identity* | | 0.13 | 0.64 | 2.67 | 7.12 | 0.1 | 0.12 | 2 | 6.67 | 0.02 | 3.8 | 86.4 | 343 |
| ASBM-IMF[†] | IPMF | **0.016** | **0.1** | 0.85 | 11.05 | 0.02 | 0.34 | 1.57 | **3.8** | 0.013 | 0.25 | 1.7 | 4.7 |
| ASBM-IPF | | 0.05 | 0.73 | 32.05 | 10.67 | 0.02 | 0.53 | 4.19 | 10.11 | **0.002** | **0.18** | 2.2 | 5.08 |
| ASBM-*Identity* | | 0.12 | 2.65 | 4.59 | 40.3 | 0.04 | 0.45 | 2.02 | 4.76 | 0.03 | 0.2 | **1.43** | **2.71** |
| SF$^2$M-Sink[†] | Bridge Matching | 0.04 | 0.18 | **0.39** | **1.1** | 0.07 | 0.3 | 4.5 | 17.7 | 0.17 | 4.7 | 316 | 812 |

Table 6: Comparisons of $B\mathbb{W}_2^2$-UVP $\downarrow$ (%) between the ground truth static SB solution $p^T(x_0, x_1)$ and the learned solution on the SB benchmark. The best metric over is **bolded**. Results marked with † are taken from (Gushchin et al., 2024b) or (Gushchin et al., 2023b).

## E.3 CELEBA SDEDIT STARTING PROCESSES DESCRIPTION

The IPMF framework does not require the starting process to have $p_0, p_1$ marginals or to be a Schrödinger bridge. One can then try other starting processes that would improve the practical performance of the IPMF algorithm. Properties of the starting process that would be desirable are (1) $q(x_0) = p_0(x_0)$ and marginal $q(x_1)$ to be close to $p_1(x_1)$ and (2) $q(x_0, x_1)$ to be close to SB. In the IMF or IPF, we had to choose one of these properties because we can not easily satisfy them both.

We propose to take a basic image-to-image translation method and use it as a coupling to induce a starting process for the IPMF procedure. Such a coupling could provide the two properties mentioned above. We use SDEdit (Meng et al., 2022) which requires an already trained diffusion model (SDE prior). Given an input image $x$, SDEdit first adds noise to the input and then denoises the resulting image by the SDE prior to make it closer to the target distribution of the SDE prior. Various models can be used as an SDE prior. We explore two options: trainable and train-free. As the first option, we train the DDPM (Ho et al., 2020) model on the CelebA 64×64 size female only part. As the second option we take an already trained Stable Diffusion (SD) V1.5 model (Rombach et al., 2022) with text prompts conditioned on which model generates 512×512 images similar to the CelebA female part. We then apply SDEdit with the CelebA male images as input to produce similar female images using trainable DDPM and train-free SDv1.5 approaches, we call the starting processes generated by these SDEdit induced couplings DDPM-SDEdit and SD-SDEdit. Hyperparameters of SDEdit, DDPM and SDv1.5 are provided in Appendix E.9.

The visualization of the DSBM and ASBM implementations of the IPMF procedure starting from *DDPM-SDEdit* and SD-SDEdit processes is in Figure 4.

## E.4 CELEBA EXPERIMENT ADDITIONAL QUANTITATIVE STUDY

In Table 8, we report the final CMMD (Jayasumana et al., 2024) values for IPMF, while Figure 9 illustrates how this metric evolves over IPMF iterations. Both evaluations are performed on the

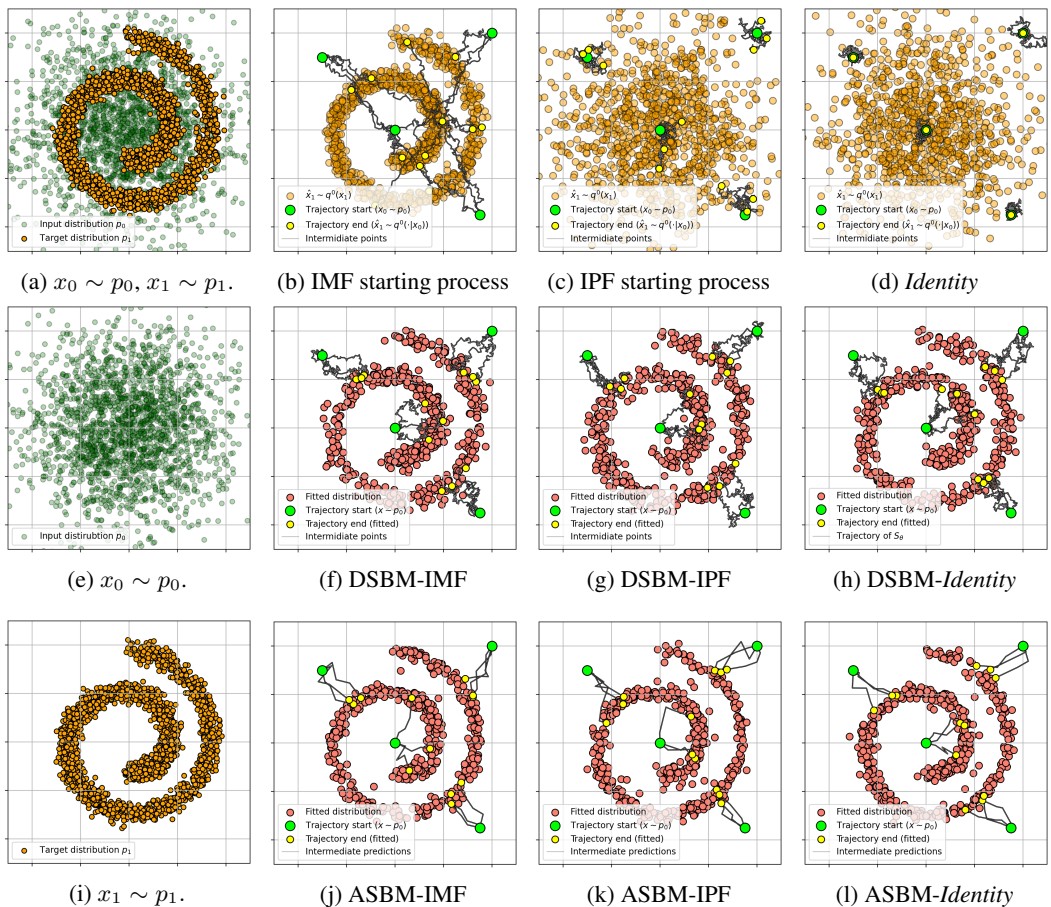

Figure 7: Visualization of learned processes with DSBM and ASBM solvers for *Gaussian→Swiss roll* translation using IMF, IPF, *Identity* starting processes for $\epsilon = 0.1$.

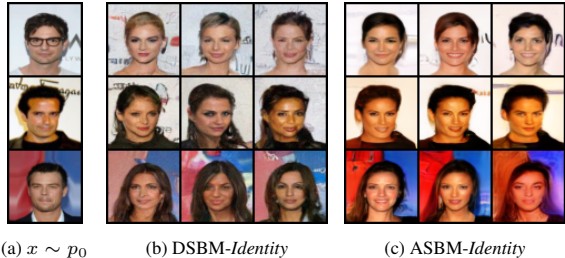

(a) $x \sim p_0$     (b) DSBM-*Identity*     (c) ASBM-*Identity*

Figure 8: Results on the CelebA dataset for the *male → female* translation task, where $x_0 \sim p_0$ represents samples from the source distribution. DSBM-*Identity* and ASBM-*Identity* refers to the outputs generated using trained DSBM/ASBM with the *Identity* initialization. The model was trained with $\epsilon = 10$.

same test set as in §4.4. Notably, the resulting CMMD curve closely mirrors the behavior observed for FID in Figure 5. Additionally, Figure 8 and Table 7 present results obtained using DSBM and ASBM with the *Identity* initialization process on the CelebA dataset, with $\epsilon = 10$.

## E.5 GENERAL EXPERIMENTAL DETAILS

Authors of ASBM (Gushchin et al., 2024b) kindly provided us the code for all the experiments. All the hyperparameters including neural networks architectures were chosen as close as possible to the ones used by the authors of ASBM in their experimental section. Particularly, as it is described in

| | Initialisation (coupling) | | | | DSBM | | | | | ASBM | | | | |
|---|---|---|---|---|---|---|---|---|---|---|---|---|---|---|
| | IMF | DDPM SDEdit | SD SDEdit | Identity | IMF | DDPM SDEdit | SD SDEdit | Identity | Identity $\epsilon=10$ | IMF | DDPM SDEdit | SD SDEdit | Identity | Identity $\epsilon=10$ |
| FID↓ | 0.0 | 35.23 | 28.77 | 61.56 | **13.65** | 14.84 | 22.65 | 33.11 | 65.50 | **19.32** | 21.84 | 20.64 | 19.58 | 27.47 |
| MSE$(x_0, \widehat{x}_1)$↓ | 0.16 | 0.02 | 0.02 | 0.0 | 0.16 | 0.09 | 0.04 | **0.03** | 0.16 | 0.17 | **0.07** | 0.08 | 0.07 | 0.11 |

Table 7: Extended for $\epsilon = 10$ qualitative results on CelebA ($64 \times 64$) for *male→female* translation with ASBM and DSBM across different starting processes. Generative quality (FID↓) and similarity (MSE$(x_0, \widehat{x}_1)$↓) are reported on the test set. Best and second-best values for solvers are marked in **bold** and underline, respectively.

| | Initialisation (coupling) | | | | DSBM | | | | ASBM | | | |
|---|---|---|---|---|---|---|---|---|---|---|---|---|
| | IMF | DDPM SDEdit | SD SDEdit | Identity | IMF | DDPM SDEdit | SD SDEdit | Identity | IMF | DDPM SDEdit | SD SDEdit | Identity |
| CMMD↓ | 0.0 | 0.31 | 0.69 | 0.84 | **0.32** | 0.46 | 0.34 | 0.33 | **0.28** | 0.42 | 0.32 | 0.51 |

Table 8: Qualitative results on CelebA ($64 \times 64$) for *male→female* translation with ASBM and DSBM across different starting processes. Generative quality (CMMD↓) is reported on the test set. Best and second-best values for solvers are marked in **bold** and underline, respectively.

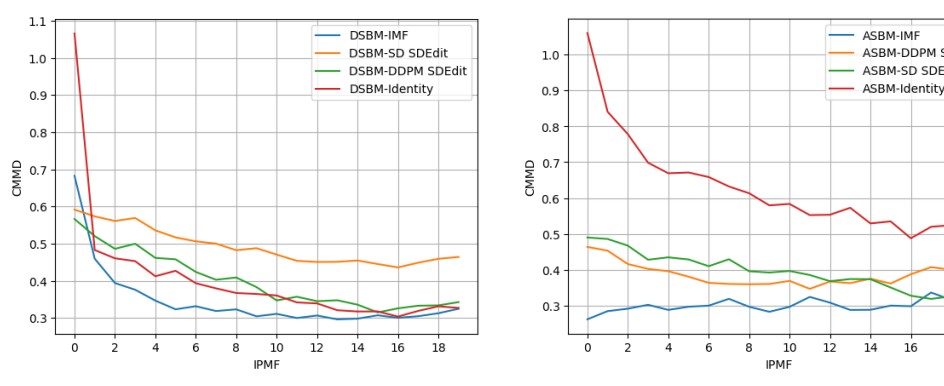

(a) CMMD for DSBM with various couplings.  (b) CMMD for ASBM with various couplings.

Figure 9: CMMD metric in CelebA *male→female* ($64 \times 64$) as a function of IPMF iteration for various starting couplings.

(Gushchin et al., 2024b, Appendix D), authors used DD-GAN (Xiao et al.) with Brownian Bridge posterior sampling instead of DDPM's one and implementation from:

```
https://github.com/NVlabs/denoising-diffusion-gan
```

DSBM (Shi et al., 2023) implementation is taken from the official code repository:

```
https://github.com/yuyang-shi/dsbm-pytorch
```

Sampling on the inference stage is done by Euler Maryama SDE numerical solver (Kloeden, 1992) with indicated in Table 9 NFE.

The Exponential Moving Average (EMA) has been used to enhance generator's training stability of both ASBM and DSBM. The parameters of the EMA are provided in Table 9, in case the EMA decay is set to "N/A" no averaging has been applied.

### E.6 ILLUSTRATIVE 2D EXAMPLES DETAILS

**ASBM**. For toy experiments the MLP with hidden layers $[256, 256, 256]$ has been chosen for both discriminator and generator. The generator takes vector of $(dim+1+2)$ length with data, latent variable and embedding (a simple lookup table `torch.nn.Embedding`) dimensions, respectively. The networks have `torch.nn.LeakyReLU` as activation layer with $0.2$ angle of negative slope. The optimization has been conducted using `torch.optim.Adam` with running averages coefficients $0.5$ and $0.9$. Additionally, the `CosineAnnealingLR` scheduler has been used only at

| Model | Dataset | Start process | IPMF iters | IPMF-0 Grad Updates | IPMF-k Grad Updates |
|-------|---------|---------------|-----------|---------------------|---------------------|
| ASBM | CelebA | All | 20 | 200,000 | 20,000 |
| DSBM | CelebA | All | 20 | 100,000 | 20,000 |
| ASBM | Swiss Roll | All | 20 | 400,000 | 40,000 |
| DSBM | Swiss Roll | All | 20 | 20,000 | 20,000 |
| ASBM | cMNIST | All | 20 | 75,000 | 38,000 |
| DSBM | cMNIST | All | 20 | 100,000 | 20,000 |
| ASBM | SB Bench | All | 20 | 133,000 | 67,000 |
| DSBM | SB Bench | All | 20 | 20,000 | 20,000 |

| Model | Dataset | Start process | NFE | EMA decay | Batch size | D/G opt ratio | Lr G | Lr D |
|-------|---------|---------------|-----|-----------|-----------|---------------|------|------|
| ASBM | CelebA | All | 4 | 0.999 | 32 | 1:1 | 1.6e-4 | 1.25e-4 |
| DSBM | CelebA | All | 100 | 0.999 | 64 | N/A | 1e-4 | N/A |
| ASBM | Swiss Roll | All | 4 | 0.999 | 512 | 1:1 | 1e-4 | 1e-4 |
| DSBM | Swiss Roll | All | 100 | N/A | 128 | N/A | 1e-4 | N/A |
| ASBM | cMNIST | All | 4 | 0.999 | 64 | 2:1 | 1.6e-4 | 1.25e-4 |
| DSBM | cMNIST | All | 30 | 0.999 | 128 | N/A | 1e-4 | N/A |
| ASBM | SB Bench | All | 32 | 0.999 | 128 | 3:1 | 1e-4 | 1e-4 |
| DSBM | SB Bench | All | 100 | N/A | 128 | N/A | 1e-4 | N/A |

Table 9: Hyperparameters of models from CelebA (§4.4), SwissRoll (§4.2), cMNIST (§4.4) and Benchmark (§4.3) experiments. In "Start process", the column "All" states for all the used options. "N/A" corresponds to either not used or not applicable, the corresponding option.

pretraining iteration with minimal learning rate set to 1e-5 and no restarting. To stabilize GAN training R1 regularizer with coefficient 0.01 (Mescheder et al., 2018) has been used.

**DSBM**. MLP with $[\dim + 12, 128, 128, 128, 128, 128, \dim]$ number of hidden neurons, `torch.nn.SiLU` activation functions, residual connections between 2nd/4th and 4th/6th layers and Sinusoidal Positional Embedding has been used.

### E.7 SB BENCHMARK DETAILS

Scrödinger Bridges/Entropic Optimal Transport Benchmark (Gushchin et al., 2023b) and $cBW_2^2$-UVP, $BW_2^2$-UVP metric implementation was taken from the official code repository:

$$\text{https://github.com/ngushchin/EntropicOTBenchmark}$$

Conditional plan metric $cBW_2^2$-UVP , see Table 1, was calculated over predefined test set and conditional expectation per each test set sample estimated via Monte Carlo integration with 1000 samples. Target distribution fitting metric, $BW_2^2$-UVP, see Table 6, was estimated using Monte Carlo method and 10000 samples.

**ASBM**. The same architecture and optimizer have been used as in toy experiments E.6, but without the scheduler.

**DSBM**. MLP with $[\dim + 12, 128, 128, 128, 128, 128, \dim]$ number of hidden neurons, `torch.nn.SiLU` activation functions, residual connections between 2nd/4th and 4th/6th layers and Sinusoidal Positional Embedding has been used.

### E.8 CMNIST DETAILS

Working with the MNIST dataset, we use a regular train/test split with 60000 images and 10000 images respectively. We RGB color train and test digits of classes "2" and "3". Each sample is resized to $32 \times 32$ and normalized by 0.5 mean and 0.5 std. **ASBM**. The cMNIST setup mainly differs by the architecture used. The generator model is built upon the NCSN++ architecture (Song et al.), following the approach in (Xiao et al.) and (Gushchin et al., 2024b). We use 2 residual and attention blocks, 128 base channels, and $(1, 2, 2, 2)$ feature multiplications per corresponding resolution level. The dimension of the latent vector has been set to 100. Following the best practices of time-dependent neural networks sinusoidal embeddings are employed to condition on the integer

time steps, with a dimensionality equal to $2\times$ the number of initial channel, resulting in a 256-dimensional embedding. The discriminator adopts ResNet-like architecture with 4 resolution levels. The same optimizer with the same parameters as in toy E.6 and SB benchmark E.7 experiments have been used except ones that are presented in Table 9. No scheduler has been applied. Additionally, R1 regularization is applied to the discriminator with a coefficient of 0.02, in line with (Xiao et al.) and (Gushchin et al., 2024b).

**DSBM**. The model is based on the U-Net architecture (Ronneberger et al., 2015) with attention blocks, 2 residual blocks per level, 4 attention heads, 128 base channels, $(1, 2, 2, 2)$ feature multiplications per resolution level. Training was held by Adam (Kingma & Ba, 2014) optimizer.

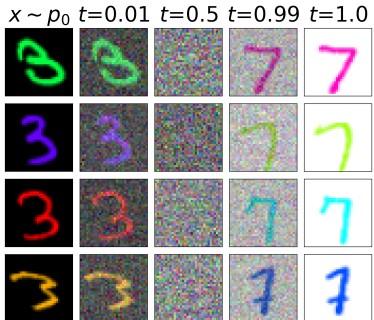

Figure 10: *Inverted 7* starting process, i.e., process in the reciprocal class with marginals $p_0$ and $p^{\text{inv7}}$, visualization.

### E.9 CELEBA DETAILS

Test FID, see Figure 5 is calculated using pytorch-fid package, test CMMD is calculated using unofficial implementation in PyTorch. Working with CelabA dataset (Liu et al., 2015b), we use all 84434 male and 118165 female samples ($90\%$ train, $10\%$ test of each class). Each sample is resized to $64 \times 64$ and normalized by $0.5$ mean and $0.5$ std.

**ASBM.** As in cMNIST experiments E.8 the generator model is built upon the NCSN++ architecture (Song et al.) but with small parameter changes. The number of initial channels has been lowered to 64, but the number of resolution levels has been increased with the following changes in feature multiplication, which were set to $(1, 1, 2, 2, 4)$. The discriminator also has been upgraded by growing the number of resolution levels up to 6. No other changes were proposed.

**DSBM**. Following Colored MNIST translation experiment exactly the same neural network and optimizer was used.

**SDEdit coupling**. DDPM (Ho et al., 2020) was trained on CelebA female train part processed in the same way as for other CelebA experiments. Number of diffusion steps is equal to 1000 with linear $\beta_t$ noise schedule, number of training steps is equal to 1M, UNet (Ronneberger et al., 2015) was used as neural network with 78M parameters, EMA was used during training with rate 0.9999. The DDPM code was taken from the official DDIM (Song et al., 2021) github repository:

https://github.com/ermongroup/ddim

The SDEdit method (Meng et al., 2022) for DDPM model was used with 400 steps of noising and 400 steps of denoising. The code for SDEdit method was taken from the official github repository:

https://github.com/ermongroup/SDEdit

The Stable Diffusion V1.5 (Rombach et al., 2022) model was taken from the Huggingface (Wolf et al., 2020) model hub with the tag *"runwayml/stable-diffusion-v1-5"*. The text prompt used is *"A female celebrity from CelebA"*. The SDEdit method implementation for the SDv1.5 model was taken from the Huggingface library (Wolf et al., 2020), i.e. *"StableDiffusionImg2ImgPipeline"*, with hyperparameters: *strength* 0.75, *guidance scale* 7.5, *number of inference steps* 50. The output of SDEdit pipeline has been downscaled from $512\times512$ size to $64\times64$ size using bicubic interpolation.

### E.10 AFHQ DETAILS

We first pretrain the networks using Bridge Matching for 100000 steps, then run DSBM for 20 iterations with 25000 steps per outer iteration. We follow (Shi et al., 2023) and use the same U-Net architecture. The batch size is 4, and the EMA rate is 0.999. We choose $\sigma^2 = 5$, and again we use 100 sampling steps with constant stepsizes.

### E.11 COMPUTATIONAL RESOURCES

The experiment on CelebA for each of the starting processes takes approximately 5 days and 7 days on Nvidia A100 for DSBM and ASBM, respectively. Experiments with Colored MNIST take less than 2 days of training on an A100 GPU for ASBM or DSBM, and for each starting process. Illustrative 2D examples and Schrödinger Bridge benchmark experiments take several hours on GPU A100 each for ASBM or DSBM and for each starting process.

