# OpenReview forum: "Diffusion & Adversarial Schrödinger Bridges via Iterative Proportional Markovian Fitting"
_ICLR.cc/2026/Conference — ICLR 2026 Poster_

### Official Review · Reviewer_N3B2 · 2025-10-28

**Soundness:** 3
**Presentation:** 3
**Contribution:** 3
**Rating:** 6
**Confidence:** 3

**Summary:**

This paper shows that a heuristic bidirectional modification of the Iterative Markovian Fitting (IMF) procedure such as Diffusion Schrodinger Bridge Matching or Adversarial Schrodinger Bridge Matching is also a valid algorithm for solving the Schrodinger Bridge Problem (SBP). The authors prove that this heuristic modification of IMF, dubbed Iterative Proportional Markovian Fitting (IPMF), converges exponentially to the solution when the marginals are Gaussian. IPMF is also shown to converge to the solution under a weaker condition, when the marginals have bounded supports. In contrast to IPF or IMF, IPMF can be initialized with any starting process. The authors show the benefits of flexible initialization with experiments on Gaussians, SB benchmarks, unpaired image-to-image translation, etc.

**Strengths:**

- **[S1] This paper is significant in the aspect that it bridges a gap between theory and practice.** Specifically, the paper provides several proofs regarding the convergence of a heuristic modification of the IMF procedure, while proofs in previous works [1,2] show the convergence of the original IMF procedure.

- **[S2] This paper is original in the aspect that it provides a novel insight into better training of SBs.** The authors also show that unlike IMF or IPF, IPMF admits any initial starting process. Hence, one may potentially achieve faster training of SBs via IPMF by using a well-designed starting process.

[1] Diffusion Schrödinger Bridge Matching

[2] Adversarial Schrödinger Bridge Matching

**Weaknesses:**

- **[W1] Experimental results only weakly support the benefits of using arbitrary starting processes.** Experiments in Section 4 do not really show the strength of using arbitrary starting processes in terms of both scale and performance. In terms of scale, the only non-trivial task is male to female on CelebA-64. In terms of performance, non-IMF initializations such as SD SDEdit, DDPM SDEdit, or Identity all suffer from FID degradation at the cost of smaller MSE between $x_0$ and $\widehat{x}_1$. Previous works on SB such as DSBM or ASBM demonstrate consistent performance improvements on more difficult data with resolution $\geq 128$, so it would be nice to see results on similar data in this paper as well.

**Questions:**

- **[Q1] In Table 2, can the authors provide FIDs for initial processes as well?** I believe this FID is necessary in order to judge whether there are performance gains after running IPMF on the given initial process such as SDEdit or Identity.

- **[Q2] In Table 2, can the authors provide LPIPS between $x_0$ and $\widehat{x}_1$?**

- **[Q3] Can the authors provide results with larger $\epsilon$ when starting with the identity process?** I am curious whether this improves FID by increasing diversity in the outputs.

- **[Q4] Do observations made on CelebA 64x64 scale to higher resolution images?**

---

> ### Author Response · Authors · 2025-11-23
>
> Dear Reviewer N3B2, thank you for your comments. Here are the answers to your questions and comments.
>
> **[W.1-1] Experiments in Section 4 do not really show the strength of using arbitrary starting processes in terms of both scale and performance. Non-IMF initializations such as SD SDEdit, DDPM SDEdit, or Identity all suffer from FID degradation at the cost of smaller MSE between $x_0$ and $\hat{x}_1$.**
>
> Dear reviewer, thank you for this remark.
> First, we address the question concerning the performance (**[W.1-2]** handles the scale matter).
> Indeed, you are right: the performance of some initializations gain in MSE while demonstrating the FID degradation. However, this is fully consistent with our main practical goal, which is to demonstrate how one can balance quality and input-output similarity by choosing different initializations in the IPMF framework (**Table 2** illustrates the idea). We rephrase this clearly in a revised version (see lines **91-93**) and believe that novel way of tuning SB-based models could be particularly useful for practitioners.
>
> **[W.1-2] In terms of scale, the only non-trivial task is male to female on CelebA-64. ... Previous works on SB such as DSBM or ASBM demonstrate consistent performance improvements on more difficult data with resolution $\geq 128$, so it would be nice to see results on similar data in this paper as well.**
>
> The study of higher resolutions mainly requires access to substantial GPU resources, both to train larger models and to perform more extensive hyperparameter tuning. Moreover, as you mentioned, the scalability of the DSBM and ASBM solvers has already been investigated in the original papers [DSBM, ASBM]. Given our strict GPU resource constraints, we chose instead to focus on conducting experiments on a diverse set of initializations, rather than to an additional scalability study.
>
> **[Q.1] In Table 2, can the authors provide FIDs for initial processes as well? I believe this FID is necessary in order to judge whether there are performance gains after running IPMF on the given initial process such as SDEdit or Identity.**
>
> Thank you for your suggestion. As per request, we added these results to **Table 2**. Note that the results follow *The general principle* is: the better the initial coupling performs with respect to a given property, the better the final performance in that property.
>
> **[Q.2] In Table 2, can the authors provide LPIPS between $x_0$ and $\hat{x}_1$?**
>
> Thank you for the suggestion. We agree that reporting LPIPS between $x_0$ and $\hat{x}_1$ would provide additional insight. We are currently computing these values and will include them in the updated version of the paper and further rebuttal answers.
>
> **[Q.3] Can the authors provide results with larger eps when starting with the identity process? I am curious whether this improves FID by increasing diversity in the outputs.**
>
> Thank you for the question. We appreciate your interest in understanding the effect of using a larger $\epsilon$ when starting from the identity process. We provide the results of this experiment in **Figure 7** and **Table 6** of **Appendix E.4**. We observe that the FID metric ($65.5$) became worse as well as the quality of images, due to the remaining noise in them. Also, we observe that the SE metric ($0.16$) is the same as the IMF coupling, thus providing significantly fewer images than in the case of $\epsilon=1.0$, which is in agreement with the theory.
>
> **[Q.4] Do observations made on CelebA 64x64 scale to higher resolution images?**
>
> Please see the answer for the weakness **[W.1]**.
>
> **Concluding remarks**. We would be grateful if you could let us know if our explanations have been satisfactory. If so, we kindly ask that you consider increasing your rating. We are also open to discussing any other questions you may have.

---

> ### Comment · Reviewer_N3B2 · 2025-11-26
>
> Thank you for the detailed reply! However, I have remaining concerns regarding the practicality of IPMF, and hope the authors can address them.
>
> **Regarding the authors' reply to [W1] -- weak experimental results.** Yes, the scalability of the DSBM and ASBM solvers has already been investigated in the original papers [DSBM, ASBM], but not with arbitrary couplings. Results on higher resolution data showing that those solvers work even with arbitrary couplings would substantially strengthen this paper. If pixel-space experiments are difficult, one could also consider latent domain experiments. For instance, see [1].
>
> [1] Generalized Schrödinger Bridge Matching, ICLR, 2024.
>
> **Regarding the authors' reply to [Q3] -- results with larger $\epsilon$.** I believe one of the largest benefits of SB-based unpaired image-to-image translation compared to conventional GAN-based approaches (e.g., CycleGAN, CUT, etc.), is its ability to trade-off diversity vs. fidelity by controlling $\epsilon$. However, in Table 7 of the revised manuscript, I see that IPMF with $\epsilon=10$ performs worse even than the initial identity coupling. This to me feels like a major limitation of the practicality of IPMF.

---

> ### Author Response · Authors · 2025-12-03
>
> **New 512 x 512 AFHQ image results.** While the scalability of the DSBM and ASBM solvers for the Markovian projection part has already been investigated in the original papers [DSBM, ASBM], per the request to run experiments with higher image resolution, we provide results for the AFHQ [1] image dataset with $512 \times 512$ resolution images. Following [DSBM] paper we consider classes cat and wild. We run DSBM with IMF-OT and Identity couplings and present the results in Figure 6 and Table 3. **We observe a similar quality-similarity tradeoff as the Celeba setup**, i.e., the higher the similarity or generation quality of the coupling, the better the model performs on the corresponding metric.
>
> **Regarding results with larger epsilon.** DSBM may be less stable for larger coefficient $\epsilon$ so we also provide the results of ASBM also for $\epsilon=10$ in Appendix E.4, which is far more stable and provides better FID ($27$) and MSE metric ($0.11$). And these results also agree with the theory, since those for $\epsilon=10$ are more diverse than those for $\epsilon=1$.

---

### Official Review · Reviewer_aDs4 · 2025-10-31

**Soundness:** 3
**Presentation:** 3
**Contribution:** 2
**Rating:** 4
**Confidence:** 5

**Summary:**

Schr\"odinger Bridge techniques have been widely used to address unpaired data-to-data translation tasks. This paper focuses on a practical heuristic implementation of the Iterative Markovian Fitting (IMF) procedure that has been proposed in the literature because it improves empirical performance. The central observation in this paper is that the ``bidirectional'' implementations of IMF used in practice (namely Diffusion Schr\"odinger Bridge Matching (DSBM) and Adversarial Schr\"odinger Bridge Matching (ASBM) implicitly interleave both IMF and Iterative Proportional Fitting (IPF) updates. The authors formalize this approach as IPMF and analyze its convergence properties. Conceptually, IPMF alternates (1) reciprocal/Markovian projections (IMF), with (2) projections that enforce boundary distributions (IPF).

In the Gaussian case, the paper introduces an optimality matrix $A(q)$ for couplings and shows that any two-variable Gaussian coupling with D-dimensional marginals plan is the entropic Optimal Transport (OT) solution for a suitable bilinear cost $c_A(x_0,x_1)=-x_1^\top A x_0$ (Theorem 3.1). It then proves exponential convergence (Theorem 3.2;) for various cases : $D=1$ (discrete/continuous, any $\varepsilon>0$) and $D>1$ (discrete time, sufficiently large $\varepsilon$)} of IPMF to the SB (1D in continuous/discrete time; $D>1$ in discrete time under a large-$\varepsilon$ regime). Beyond the Gaussian settings, it establishes weak convergence under boundness assumptions (Theorem 3.3).

Finally, the paper provides numerical experiments: convergence on high-dimensional Gaussians, a 2D toy problem, quantitative results on the SB benchmark (Table~1) , and qualitative unpaired image-to-image translation on Colored MNIST and CelebA. Beyond unification, the paper shows that selecting different initial couplings yields a controllable trade-off between input-output similarity and generative fidelity.

**Strengths:**

Clarity: The paper is well-written and the arguments are presented with excellent clarity. Theoretical concepts are well-motivated and well illustrate; e.g. Figure1 illustrates neatly how IPF and IMF projections combine within the IPMF framework.

Originality: As clearly acknowledged by the authors, the proposed method is not new but was presented as some heuristic implementation of IMF in the literature. The originality of the paper is to formalize it as an explicit technique alternating IMF and IPF.  This is significant as it connects two separate approaches in the literature, i.e. Sinkhorn/IPF and IMF/flow matching-type ideas, into a unified procedure. In particular, it helps understanding why bidirectionality helps mitigate the "prior forgetting" behaviour of the diffusion implementation of  IPF alone) and error accumulation of the naive implementation of IMF alone (which motivated the introduction of the bidirectional scheme). This will be of interest to people working on unpaired data-to-data translation.

Quality: Beyond formalizing the bidirectional method as a rigorous combination of IPF/IMF, the paper presents some interesting theoretical results. In particular, it provides (to the best of my knowledge) the first theoretical analysis of the bidirectional variant used in practice. While limited in scope, the analysis of the Gaussian case is neat and rigorous. The empirical section is fairly complete, including high-dimensional Gaussians, a 2D toy example, the SB benchmark, and real images (Colored MNIST, CelebA) . It also includes convergence diagnostics (KL forward/reverse etc.).
The literature in this domain is plagued with over-the-top claims, this paper does not make any such claim and remains very factual which I really appreciate.

Significance and Practical Impact: The experiments convincingly illustrate that  different couplings induce a principled, tunable trade-off between input-output similarity and generative fidelity. This is useful for practitioners working on unpaired translation.

**Weaknesses:**

Theory results: They remain limited. Exponential convergence is only proved in the Gaussian case for (a) $D=1$ continuous/discrete settings and (b) $D>1$ in discrete time under a fairly restrictive large-$\varepsilon$ condition. For applications, we care about $D$ large and non-Gaussian distributions. In this case, exponential convergence is only conjectured.  The authors should try to clarify the practical implications of the large-$\varepsilon$ assumption. It  would be helpful to quantify how large $\varepsilon$ must be for the proof to hold in typical image dimensions, and whether this aligns with the values used in practical DSBM implementations.

Continuous-Time IPMF: The continuous-time version of IPMF is given in (22)-(23) but theoretical guarantees for that case are not provided beyond $D=1$ for the Gaussian case. The authors should discuss the specific challenges in extending the contraction arguments to the continuous-time and high-dimensional setting.

Benchmarking & Ablations: On the SB benchmark, performance is sometimes on par with prior matching solvers and sometimes worse (e.g., DSBM variants at higher $D$/$\varepsilon$ show large $\mathrm{cBW}^2_2$ errors). The paper argues that different initializations converge to similar outcomes within each solver, but stronger head-to-head comparisons would make the empirical case more compelling.

**Questions:**

Tightness and Practicality of the Large-$\varepsilon$, Can you quantify how large $\varepsilon$ needs to be in the discrete $D>1$ proof for typical image resolutions (i.e., dimension $D$), and how this compares to $\varepsilon$ used in DSBM practice?

Continuous-Time Guarantees. Is there any way you could extendthe Gaussian contraction argument to continuous time beyond $D=1$ (e.g., by controlling the Markovian projection via stability of SDE discretizations)?

Number of IPMF Rounds and Stopping Criteria. How sensitive are results to the number of  IPMF iterations? Can you think of a reliable practical stopping criterion?  A small experiment (quality/similarity vs.\ rounds) on the CelebA dataset would be good.

Trade-off via Starting Couplings. The Identity and SDEdit couplings improve similarity, sometimes at the expense of FID. Could you relate the similarity/quality outcome to properties of the initial coupling (e.g. entropy or cost)?

Relation to Rectified Flows. It is correctly suggested that bidirectional IPMF could mitigate error accumulation in rectified flow. Could you provide some results on a small dataset (e.g. CIFAR10) showing that IPMF (with \epsilon=0) stabilizes Rectified Flow training? I conjecture using \epsilon decreasing to zero across iterations might even work better.

Positioning. Given the novelty relative to prior bidirectional heuristics (DSBM/ASBM), I think the paper would benefit from a short table in the appendix clarifying the  precise technical advances of IPMF (e.g., convergence rates, new unifying formulation, starting coupling analysis) over these related works.

---

> ### Author Response · Authors · 2025-11-23
>
> Dear Reviewer aDs4, thank you for your comments. Please find below the answers to your questions and comments.
>
> **[W.1-1] Exponential convergence is only proved in the Gaussian case for (a) D=1 continuous/discrete settings and (b) D > 1 in discrete time under a fairly restrictive large-eps condition. For applications, we care about D large and non-Gaussian distributions. In this case, exponential convergence is only conjectured.**
>
> Indeed, the exponential convergence in the general case is only conjectured. However, this issue is deeply rooted in the necessity of developing a genuinely novel proof approach/technique. Specifically, as we wrote in **lines 158 and 179**, both the IPF and IMF steps minimize their respective objective functions via iterative projections. In contrast, to the best of our knowledge, IPMF cannot be cast as the minimization of a single objective function, setting it apart from classical schemes that guarantee a decrease of an objective functional.
>
> In **our genuinely novel proof**, we have noticed that an IPF iteration does not change the "copula", i.e., the information about the joint distribution that is invariant w.r.t. changes in marginals $p_0$ and $p_1$. In contrast, an IMF iteration changes the copula but preserves the marginals. The main difficulty is to find a representation of the copulas' representation that is convenient for the convergence analysis. This question is left as a direction for future research.
>
> From the practical point of view, the results of our image experiments support the conjecture for large $D$ and non-Gaussian distributions.
>
> **[W.1-2] The authors should try to clarify the practical implications of the large-eps assumption. It would be helpful to quantify how large eps must be for the proof to hold in typical image dimensions, and whether this aligns with the values used in practical DSBM implementations.**
>
> The large-$\epsilon$ assumption in our Gaussian analysis is tied to the explicit representation of the IMF step for the marginals, whose supports are unbounded. In practice, however, most real-world data distributions (e.g., images, audio, video) are effectively supported on bounded domains. As a result, **the sufficient bound on $\epsilon$ produced by our Gaussian proof should not be interpreted as a quantitatively meaningful prescription for realistic image models.**
>
> Whether a condition of a similar flavor is actually needed (and, if so, in what quantitative form) in a proof of exponential convergence under bounded support assumptions (e.g., for images, audio, video) requires a separate, careful analysis, which we leave for future work (see our answer to **[W.1-1]**).
>
> We would like to emphasize that the main practically relevant contribution of this study is establishing the general convergence of the IPMF procedure for bounded-support distributions. Please note that, following the reviewers' request, we extended the results to the continuous case (see answer to **[Q.1] of Reviewer hGUD**).
>
> **[W.2] The authors should discuss the specific challenges in extending the contraction arguments to the continuous-time and high-dimensional setting... Is there any way you could extend the Gaussian contraction argument to continuous time beyond $D=1$?**
>
> (1) General high-dimensional case.
> Following the reviewers’ request, we provide the proof of general convergence of the continuous-time IPMF (see answer to **[Q.1] of Reviewer hGUD**).
> The exponential convergence in the continuous-time and high-dimensional settings is a matter for further research (see our response to your **[W.1-1]**).
>
> (2) Gaussian case. Unfortunately, in the higher-dimensional continuous case, the analytical formula for the Gaussian parameters after the IMF step is unknown. This is the main obstacle preventing us from providing proof. However, this obstacle is mostly related to the original IMF rather than our IPMF.
>
> **[W.3] Benchmarking \& Ablations: On the SB benchmark, performance is sometimes on par with prior matching solvers and sometimes worse. The paper argues that different initializations converge to similar outcomes within each solver, but stronger head-to-head comparisons would make the empirical case more compelling.**
>
> Thank you for this comment.
> You mentioned that ``stronger head-to-head comparisons'' are necessary for the SB benchmark. Could you please elaborate on which additional metrics or benchmark conditions you would advise us to use to strengthen the empirical argument? We wish to highlight that the comprehensive comparison is already present in **Table 1**, which covers all available values of $(D,\varepsilon)$ for DSBM and ASBM solvers.

---

> > ### Author Response · Authors · 2025-11-23
> >
> > **[Q.1] Number of IPMF Rounds and Stopping Criteria. How sensitive are results to the number of IPMF iterations? Can you think of a reliable practical stopping criterion? A small experiment (quality/similarity vs. rounds) on the CelebA dataset would be good.**
> >
> > Initially, we presented this result in **Figure 7** of **Appendix E.4**. The plot shows that different couplings (see our response to your comment **[Q.3]** for details) can lead to faster convergence with respect to specific properties, such as generative quality or input–output similarity. Following your request, we have moved this figure to the main part of the paper (now **Figure 5**).
> >
> > For more details on practical benefits, please refer to the answer to **[W.3] of Reviewer hGUD.**
> >
> > **[Q.2] Trade-off via Starting Couplings. The Identity and SDEdit couplings improve similarity, sometimes at the expense of FID. Could you relate the similarity/quality outcome to properties of the initial coupling (e.g. entropy or cost)?**
> >
> > Yes, you are completely right that the Identity and SDEdit couplings improve similarity at the cost of generation quality (FID), and thereby illustrate how different couplings provide a way to trade off between generation quality and similarity. *The general principle* is: the better performance of the initial coupling with respect to a given property, the better the final performance in that property. This principle is supported by the convergence plot (**Figure 5**). We further support this claim by providing coupling metrics (FID and MSE) alongside final metrics in **Table 2** of the main text in the revised version.
> >
> > **[Q.4] Relation to Rectified Flows. Could you provide some results on a small dataset (e.g. CIFAR10) showing that IPMF (with $\epsilon=0$) stabilizes Rectified Flow training?**
> >
> > Thank you very much for this insightful suggestion. We fully agree that studying the interaction between bidirectional IPMF (with $\epsilon$=0) and Rectified Flow is an interesting and promising direction. We are currently running these experiments within our CelebA setup, and we plan to report the corresponding results as soon as they are ready in the following rebuttal answers.
> >
> > **[Q.5] Given the novelty relative to prior bidirectional heuristics (DSBM/ASBM), I think the paper would benefit from a short table in the appendix clarifying the precise technical advances of IPMF over these related works.**
> >
> > We appreciate your suggestion and have added a summary in **Table 3** in the **Appendix A** with discussion.
> >
> > **Concluding remarks**. We would be grateful if you could let us know if our explanations have been satisfactory. If so, we kindly ask that you consider increasing your rating. We are also open to discussing any other questions you may have.

---

> > > ### Comment · Reviewer_aDs4 · 2025-11-28
> > > **--**
> > >
> > > Thanks for your detailed rebuttal. I will increase my score to 6.

---

### Official Review · Reviewer_6f73 · 2025-10-31

**Soundness:** 3
**Presentation:** 2
**Contribution:** 2
**Rating:** 4
**Confidence:** 4

**Summary:**

This paper proposes a new Schrodinger Bridge training method for combining IMF and IPF.

**Strengths:**

It combines the two processes of IMF and IPF, with alternating method.
It theoretically combined the IPF and IMF with thorough analysis and also proposed to apply proposed method to both of DSBM and ASBM.

**Weaknesses:**

1. Although it proved convergence in gaussian and bounded support, it does not contain any generalization or guarantee on other general distrubutions.

2. To apply this proposed method, it requires alternating process for IMF and IPF. It takes heavier computation burden compared to previous processes. Although the paper proposes faster convergence, it does not contain any comparison between previous methods in terms of computation efficiency and time complexity.

3. The performance heavily rely on initial hyperparameter setting such as initial coupling choice.

4. The most important and critiral issues is the practical usage of proposed method. Although the theoretical analysis and proposed methods is quite new, but the shown experiments are only limited on simple dataset such as toy data , colored MNIST and CelebA. Also the used dataset has relatively low resolution. To prove the practical generalizability of proposed method, the paper must include experiments on more dataset with higher resolution.

**Questions:**

See weakness

---

> ### Author Response · Authors · 2025-11-23
>
> Dear Reviewer 6f73, thank you for your comments. We think there might be some misunderstanding. The aim of our study lies in **the theoretical and practical analysis** of the popular heuristic bidirectional IMF (IPMF) rather than proposing a novel method. Taking this into account, please find below the answers to your questions and comments.
>
> **[W.1] Although it proved convergence in gaussian and bounded support, it does not contain any generalization or guarantee on other general distrubutions.**
>
> Our theoretical analysis focuses on a common and practically relevant setting, namely that *typical data distributions have bounded support* (e.g., images, audio, and video). Importantly, both our analyses of the Gaussian case and the bounded-support case require genuinely novel proof techniques. Extending these results to more general, non-bounded-support distributions is mathematically challenging (for the discussion, please refer to the answer to **[W.1-1 and W.2]** of **Reviewer aDs4**). We regard this as an interesting direction for future work.
>
> **[W.2] To apply this proposed method, it requires alternating process for IMF and IPF. It takes heavier computation burden compared to previous processes. Although the paper proposes faster convergence, it does not contain any comparison between previous methods in terms of computation efficiency and time complexity.**
>
> We would like to emphasize that IPMF is the well-known heuristic bidirectional IMF and is state-of-the-art in the SB literature [DSBM, ASBM]. Our analysis shows that IPMF can be decomposed into a combination of IPF and IMF steps; however, this decomposition is used solely for theoretical analysis and does not alter the computational procedure (see **lines 342–346**). Furthermore, our experimental analysis compares the convergence behavior of IPMF under different couplings (including those used in prior work: IMF and IPF, as well as newly introduced couplings in our paper); please see the answer below for details.
>
> **[W.3] The performance heavily rely on initial hyperparameter setting such as initial coupling choice.**
>
> Dear Reviewer, thank you for this remark. We do not consider this observation a weakness; on the contrary, we regard it as one of the central features of our framework. As stated in **lines 92–93**, this dependence is a practical feature of IPMF: it enables tuning the trade-off between generation quality and similarity of the produced images to the inputs. For more details, please refer to the answer to **[W.3]** of **Reviewer hGUD**.
>
> **[W.4] The most important and critiral issues is the practical usage of proposed method. To prove the practical generalizability of proposed method, the paper must include experiments on more dataset with higher resolution.**
>
> Please refer to the answer to **[W.1-2] of Reviewer N3B2.**
>
> **Concluding remarks**. We would be grateful if you could let us know if our explanations have been satisfactory. If so, we kindly ask that you consider increasing your rating. We are also open to discussing any other questions you may have.

---

> ### Author Response · Authors · 2025-12-03
>
> **New 512 x 512 AFHQ image results.** While the scalability of the DSBM and ASBM solvers for the Markovian projection part has already been investigated in the original papers [DSBM, ASBM], per the request to run experiments with higher image resolution, we provide results for the AFHQ [1] image dataset with $512 \times 512$ resolution images. Following [DSBM] paper we consider classes cat and wild. We run DSBM with IMF-OT and Identity couplings and present the results in Figure 6 and Table 3. **We observe a similar quality-similarity tradeoff as the Celeba setup**, i.e., the higher the similarity or generation quality of the coupling, the better the model performs on the corresponding metric.

---

### Official Review · Reviewer_hGUD · 2025-11-03

**Soundness:** 3
**Presentation:** 3
**Contribution:** 2
**Rating:** 4
**Confidence:** 4

**Summary:**

The paper introduces **Iterative Proportional Markovian Fitting (IPMF)**, a unified framework connecting **Iterative Proportional Fitting (IPF)** and **Iterative Moment Fitting (IMF)** in stochastic bridge learning.
The authors show that practical bidirectional IMF procedures implicitly perform IPF-like proportional updates, and formalize this equivalence into IPMF with convergence guarantees in Gaussian settings.
Experiments on synthetic and low-resolution image domains demonstrate stable coupling and controllable trade-offs, situating existing DSBM and ASBM methods as special cases.

**Strengths:**

- Provides a **theoretical unification** between IPF and IMF with clear mathematical derivation.
- Clarifies the conceptual foundation of recent bridge-based diffusion methods.
- Demonstrates improved stability under diverse initial couplings.

**Weaknesses:**

*Limited novelty:** IPMF mainly reinterprets existing IMF practices under an IPF perspective; lacks a genuinely new algorithm.
- **Narrow empirical scope:** Evaluations are confined to Gaussian and 64×64 image settings without large-scale or continuous-time experiments.
- **Unclear practical benefit:** No evidence of faster convergence or lower compute cost compared to IMF or DSBM.

## Minor
- Heavy notation reduces accessibility.
- Missing comparison with recent OT/consistency bridge baselines.

**Questions:**

1. Beyond theoretical unification, does IPMF yield measurable training or sampling improvements over IMF or DSBM?
2. Is the convergence guarantee valid for non-Gaussian or continuous-time bridges?
3. How sensitive is IPMF to the choice of starting coupling (e.g., SDEdit vs. identity)?
4. Can IPMF incorporate stochastic regularization (e.g., adversarial or entropy terms) without breaking convergence?
5. Are there cases where separate IPF or IMF updates remain preferable in practice?

---

> ### Author Response · Authors · 2025-11-23
>
> Dear Reviewer hGUD, thank you for your comments. Please find below the answers to your questions and comments.
>
> **[W.1] Limited novelty: IPMF mainly reinterprets existing IMF practices under an IPF perspective; lacks a genuinely new algorithm.**
>
> We kindly remind you that the goal of this study is a theoretical analysis of heuristic bidirectional IMF (IPMF). Thus, we do not aim to introduce a novel algorithm.
>
> To the best of our knowledge, all our analysis is completely novel (for details, see the answer to **[W.1] of Reviewer aDs4**). Previous works [ASBM, DSBM] only focus on investigating some theoretical properties of **non-bidirectional IMF** and the **special case of bidirectional IMF** with IPF starting coupling [DSBM].
>
> In contrast to prior works, our study demonstrates convergence with **any** starting coupling, which allows practitioners to better tailor the method for specific needs.
>
> To highlight the points mentioned above, we summarize them in a new **Table 3.**
>
> **[W.2] Narrow empirical scope: Evaluations are confined to Gaussian and 64×64 image settings without large-scale or continuous-time experiments.**
>
> First, we would like to emphasize that we do include the experiments with the continuous-time setting. In particular, we use DSBM as a continuous-time solver for the SB problem in image experiments (**Section 4.4**), SB benchmark experiments (**Section 4.3**), and illustrative 2D experiments (**Section 4.2**).
>
> Concerning scalability, please refer to the answer to **[W.1-2]** of **Reviewer N3B2**.
>
> **[W.3] Unclear practical benefit. Beyond theoretical unification, does IPMF yield measurable training or sampling improvements over IMF or DSBM? How sensitive is IPMF to the choice of starting coupling (e.g., SDEdit vs. identity)?**
>
> We answer to each issue mentioned in this weakness.
>
> **Plots with the convergence analysis.**
> Initially, we provided plots of generation quality metrics (FID and CMMD) and input-output similarity (MSE between input and output) over IPMF iterations for various couplings, including SDEdit and the identity coupling, in **Figure 7** of **Appendix E.4**. We have now moved this figure to the main text (**Figure 5**) to better showcase this behavior, taking advantage of the additional page permitted in the rebuttal revision and following your advice to explain the practical benefits more clearly. We summarize the results presented in **Figure 5** related to your questions as follows:
>
> **Improvements over IMF DSBM coupling.** According to **Figure 5** of the revised version, the IMF coupling has better performance in the quality of the generation FID metric, but produces significantly less similar images. In turn, identity or SDEdit couplings provide a way to balance performance in these properties.
>
> **Comparison of SDEdit and Identity couplings.** According to **Figure 5** of the revised version, identity coupling starts from the lower MSE, but higher FID compared to the SDEdit coupling. Eventually, during the iteration, two couplings start to show similar performance in MSE, while in terms of FID, the results are also similar for the DSBM solver but less similar for the ASBM solver. These results follow the **general logic** that the better the quality (generation or similarity) of the used coupling, the better the model will behave in this property during the iterations.
>
> **[W.4] Heavy notation reduces accessibility.**
> We appreciate this concern. In the paper, we adopt the discrete-time notation used in [ASBM], which is considerably simpler and more accessible than the continuous-time notation employed in [DSBM].
>
> **[W.5] Missing comparison with recent OT/consistency bridge baselines.**
> We would like to note that our aim is not to propose a new SB method (see our response to your comment **[W.1]**). Instead, our focus is to investigate the behavior of ASBM and DSBM (both state-of-the-art implementations of bidirectional IMF, i.e. IPMF) under different couplings. Within this scope, additional comparisons with other methods are not essential.
>
> **[Q.1] Is the convergence guarantee valid for non-Gaussian or continuous-time bridges?**
>
> Yes, indeed, the convergence is guaranteed for non-Gaussian bridges, as provided in our result concerning the discrete case (**Theorem 3.3**). Following your (and other reviewers') question, we extend the convergence proof to continuous time. Now we cover both cases, see our updated **Theorem 3.3**.
>
> **[Q.2] Can IPMF incorporate stochastic regularization (e.g., adversarial or entropy terms) without breaking convergence?**
>
> Would you be so kind as to clarify your question? Specifically, do you mean stochastic regularization in the sense of this paper:
>
> Song, Ki-Ung. "Applying Regularized Schr\"odinger-Bridge-Based Stochastic Process in Generative Modeling."

---

> > ### Author Response · Authors · 2025-11-23
> >
> > **[Q.3] Are there cases where separate IPF or IMF updates remain preferable in practice?**
> >
> > We are not aware of any cases in which one would prefer IPF or unidirectional IMF over IPMF.
> > As noted in **lines 159–161**, the former suffers from "prior forgetting", while unidirectional IMF fails to match the true target distribution $p_1$ due to error accumulation, as discussed in **line 367** (see also [DSBM, Appendix I]).
> >
> > **Concluding remarks**. We would be grateful if you could let us know if our explanations have been satisfactory. If so, we kindly ask that you consider increasing your rating. We are also open to discussing any other questions you may have.

---

> > > ### Comment · Reviewer_hGUD · 2025-11-28
> > >
> > > Dear Author,
> > >
> > > Thank you for your detailed response. I thoroughly read my previous concerns and your response in detail. I acknowledge my misunderstanding of your work—that it does not introduce new SB algorithms but focuses on investigating the properties of IPF and IMF and demonstrating convergence with any coupling. Some of my concerns about your work have been addressed, but I think it is very important to demonstrate the scalability of your work on arbitrary coupling with higher resolution. I read **Reviewer N3B2**’s comment: “Results on higher resolution data showing that those solvers work even with arbitrary couplings would substantially strengthen this paper.” I agree with the reviewer’s opinion. If the authors address this issue, preferably with recommendations related to the latent space, my concerns would be almost entirely addressed.

---

> > > > ### Author Response · Authors · 2025-12-03
> > > >
> > > > **New 512 x 512 AFHQ image results.** While the scalability of the DSBM and ASBM solvers for the Markovian projection part has already been investigated in the original papers [DSBM, ASBM], per the request to run experiments with higher image resolution, we provide results for the AFHQ [1] image dataset with $512 \times 512$ resolution images. Following [DSBM] paper we consider classes cat and wild. We run DSBM with IMF-OT and Identity couplings and present the results in Figure 6 and Table 3. **We observe a similar quality-similarity tradeoff as the Celeba setup**, i.e., the higher the similarity or generation quality of the coupling, the better the model performs on the corresponding metric.

---

### Author Response · Authors · 2025-11-23

Dear reviewers, thank you for taking the time to review our paper. We are glad that reviewer **hGUD** highlighted the value of our theoretical unification of IPF and IMF, and that reviewers **aDs4** and **N3B2** emphasized the significance of providing the first rigorous analysis of the bidirectional SB procedures used in practice. Reviewers **aDs4** and **6f73** also appreciated the clarity of the exposition, while reviewers **aDs4** and **N3B2** noted the insight our framework provides into mitigating the respective limitations of IMF and IPF. Finally, reviewers **aDs4**, **hGUD**, and **N3B2** acknowledged the breadth and factual nature of our empirical evaluation. Please, find the answers to your shared questions below.

**(1) Theoretical extension.** We extended the general convergence result (**Theorem 3.3**), now it covers both discrete- and continuous-time settings.

**(2) Extended analysis of experimental results.** We added the initial coupling metrics and visualization results for the CelebA experiment in **Section 4.4** to further support our new way of controlling the quality-similarity trade-off in SB models through designing couplings which become possible thanks to IPMF procedure.

**Revised paper.** To sum up, the main edits to the paper are highlighted in **blue color** (newly added):

- We moved the plot of quality and similarity metrics during IPMF iterations to the main text. Now they are provided in **Figure 5.**

- We added generation quality (FID) and input-output similarity of initial couplings in **Table 2** and images of initial coupling in **Figure 4.**

- The proof of continuous-time setting of **Theorem 3.3** is provided in **Appendix D.7.**

- **Table 3** clarifying the precise technical advances of IPMF (convergence rates, new unifying formulation, starting coupling analysis) over DSBM and ASBM.

- **Table 6** and **Figure 7** with the results of running DSBM with $\epsilon=10$ and Identity coupling.

---

### Author Response · Authors · 2025-12-03
**Rebuttal summary**

In this paper, we show that the heuristic bidirectional IMF procedure used in practice in fact uses IPF iterations. Therefore, we propose calling the heuristic bidirectional IMF procedure Iterative Proportional Markovian Fitting (IPMF). From a theoretical point of view, **we derive convergence guarantees** for Gaussian and bounded support cases. From a practical point of view, we show that this procedure provides **a new way to tune SB-based solvers** to trade-off generation quality and input-output similarity.

**Summary of the discussion before the data leak incident**.
We would like to note that before the end of the discussion, due to the incident, **the reviewer aDs4 increased the score to 6**. We also would like to cite the reviewer hGUD about the *large-scale experiments*: "If the authors address this issue, preferably with recommendations related to the latent space, my concerns would be almost entirely addressed."

**Overview of the reviewer questions** that we address in this rebuttal.

| Questions | Reviewer(s) | Responses |
|---|---|---|
| large-scale experiments? | hGUD, 6f73,N3B2 | While the scalability of the DSBM and ASBM solvers for the Markovian projection part  has already been investigated in the original papers [DSBM, ASBM],  per request **we provide results of IPMF on 512x512 images using AFHQ data in Section 4.4** of revised version.  |
| practical benefit of IPFM and different couplings? | hGUD, 6f73, aDs4 | Figure 5 with convergence plots shows that **IPMF allows to trade-off generation quality and input-output similarity** by choosing different couplings. |
| continuous-time IPMF? | hGUD, aDs4  | **We extended the general convergence result (Theorem 3.3)**, now it covers both discrete- and continuous-time settings. |
| new algorithm? | hGUD | We kindly remind you that the goal of this study is a theoretical analysis of heuristic bidirectional IMF (IPMF). Thus, we do not aim to introduce a novel algorithm. However, **the result of our analysis provides a practical benefit** in a novel way to trade-off generation quality and input-output similarity. |
| larger-eps for identity? | N3B2 | We provided additional results for DSBM and ASBM in Appendix E4. |
| table clarifying the advances of IPMF? | aDs4 | We provided Table 4 in Appendix A. |
| metrics of initial processes? | N3B2 | We provided metrics in Table 2 in the main text. |
| IPMF computationally heavier than IPF/IMF? | 6f73 | Our analysis shows that IPMF can be decomposed into a combination of IPF and IMF steps; however, this decomposition is used solely for theoretical analysis and does not alter the computational procedure (see **lines 342–346**). Furthermore, our experimental analysis of IPMF includes same couplings as IMF and IPF |
| generalization on other general distributions? | 6f73 | We provide convergence guarantees for the Gaussian and bounded support cases. Much real-world data like images, video or sounds, lies on a bounded support. |

**Revised paper.** The main edits to the paper are highlighted in **blue color** (newly added):

- We moved the plot of quality and similarity metrics during IPMF iterations to the main text. Now they are provided in **Figure 5.**

- We added generation quality (FID) and input-output similarity of initial couplings in **Table 2** and images of initial couplings in **Figure 4.**

- The proof of the continuous-time setting of **Theorem 3.3** is provided in **Appendix D.7.**

- **Table 4** clarifying the precise technical advances of IPMF (convergence rates, new unifying formulation, starting coupling analysis) over DSBM and ASBM.

- **Table 6** and **Figure 7** with the results of running DSBM and ASBM with $\epsilon=10$ and Identity coupling.

---

### Meta-Review · Area_Chair_Qfud · 2026-01-04

**Summary:**

The reviewers raise concerns related to: the generality of the theoretical results on the convergence analysis, the practical implications of the proposed work in terms of new algorithms and additional capabilities that these algorithms could have, the lack of the large-scale experiments, and finally the sensitivity of the numerical results with respect to hyperparameters. The authors addressed most of the reviewers concerns and I think that they did a great job in the rebutall phase. The work has theoretical and overall scientific merit and it is a step towards a deeper theoretical understanding of the relation Iterative Markovian Fitting and the Iterative Proportional Fitting method for solving Schrodinger Bridge problems.  For this reason I recommend acceptance of the paper.

**Reviewer Concerns:**

The authors address most of reviewers concerns during the rebuttal phase. The concern that I think still remains is with respect to the generality of the convergence proof in the case of general distributions. This is raised by reviewer aDs4 and hGUD.  Overall I think that providing a general proof for convergence in case of general non-Gaussian distributions should be considered as the next future step of this work. I think that the paper has enough theoretical analysis  and merit for the simpler (Gaussian Cases). From a methodological perspective, it also correct to aim for theoretical analysis on simpler versions of the problem in consideration and then slowing relaxing the assumption towards greater generality.  The authors are on the right path and the paper is a step towards this direction.

**Reviewer Scores:**

Reviewer hGUD would have definetely increased their score to 6 and above.  The reviewer also mentions the need to extra experiments that would have addressed completely his concerns. The authors provide these experiments.

Reviewer  aDs4 would have also increase the score to 6. This is also mentioned by the reviewer.

Reviewer 6f73 comments and overall review are short. I see no reason not to increase the score as most of reviewer's concerns are addressed by the authors.

Reviwer N3B2 score is already  6 and I see no reason for not increasing the score.

---

### Decision · Program_Chairs · 2026-01-26

Accept (Poster)